# Single cell multi-omic analysis identifies a *Tbx1*-dependent multilineage primed population in murine cardiopharyngeal mesoderm

Hiroko Nomaru[1], Yang Liu[1,10], Christopher De Bono[1,10], Dario Righelli[2,3], Andrea Cirino[4,5], Wei Wang[6], Hansoo Song [1], Silvia E. Racedo[1], Anelisa G. Dantas[1,7], Lu Zhang[8], Chen-Leng Cai[8], Claudia Angelini[2], Lionel Christiaen [6], Robert G. Kelly[9], Antonio Baldini[4,5], Deyou Zheng [1] & Bernice E. Morrow [1✉]

The poles of the heart and branchiomeric muscles of the face and neck are formed from the cardiopharyngeal mesoderm within the pharyngeal apparatus. They are disrupted in patients with 22q11.2 deletion syndrome, due to haploinsufficiency of *TBX1*, encoding a T-box transcription factor. Here, using single cell RNA-sequencing, we now identify a multilineage primed population within the cardiopharyngeal mesoderm, marked by *Tbx1*, which has bipotent properties to form cardiac and branchiomeric muscle cells. The multilineage primed cells are localized within the nascent mesoderm of the caudal lateral pharyngeal apparatus and provide a continuous source of cardiopharyngeal mesoderm progenitors. *Tbx1* regulates the maturation of multilineage primed progenitor cells to cardiopharyngeal mesoderm derivatives while restricting ectopic non-mesodermal gene expression. We further show that TBX1 confers this balance of gene expression by direct and indirect regulation of enriched genes in multilineage primed progenitors and downstream pathways, partly through altering chromatin accessibility, the perturbation of which can lead to congenital defects in individuals with 22q11.2 deletion syndrome.

[1] Department of Genetics, Albert Einstein College of Medicine, Bronx, NY, USA. [2] Institute for Applied Computing, National Research Council, Naples, Italy. [3] Department of Statistical Sciences, University of Padova, Padova, Italy. [4] Department of Molecular Medicine and Medical Biotechnology, University Federico II School of Medicine, Naples, Italy. [5] Institute of Genetics and Biophysics, National Research Council, Naples, Italy. [6] Center for Developmental Genetics, Department of Biology, New York University, New York, NY, USA. [7] Federal University of Sao Paulo, Sao Paulo, Brazil. [8] Department of Pediatrics, Indiana University School of Medicine, Indianapolis, IN, USA. [9] Aix-Marseille University, CNRS UMR 7288, IBDM, Marseille, France. [10] These authors contributed equally: Yang Liu, Christopher De Bono. ✉email: bernice.morrow@einsteinmed.org

The heart develops from two successive waves of meso-dermal progenitor cells during early embryogenesis. The first heart field (FHF) constitutes the first wave of cardiac progenitors and results in the primitive beating heart, while the second heart field (SHF) forms the second wave that builds upon the two poles of the heart[1,2]. The SHF can be anatomically partitioned to the anterior SHF (aSHF[3,4]) and posterior SHF (pSHF[4–6]), whose cells migrate to the heart via the dorsal pericardial wall to the outflow tract or inflow tract, respectively. Expression of *Mesp1* at gastrulation marks the earliest mesodermal cells that will form the heart[7]. Using single-cell RNA-sequencing (scRNA-seq) of the *Mesp1* lineage, it was discovered that the FHF, aSHF, and pSHF are specified at gastrulation[8].

Retrospective clonal analysis[9,10] and lineage tracing studies[11] revealed that the branchiomeric skeletal muscles (BrM) of the craniofacial region and neck share a clonal relationship with the SHF. The bipotent nature of these cardiac and skeletal muscle progenitor cells is supported by studies in the ascidian, *Ciona*, an invertebrate chordate, in which single cells gives rise to both cardiac and skeletal muscle cells[12]. When taken together, a new term, cardiopharyngeal mesoderm (CPM), was introduced to clearly include both SHF cardiac and skeletal muscle progenitor populations[2]. A cartoon of these populations in the mouse embryo is shown in Fig. 1a. The *Tbx1* gene, encoding a T-box transcription factor, and gene haploinsufficient in 22q11.2 deletion syndrome (22q11.2DS), is expressed in the CPM and is required for cardiac outflow tract and BrM development[2], implicating its essential roles in the CPM.

A total of 60–75% of patients with 22q11.2DS have cardiac outflow tract defects, which often require life-saving surgery during the neonatal period[13]. In addition, most individuals with this condition have speech, feeding, and swallowing difficulties in infancy, due in part to BrM hypotonia[14]. Further, heterozygous mutations of *TBX1* in rare, non-deleted individuals, phenocopy symptoms of the deletion[15]. Inactivation of both alleles of *Tbx1* in the mouse results in a persistent truncus arteriosus[16–18] and significant failure to form BrMs[19]. Gene expression profiling of wild-type versus *Tbx1* global null mutant embryos identified genes that changed in expression but it was unclear whether the changes were autonomous in the CPM or in other cell populations, such as neural crest cells[20,21]. We therefore do not yet understand the functions of *Tbx1* on a single-cell level, which is needed to elucidate the true molecular pathogenesis of 22q11.2DS.

The CPM is distributed throughout the embryonic pharyngeal apparatus during early gestation. The pharyngeal apparatus consists of individual bulges of cells termed arches that form in a rostral to caudal manner from mouse embryonic days (E)8–10.5. The formation of individual arches is a highly dynamic but regulated process that requires sufficient cell populations to elongate the OFT and to form diverse BrMs. The cellular and molecular mechanisms of how CPM cells in the pharyngeal apparatus both are maintained in a progenitor state and are allocated to form the heart and BrMs in mammals are unknown.

To fill these gaps, we performed scRNA-seq of mesodermal cell lineages from the pharyngeal apparatus plus heart at multiple stages during embryogenesis. We discovered a multilineage primed progenitor (MLP) population within the CPM, which is maintained and gradually matures from E8-10.5 and has differentiation branches toward cardiac and skeletal muscle fates, serving as common lineage progenitors. MLP cells are localized to the nascent lateral mesoderm of the pharyngeal apparatus, deploying cells to the heart and BrMs. We found that the *Tbx1* cell lineage marks MLPs and TBX1 activity is critical for their function. Inactivation of *Tbx1* disrupts a MLP gene expression program needed for differentiation and results in ectopic expression of non-mesoderm genes. We further identify the gene regulatory network downstream of *Tbx1* in the MLPs providing insights into the molecular mechanism of mammalian CPM function, essential for understanding the etiology of 22q11.2DS.

## Results

**Identification of common progenitor cells in the CPM**. To identify the various populations that constitute the CPM (Fig. 1a), we performed droplet-based scRNA-seq on fluorescence-activated cell sorted GFP expressing cells from microdissected *Mesp1^Cre^;ROSA26-GFP^f/+^* (f, flox) embryos at E8.0, E8.25, E9.5, and E10.5 (Table 1, Fig. 1b, Supplementary Figs. 1a, b, 2a, b, 3, and Supplementary Table 1). These stages were chosen because they are the critical periods when *Tbx1* is expressed, reaching the highest expression at E9.5, and when the pharyngeal apparatus is dynamically elongating; this is coordinated with heart development and BrM specification. To better understand the developmental sequence of events, we integrated the four time point datasets (Fig. 1c) and identified 20 cell clusters (Fig. 1d). Utilizing knowledge of the expression of known marker genes in each cluster (Fig. 1e, Supplementary Fig. 2c–e, and Supplementary Data 1), we identified cell types of all the clusters, half of which include cardiovascular progenitor cell populations or their derivatives (Fig. 1d, bold font).

We identified CPM clusters (C1, C3, C9, C15, and C18) with marker gene expression for the CPM including *Tbx1, Isl1, Wnt5a,* and *Tcf21*, among other genes[1,22] (Fig. 1f, g and Supplementary Fig. 2e). Cluster C9 contains BrM progenitor cells identified by expression of *Tcf21, Lhx2,* and *Myf5*[22] (Fig. 1f and Supplementary Fig. 2e). Clusters C1 and C18 contain pSHF populations as identified by expression of *Hoxb1*[5], *Tbx5, Foxf1,* and *Wnt2*[23,24] (Fig. 1f and Supplementary Fig. 2e). Many of the pSHF cells are located more medially and caudally in the embryo and contribute to posterior organ development, such as the formation of the lung[23,24]. The cells in C3 express *Nkx2-5* and *Mef2c*[25] and cardiac structural protein genes (*Tnnt2, Tnni1, Myl4* (Fig. 1f and Supplementary Fig. 2e)). In addition, subdomains express either FHF genes (*Tbx5* but not *Isl1* or *Tcf21*) or aSHF genes (*Fgf10, Isl1, Tcf21* but not *Tbx5* (Fig. 1h)). We discovered that cluster C15 expresses genes shared by CPM clusters, including *Tbx1, Isl1, Wnt5a, Mef2c, Tcf21,* and *Foxf1* (Fig. 1f and Supplementary Fig. 2e). Based upon this, we suggest that C15 as a multilineage primed progenitor (MLP) population within the CPM.

**Multilineage primed progenitors of the CPM differentiate into cardiac and skeletal muscles**. We next investigated the relationship between the MLPs and more differentiated CPM cells using partition-based graph abstraction (PAGA)[26]. Several clusters that are not part of the CPM and were already well separated in the above cluster analysis (C4, C5, C7, C13, and C17), were excluded from PAGA analysis (Supplementary Fig. 2d). The PAGA analysis partitioned the CPM cells into five branches (Fig. 2a and Supplementary Fig. 4a), connecting all mesodermal cell populations. The five branches include cardiomyocytes (CMs), pSHF with lung progenitor cells (pSHF, Lung PC), connective tissue (CT), branchiomeric muscle progenitor cells (BrM), and skeleton/limb (Sk/L). Convergent results from pseudotime analysis (Fig. 2b) and real-time point information (Fig. 2c and Supplementary Fig. 4b), infer that MLP cells (C15) in the center are in a progenitor state of the CPM while more differentiated cells are toward the outside in each branch (BrM, C9; CMs with aSHF, C3; part of the pSHF, C1 + C18; Fig. 2d and Supplementary Fig. 4c). The MLP cells exist not only in early time points (E8.0, E8.25) but also in later time points (E9.5, E10.5), but with fewer cells at E10.5 (Fig. 2e).

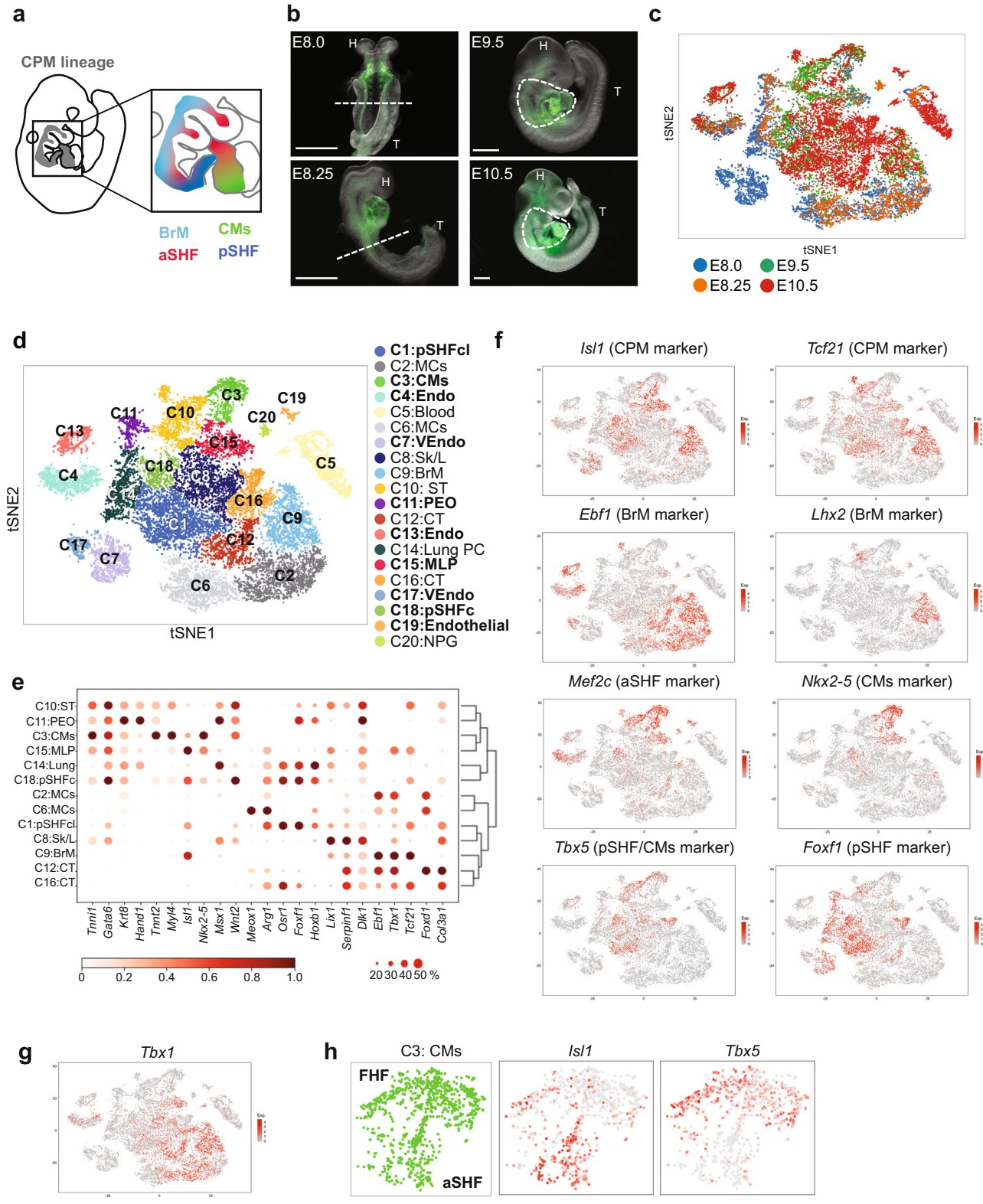

Based on the specific expression of genes (Supplementary Data 1) and distribution of gene expression patterns in the PAGA plot (Fig. 2f and Supplementary Fig. 4d–f), we focused on two marker genes for MLPs, *Aplnr* (Apelin receptor) and *Nrg1* (Neuregulin 1). *Aplnr* is expressed in the CPM[27] but not known for the specific presence in MLPs, while *Nrg1* is not known to be a CPM gene, but it is required in the embryonic heart for the development of the

chamber myocardium[28]. We examined the co-expression of genes in the cells in CPM populations. The heatmap in Fig. 2g (Supplementary Fig. 4e, f) shows the expression of genes enriched in MLPs, with the same genes also expressed in more differentiated CPM populations (BrM, CMs, and pSHF), but at different expression levels, indicating that they are multilineage primed. Taken together, these results further support the progenitor status of MLPs.

**Fig. 1 Single-cell analysis of *Mesp1*+ lineages at E8.0 to E10.5 identifies CPM lineages. a** Cartoon of an E9.5 embryo in a right lateral view (left, CPM, gray). The CPM includes the branchiomeric muscle progenitor cells (BrM; aqua), anterior SHF (aSHF; red), and part of the posterior SHF (pSHF; blue). Because cardiomyocyte progenitors (CMs; green) are from both the FHF and CPM, only CPM-derived CMs are included. **b** Whole-embryo images with GFP fluorescence of *Mesp1^Cre*; *ROSA26-GFP^{f/+}* embryos at E8.0, E8.25, E9.5, and E10.5 used for scRNA-seq. The white dotted line represents the region that was dissected. Only GFP-positive cells were used for scRNA-seq. H head, T tail. Scale bar E8.0, E8.25 is 750 μm; E9.5 and E10.5 is 500 μm. **c** t-distributed stochastic neighbor embedding (tSNE) plot colored by developmental stage. **d** tSNE plot colored by clusters (C1-20). Cardiac relevant clusters are in bold font. C1: pSHFcl, posterior CPM includes cardiac and lung progenitor cells; C2: MCs, mesenchyme expressing epithelial–mesenchymal transition markers; C3: CMs cardiomyocyte progenitor cells; C4: Endo endocardium and endothelial cells; C5: Blood blood cells; C6: MCs mesenchyme expressing epithelial–mesenchymal transition markers; C7: VEndo vascular endothelial cells; C8: Sk/L skeleton/limb progenitor cells; C9: BrM branchiomeric muscle progenitor cells; C10: ST septum transversum progenitor cells; C11: PEO proepicardial organ; C12: CT connective tissue progenitor cells; C13: Endo endothelial cells; C14: Lung PC lung progenitor cells; C15: MLP multilineage progenitor cells; C16: CT connective tissue progenitor cells; C17: VEndo vascular endothelial cells; C18: pSHFc posterior CPM of cardiac progenitor cells; C19: Endothelial cells; C20: NPG neural progenitor cells. **e** Heatmap of average gene expression of marker genes in representative clusters. Dot size indicates the fraction of cells expressing the genes in each cluster and color indicates scaled mean expression (bar below; dark red is the strongest expression). **f, g** tSNE plots showing expression of CPM marker genes (**f**) and *Tbx1* (**g**). The color spectrum from gray to red indicates expression levels from low to high. **h** tSNE plots of CM progenitors showing the expression level of CM (*Tbx5*) and CPM (*Isl1*) marker genes.

**Table 1 Summary of scRNA-seq samples.**

| Time point | No. of embryos | Somites | Cell viability | Cell number (captured) | Mean reads/cell | Median genes/cell | Tissue |
|---|---|---|---|---|---|---|---|
| *Mesp1^Cre*; *ROSA26-GFP^{f/+}* | | | | | | | |
| E8.0 | 7 | 3–4 | 90% | 7581 | 58,422 | 4380 | Half of the body |
| E8.25 | 5 | 6–7 | 95% | 2956 | 151,357 | 5363 | Half of the body |
| E9.5 | 6 | 19–22 | 80% | 4055 | 80,919 | 4036 | PA and heart |
| E9.5 | 2 | 21–22 | 81% | 7418 | 60,500 | 4106 | PA and heart |
| E10.5 | 4 | 30–31 | 80% | 8070 | 52,326 | 3673 | PA and heart |
| *Mesp1^Cre*;*ROSA26-GFP^{f/+}*;*Tbx1^{f/f}* | | | | | | | |
| E9.5 | 1 | 22 | 83% | 5157 | 65,997 | 5015 | PA and heart |
| E9.5 | 2 | 24 | 92% | 10,281 | 43,868 | 3617 | PA and heart |
| *Tbx1^{Cre/+}*; *ROSA26-GFP^{f/+}* | | | | | | | |
| E8.5 | 7 | 8–9 | 88% | 1434 | 136,347 | 3683 | Half of the body |
| E9.5 | 10 | 19–21 | 87% | 5453 | 54,317 | 3994 | PA and heart with back |
| *Tbx1^{Cre/f}*; *ROSA26-GFP^{f/+}* | | | | | | | |
| E8.5 | 4 | 9–10 | 92% | 1090 | 140,352 | 4374 | Half of the body |
| E9.5 | 3 | 19–22 | 97% | 6267 | 54,892 | 3561 | PA and heart with back |
| *Tbx1−/−* vs wild-type | | | | | | | |
| E9.5 WT | 4 | 22–23 | 89% | 12,050 | 37,218 | 3757 | PA and heart with back |
| E9.5 KO | 4 | 20–23 | 92% | 11,504 | 37,646 | 4138 | PA and heart with back |

**The MLPs are bilaterally localized to the caudal pharyngeal apparatus**. To elucidate whether MLPs are localized within a defined embryonic region in the pharyngeal apparatus, we performed RNAscope in situ hybridization analysis using probes for genes enriched in MLPs including *Tbx1*, *Isl1*, *Aplnr*, and *Nrg1* with the *Mesp1*+ lineage marked by *EGFP* expression (Fig. 2h–m). The pharyngeal arches form in a rostral to the caudal manner where the most caudal and lateral mesoderm is the least differentiated, while the rostral mesoderm has already migrated to the core of the arch to form BrM progenitor cells or toward the dorsal pericardial wall and poles of the heart[29,30]. In embryos at E8.5 and E9.5, *Nrg1* and *Aplnr* co-expressing cells are found bilaterally in the lateral part of the caudal pharyngeal apparatus, containing nascent mesoderm that is not yet differentiated to cardiac or skeletal muscle (Fig. 2h–j and Supplementary Fig. 5a, b). At E8.5, *Nrg1* and *Aplnr* are expressed in these regions within the forming second arch (Fig. 2i) and at E9.5, by the forming fourth arch (Fig. 2j), both overlapping with *Isl1* expression. The *Mesp1*+ lineage is marked with *EGFP* expression in the second arch at E8.5 and the fourth arch at E9.5 (Fig. 2k and Supplementary Fig. 5c, d). *Tbx1* and *Isl1* are also expressed in those regions (Fig. 2l, m). We suggest that the MLPs remain in the same region of the caudal pharyngeal apparatus, while they deploy cells rostrally, medially, and dorsally thereby explaining in part the

mechanism for the extension of the pharyngeal apparatus caudally (Fig. 2n).

**MLPs dynamically transition over time**. An important question is whether MLPs as CPM progenitors, maintain the same state of gene expression over time. To address this, we examined differentially expressed genes in MLPs from E8-10.5. We identified core CPM genes that are expressed similarly at all time points, including *Isl1*, *Mef2c*, and *Nkx2-5* (Fig. 3a). However, we also found that early expressing genes such as *Aplnr*, *Nrg1*, *Irx1-5*, *Fgf8/10*, and *Tbx1* (Fig. 3b) are reduced over time, with increasing expression of cardiac developmental genes such as *Hand2*, *Gata3/5/6*, *Bmp4*, and *Sema3c* (Fig. 3c). These gradient differences are also visualized in violin plots (Fig. 3d–f). *Nkx2-5* and *Sema3c* are expressed in the caudal pharyngeal apparatus at E9.5, like that of *Isl1*, *Tbx1*, and *Aplnr*, defining MLPs (Fig. 3g). The MLP region is reduced in size in the caudal pharyngeal apparatus, pharyngeal arches 3–6, at E10.5 (Fig. 3g). Further, expression of the early-MLP marker *Aplnr* is not observed at E10.5, while co-expression of *Sema3c*, *Nkx2-5*, and *Isl1* occur strongly in the OFT (Fig. 3g). This is consistent with the model that the MLPs continuously allocate progenitor cells to BrMs and OFT-CMs, while showing a gradual maturation themselves by E10.5 (Fig. 3h).

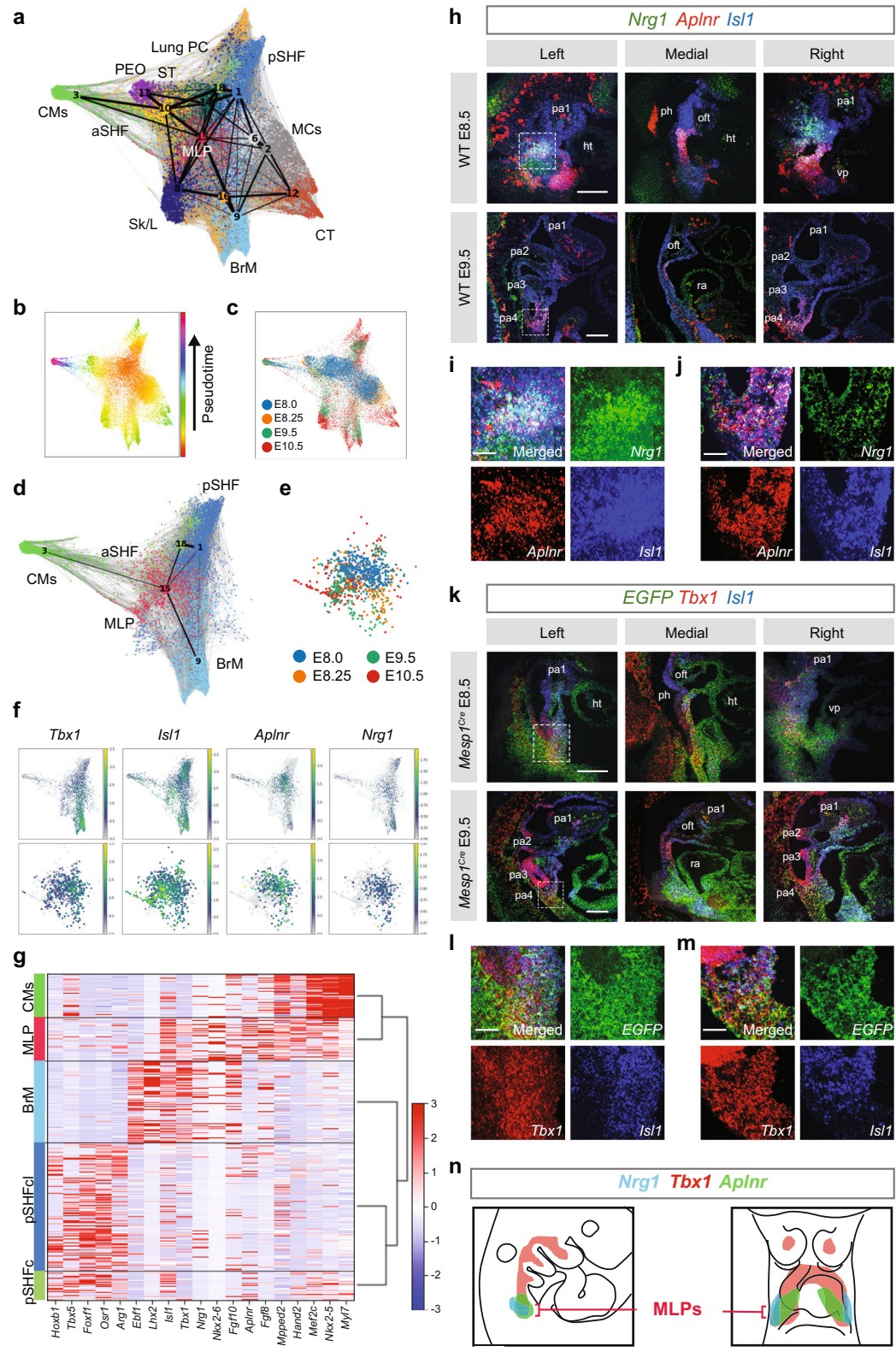

**The intersection of the *Tbx1* and *Mesp1* lineages helps to identify the CPM**. *Tbx1* function in MLPs and roles in derivative CPM cells on a single-cell level are unknown. We therefore examined the *Mesp1* and *Tbx1* lineages in control embryos to understand how the CPM lineages compare in relation to *Tbx1*. The *Mesp1* lineage contributes more broadly to the embryonic mesoderm, while *Tbx1* is expressed in pharyngeal endoderm and

distal pharyngeal ectoderm[31], in addition to the CPM[32]. Although *Tbx1* is strongly expressed in the CPM, it is not expressed in the heart, neither in the FHF nor the caudal and medial pSHF at the timepoints analyzed[33–35]. The intersection of these two datasets defines the CPM more precisely.

The *Tbx1^Cre^* allele has a knockin of *Cre* that inactivates one copy of the *Tbx1* gene. The *Tbx1^Cre^* mice do not have cardiac or

**Fig. 2 MLPs are localized to the caudal pharyngeal apparatus. a** Single-cell embedding graph with PAGA plot colored by clusters with abbreviated names identical to those in Fig. 1d. **b, c** Single-cell embedding graph colored by pseudotime with colors indicating advancing time (**b**) or real time of samples (**c**). **d** Single-cell embedding graph with PAGA plot of the CPM clusters. **e** Single-cell embedding graph of only MLPs colored by stage. **f** Single-cell embedding graph of the CPM (above) and MLPs (below) colored by gene expression level (gray to blue, green, yellow, low to high). **g** Heatmap of expression of genes enriched in the CPM. Row indicates the expression per cell (blue to white, to red; low to high). **h–j** RNAscope in situ hybridization with *Nrg1* (green), *Aplnr* (red), and *Isl1* (blue) mRNA probes on sagittal sections from wild-type embryos at E8.5 (upper panel; n = 3) and E9.5 (lower panel; n = 3). Scale bar, 100 μm. The white dotted line indicates the position in images with higher magnification (**i, j**; scale bar, 30 μm). **k–m** RNAscope in situ hybridization with *EGFP* (green) mRNA probe, marking *Mesp1*+ lineage cells, *Tbx1* (red) and *Isl1* (blue) mRNA probes on sagittal sections from *Mesp1^Cre^; ROSA26-GFP^f/+^* embryos at E8.5 (upper panel; n = 3) and E9.5 (lower panel; n = 3). Scale bar, 100 μm. The white dotted line indicates the position in images with higher magnification (**l, m**; scale bar, 30 μm.). **n** Summary of the expression pattern of *Nrg1, Tbx1,* and *Aplnr* with respect to the MLPs in the cartoon of a sagittal section. ht heart, oft outflow tract, pa pharyngeal arch, ph pharynx, ra right atrium, vp venous pole. 1, 2, and 3 indicate the first, second, and third pharyngeal arches, respectively.

aortic arch defects in the SwissWebster background[36]. We then performed scRNA-seq of the *Tbx1^Cre^* lineage at E9.5 and integrated this with data from the *Mesp1^Cre^* lineage at the same stage to compare the characteristics of the two lineages (Fig. 4a–c and Supplementary Data 2). Data integration provides consistency in defining common cell types among different samples, in addition to removing batch effects[37]. We found that the CPM can be identified in both populations (Fig. 4a, bold font). As expected, the *Mesp1* lineage includes the FHF, which is not included in the *Tbx1* lineage, while the *Tbx1* lineage includes the pharyngeal epithelia and otic vesicle, not included in the *Mesp1* lineage (Fig. 4b, c). The relative proportions of CPM populations are shown in Fig. 4d. The pSHF in the *Mesp1^Cre^* lineage includes the caudal pSHF with lung progenitors that is not included in the *Tbx1^Cre^* lineage[31]. The MLPs are found in both lineages, marked by the expression of *Isl1, Aplnr, Tbx1,* and *Nrg1* (Fig. 4e). Therefore, the data from scRNA-seq using *Tbx1^Cre^* help define the CPM better and serves as a replication for the data on the CPM from scRNA-seq using *Mesp1^Cre^*, which is relevant to *Tbx1* as shown in Fig. 4f.

**Tbx1 regulates MLP development by promoting gene expression needed for differentiation and restricting the expression of non-mesodermal genes.** *Mesp1^Cre^* and *Tbx1^Cre^* mediated *Tbx1* conditional null embryos have similar phenotypes, including hypoplasia of the caudal pharyngeal apparatus and a fully penetrant persistent truncus arteriosus[16,32]. This supports the utility of both *Cre* lines in this study and the importance of the mesoderm domain of *Tbx1* expression in mediating its function. We inactivated *Tbx1* in the *Mesp1* and *Tbx1* lineages (Fig. 5a, b) and performed scRNA-seq to ascertain how its loss affects the CPM, using both lineages as a comparison and replication for each other (Table 1). We generated embryos that were *Mesp1^Cre/+^;Tbx1^+/+^* (*Mesp1^Cre^* Ctrl) vs *Mesp1^Cre/+^;Tbx1^f/f^* (*Mesp1^Cre^* cKO) at E9.5 (Supplementary Fig. 1a, b, Fig. 5a, and Table 1) and performed two biological replicates. We also generated *Tbx1^Cre/+^* (*Tbx1^Cre^* Ctrl) vs *Tbx1^Cre/f^* (*Tbx1^Cre^* cKO) embryos at E8.5 and E9.5 (Supplementary Fig. 1c, d, Fig. 5b, and Table 1). The E8.5 stage used (8–10 somites) is only very slightly different from the E8.25 stage (6–7 somites) used for the *Mesp1^Cre^* experiment (Table 1). As above, the *ROSA26-GFP^f/+^* reporter was used to purify the lineages (Supplementary Fig. 1).

With the conditional knockout data, we performed two integrated and clustering analyses; one for *Mesp1^Cre^* Ctrl vs *Mesp1^Cre^* cKO (two replicates; Supplementary Fig. 6 and Supplementary Data 3) and another for *Tbx1^Cre^* Ctrl vs *Tbx1^Cre^* cKO embryos (Supplementary Fig. 7 and Supplementary Data 4–5), and, then again focusing on the CPM. With increased resolution afforded by the single time point experiments, we found that the aSHF population is continuous with the somatic mesoderm (SoM; Fig. 5c–f). The SoM is adjacent to the aSHF and

in embryos and gives rise to the ventral pericardial wall (expressing *Msx1, Msx2, Epha3,* but not *Nkx2-5*;[38] *Nkx2-5* and *Msx2* are shown in Fig. 5g, h).

The ratio of the number of MLPs compared to the total number of cells in each replicate of *Mesp1^Cre^* Ctrl versus cKO shows a modest variation (first replicate of WT (WT1) is 0.07 (284/4046 cells), WT2 is 0.129 (912/7048) vs first replicate of cKO (KO1) is 0.101 (474/4689) and KO2 is 0.108 (1056//9750)), nonetheless, MLPs are present in both (Fig. 5c–f). To understand how *Tbx1* affects gene expression within the MLPs and derivative cell types leading to the observed phenotypes at later stages, we analyzed differentially expressed genes (DEGs) with the scRNA-seq datasets. We identified DEGs in each cluster in the two replicates of *Mesp1* Ctrl vs cKO embryos (Supplementary Data 4) and separately DEGs in *Tbx1^Cre^* Ctrl vs *Tbx1^Cre^* cKO embryos at E9.5 (Supplementary Data 6). The DEGs in the two *Mesp1^Cre^* Ctrl vs cKO replicates overlapped significantly, and the final DEGs were from the pooled replicates. Furthermore, to focus on the most reproducible alterations, we examined only DEGs shared in both *Mesp1* replicates and between *Mesp1* and *Tbx1* scRNA-seq datasets at E9.5, which change in the same direction, with adjusted *P* value < 0.05 and have an absolute log2-fold change >0.25 (Fig. 6a). In the MLPs, we identified 651 DEGs; 468 genes were decreased and 183 were increased in both *Mesp1* and *Tbx1* cKO embryos at E9.5 (Fig. 6b, Supplementary Data Gene ontology (GO) analysis was used to identify enriched biological functions of the downregulated genes shared in both *Mesp1* and *Tbx1* cKO embryos (Supplementary Data 8). Genes affected in MLPs are involved in cell migration, organ development, and muscle development (Fig. 6c; e.g., *Sox9, Mef2c, Grem1, Hey1, Bmp7, Fgf10; Foxf1, Wnt5a*). *Mef2c*[39], *Hey1*[40], *Bmp7*[41], *Fgf10*[3], and *Wnt5a*[42,43] are required for normal cardiac development. Enriched GO terms for the upregulated genes included axonogenesis such as *Bdnf*[44] or *Pax6*[45], inner ear development (placodal formation), and cell fate determination (Fig. 6d). Further, genes not normally expressed in the mesoderm at detectable levels were expressed, such as *Pax8*[46] (Fig. 6d). The ratio of the number of *Isl1* and *Aplnr* expressing MLPs in *Mesp1^Cre^* Ctrl embryos was reduced in *Tbx1* cKO embryos (WT1, 0.246 (70/284 cells); WT2, 0.214 (195/912) vs cKO1, 0.011 (5/474); cKO2, 0.004 (5/1,056)), while the ratio of the number *Pax8* expressing MLPs was dramatically increased (WT1, 0.011 (3/284 cells), WT2, 0.003 (3/912); cKO1, 0.395 (187/474); cKO2, 0.181 (191/1,056)), further indicating functional changes in MLPs.

To confirm expression changes from the scRNA-seq experiments when *Tbx1* is inactivated, we checked the expression pattern of *Aplnr* and *Pax8* in vivo using RNAscope analysis. These genes are at the top fold change among DEGs decreased (*Aplnr)* or increased (*Pax8*) in the MLPs of *Mesp1^Cre^* cKO vs Ctrl embryos, with consistent results in *Tbx1^Cre^* Ctrl vs cKO embryos (Supplementary Data 7). Ectopic *Pax8* expression was observed in

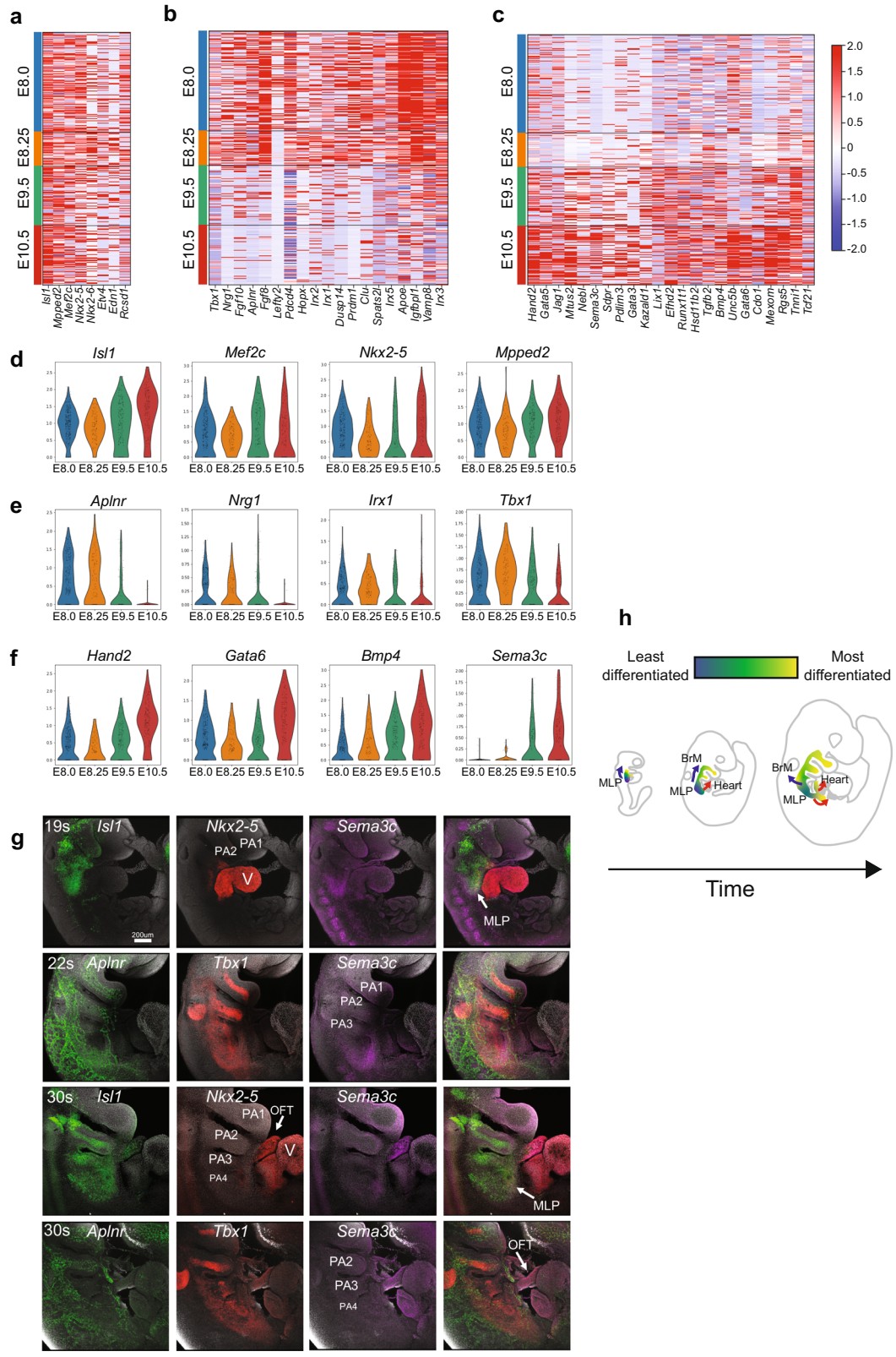

the MLPs of *Mesp1^{Cre}* and *Tbx1^{Cre}* cKO embryos (Fig. 6e, f and Supplementary Fig. 8). In serial transverse sections of both sets of cKO and control embryos at E9.5, reduced levels of *Aplnr* expression in the MLPs in the posterior pharyngeal apparatus were observed, while *Pax8* was increased in the same region containing EGFP + cells (Fig. 6g, h and Supplementary Fig. 9).

We next performed scRNA-seq of wild-type versus *Tbx1^{−/−}* embryos at E9.5, without cell sorting, as an independent replication for these experiments, and we found similar gene expression changes as when *Mesp1^{Cre}* or *Tbx1^{Cre}* lines were used for gene inactivation (Supplementary Fig. 10). We note that in this experiment, the ability to identify individual cell types is reduced

**Fig. 3 MLPs transition as cells are allocated to more differentiated states over time. a** Heatmap of expression of core genes in MLPs at E8, E8.25, E9.5, and E10.5. Row indicates the expression of each cell. **b** Heatmap of expression of the genes enriched in expression in earlier stage-MLPs (E8, E8.25) and shown in all four stages. Row indicates the expression of each cell. **c** Heatmap of expression of the genes enriched in expression in intermediate (E9.5) and later stage MLPs (E10.5) and shown in all four stages. Row indicates the expression in each cell. **d** Violin plots of the expression of core genes (*Isl1, Mef2c, Nkx2-5, Mpped2*) in MLPs over time. **e** Violin plots of expression of early-MLP genes (*Aplnr, Nrg1, Irx1, Tbx1*) in MLPs over time. **f** Violin plots of expression of late-MLP genes (*Hand2, Gata6, Bmp4, Sema3c*) in MLPs over time. **g** Whole embryo RNAscope with *Isl1* (green), *Nkx2-5* (red), *Sema3c* (purple), *Aplnr* (green), and *Tbx1* (red) at E9.5 (19 and 22 somites) and E10.5 (30 somites) to identify MLPs in the distal pharyngeal apparatus (*n* = 3 for each stage). Images on the right of each row are a composite of all three probes. V ventricle, OFT cardiac outflow tract, MLP, PA1-4 are indicated. Scale bar, 200 μm. **h** Cartoon of MLP transitions and cell allocation over time. The MLPs in the nascent pharyngeal mesoderm migrate dorsally and will differentiate to BrMs or ventrally and medially to CMs. Blue arrow indicates migration to form BrMs and red arrow(s) indicates migration to the poles of the heart to form CMs. The color spectrum from blue to yellow indicates differentiation from MLPs to their derivative cell types.

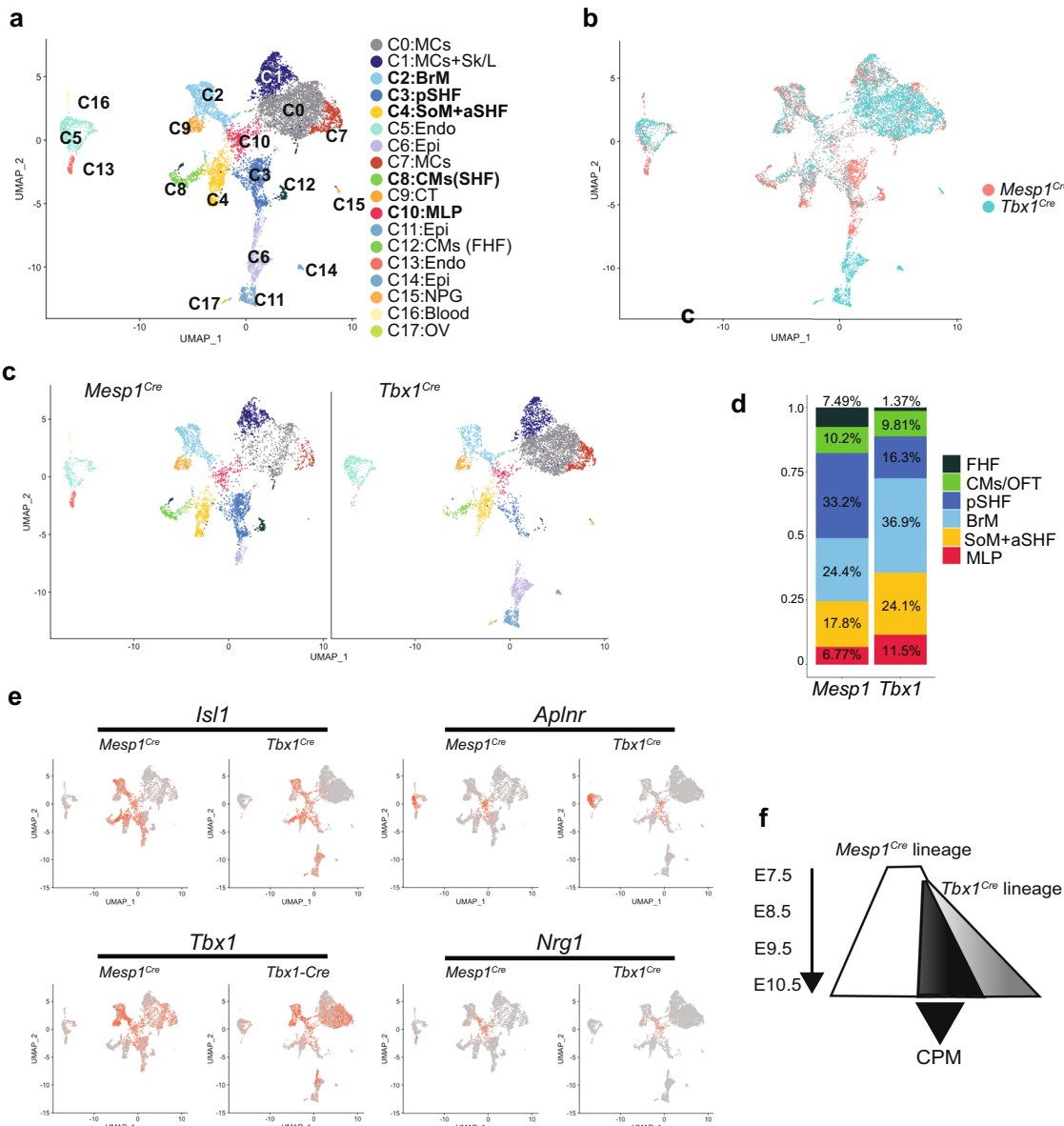

**Fig. 4 CPM clusters were found in both *Mesp1^{Cre}* and *Tbx1^{Cre}* lineages. a** UMAP plot of integrated data of *Mesp1^{Cre/+}* and *Tbx1^{Cre/+}* scRNA-seq data at E9.5, colored by cluster. CPM clusters are shown in bold font. **b** UMAP plot colored by samples (*Mesp1^{Cre}*, coral; *Tbx1^{Cre}*, aqua). **c** UMAP plots, colored by cluster, separated by samples. **d** The ratio of cell populations of CPM lineages in *Mesp1^{Cre/+}* and *Tbx1^{Cre/+}* scRNA-seq data. A two proportion *Z* test was performed in each cluster with 95% confidence interval (MLP: *P* value = 2.27e-13; pSHF: *P* value = 2.2e-16; SoM/aSHF: *P* value = 2.72e-5; OFT + CMs: *P* value = 0.70). **e** UMAP plots for showing MLP marker genes, separated by samples. The color spectrum from gray to red indicates expression level from low to high. **f** Intersection of scRNA-seq data between the two *Mesp1^{Cre}* and *Tbx1^{Cre}* populations. Black triangle shows genes shared in both populations and this represents the CPM. Genes shared in both populations have been used for further study in this report. Genes expressed in one versus the other have not been further investigated.

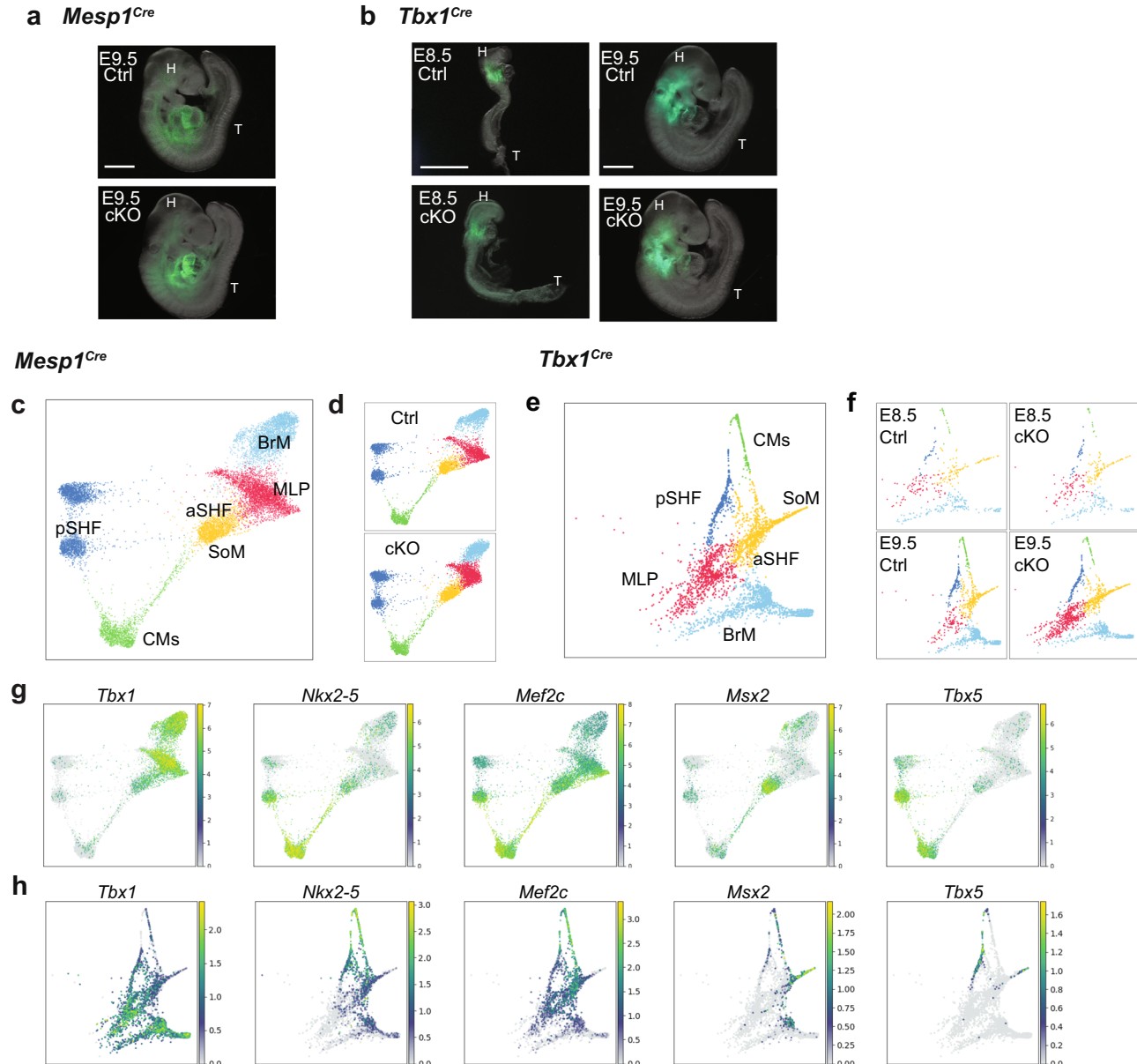

**Fig. 5 *Tbx1* is required for the progression of MLPs to more differentiated states. a** Whole-mount embryo images with GFP fluorescence of *Mesp1^Cre^; ROSA26-GFP^f/+^* (Ctrl; upper panel, similar to Fig. 1a) and *Mesp1^Cre^; ROSA26-GFP^f/+^;Tbx1^f/f^* (cKO; lower panel) embryos at E9.5 used for scRNA-seq. The pharyngeal apparatus and heart were collected. The sorted GFP-positive cells were used for scRNA-seq. H head, T tail. Scale bar, 500 μm. **b** Whole-embryo images with GFP fluorescence of *Tbx1^Cre/+^; ROSA26-GFP^f/+^* (Ctrl; upper panels) and *Tbx1^Cre/f^; ROSA26-GFP^f/+^* (cKO; lower panels) embryos at E8.5 (left) and E9.5 (right) used for scRNA-seq. The upper half of the body was collected at E8.5. The pharyngeal apparatus and heart were collected at E9.5. Sorted GFP-positive cells were used for scRNA-seq. Scale bar E8.5 is 400 μm; E9.5 is 500 μm. **c** Single-cell embedding graph of the CPM populations from the integration of two replicates of *Mesp1^Cre^* Ctrl vs cKO datasets colored by clusters. Definitions of clusters are the same as in Fig. 1, with the addition of SoM, somatic mesoderm. **d** Single-cell embedding graph separated by genotype. Upper panel, *Mesp1^Cre^* Ctrl dataset. Lower panel; *Mesp1^Cre^* cKO dataset. **e** Single-cell embedding graph of integration of CPM lineages from *Tbx1^Cre^* Ctrl and cKO datasets at E8.5 and E9.5 colored by cell clusters. **f** Single-cell embedding graph separated by samples. Upper left and right, *Tbx1^Cre^* Ctrl and *Tbx1^Cre^* cKO dataset at E8.5. Lower left and right, *Tbx1^Cre^* Ctrl and *Tbx1^Cre^* cKO dataset at E9.5. **g, h** Single-cell embedding graph of the CPM lineages from integrated *Mesp1^Cre^* Ctrl and cKO datasets (**g**) and *Tbx1^Cre^* Ctrl and cKO datasets (**h**) colored by expression level (gray to blue, green, yellow, low to high).

due to the presence of other cell populations, nevertheless, we found similar changes in MLPs with this independent comparison. Overall, the three independent scRNA-seq datasets suggest that *Tbx1* promotes lineage maturation but restricts ectopic expression of non-mesodermal genes in MLPs.

In addition to MLPs, we examined DEGs in other CPM populations (BrM, SoM+aSHF, pSHF, and CMs; Supplementary Fig. 11 and Supplementary Data 9). Of interest, *Tbx1* is expressed strongly in the BrM populations. We found that there are 667 genes decreased and 163 increased in expression in the BrM population at E9.5 (Supplementary Fig. 11a). *Lhx2* and *Myf5* are representative genes that are specific for and downregulated only in the BrM populations (Supplementary Data 9). This suggests the potential of direct transcriptional regulation. We also identified DEGs in other CPM-derivative populations, where *Tbx1* is not strongly expressed (Supplementary Fig. 11), and examined GO

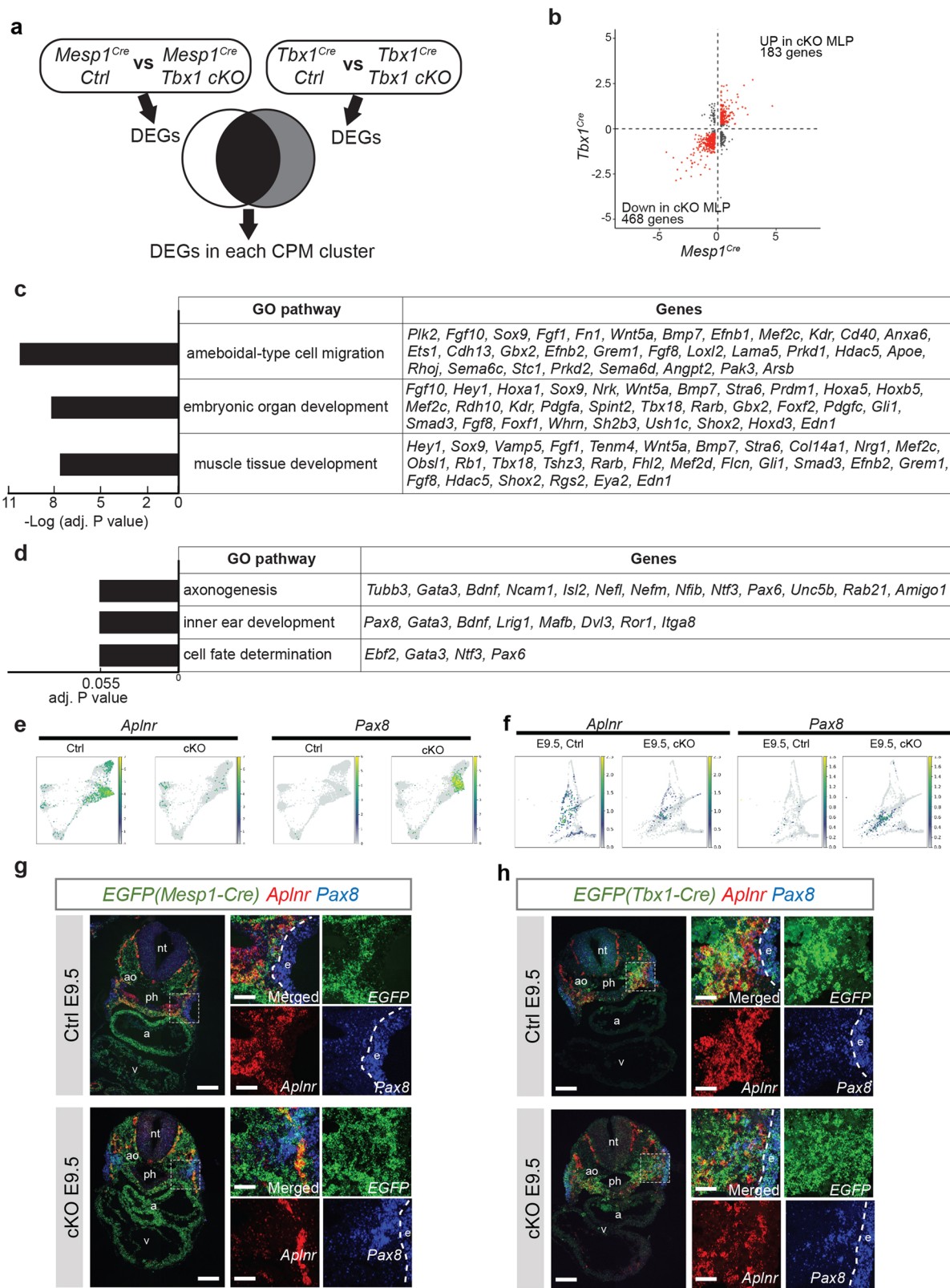

pathways affected (Supplementary Fig. 12 and Supplementary Data 10). We compared DEGs specific for MLP versus derivative CPM populations (Supplementary Fig. 13a). Representative DEGs are decreased in MLPs and derivative cells include *Bmp7*, *Fgf10*, and *Foxf1*, suggesting that these changes continue in derivative populations (Supplementary Fig. 13b–d). Other DEGs are reduced only in derivative CPM populations, even where *Tbx1* is not strongly expressed (Supplementary Fig. 13a), suggesting indirect regulation and changes in cell fate acquisition.

**TBX1 defines a gene regulatory network in the MLPs for cardiac and BrM formation.** To better understand how TBX1

**Fig. 6 *Tbx1* is required in MLPs to promote lineage progression and restrict the expression of ectopic genes. a** The cartoon of analysis strategy focus on DEGs in the CPM that are shared between both lineages (black). **b** Comparison of differentially expressed genes (DEGs) from *Mesp1*$^{Cre/+}$; *ROSA26-GFP*$^{f/+}$ (Ctrl) vs *Mesp1*$^{Cre/+}$; *ROSA26-GFP*$^{f/+}$;*Tbx1*$^{f/f}$ (cKO) embryos at E9.5 and *Tbx1*$^{Cre/+}$; *ROSA26-GFP*$^{f/+}$ (Ctrl) vs *Tbx1*$^{Cre/f}$; *ROSA26-GFP*$^{f/+}$ (cKO) in MLPs. *X* axis indicates log2-fold change of *Mesp1*$^{Cre}$ DEGs. *Y* axis indicates the log2-fold change of *Tbx1*$^{Cre}$ DEGs. Each dot indicates a gene, and red indicates common DEGs with |log2-fold change| > 0.25 in both comparisons. **c, d** Enriched biological processes in gene ontology (GO) and related genes found in downregulated (**c**) or upregulated (**d**) genes in MLPs of *Tbx1* cKO. *X* axis indicates adjusted *P* values. **e** Single-cell embedding graph of the CPM lineages from *Mesp1*$^{Cre}$ Ctrl and cKO at E9.5 separated by genotype and colored by expression level (gray to blue, green, yellow, low to high). **f** Single-cell embedding graph of the CPM lineages from *Tbx1*$^{Cre}$ Ctrl and cKO at E9.5 datasets separated by genotype and colored by expression level (gray to blue, green, yellow, low to high). **g** RNAscope in situ hybridization with *EGFP* (green), *Aplnr* (red), and *Pax8* (blue) mRNA probes on the transverse section from *Mesp1*$^{Cre/+}$;*ROSA26-GFP*$^{f/+}$;*Tbx1*$^{f/+}$ (Ctrl) embryos (upper panel; *n* = 3) and *Mesp1*$^{Cre/+}$;*ROSA26-GFP*$^{f/+}$;*Tbx1*$^{f/f}$ (cKO) embryos (lower panel; *n* = 3) at E9.5. Scale bar, 100 μm. The white dotted line indicates the position in higher magnification images shown in the right. Scale bar, 30 μm. **h** RNAscope in situ hybridization with *EGFP* (green), *Aplnr* (red), and *Pax8* (blue) mRNA probes on the transverse section from *Tbx1*$^{Cre/+}$;*ROSA26-GFP*$^{f/+}$ (Ctrl) embryos (upper panel; *n* = 3) and *Tbx1*$^{Cre/f}$;*ROSA26-GFP*$^{f/+}$ (cKO) embryos (lower panel; *n* = 3) at E9.5. Scale bar, 100 μm. The white dotted line indicates the position in higher magnification images shown in the right (scale bar, 30 μm. a atrium, ao aorta, e ectoderm, nt neural tube, ph pharynx, v ventricle.

regulates the expression of genes in the CPM at the chromatin level, we used two biological replicates of ATAC-seq experiments of *Tbx1*$^{Cre/+}$ (*Tbx1* Ctrl) versus *Tbx1*$^{Cre/f}$ (*Tbx1* cKO) mutant embryos (Supplementary Fig. 1c, d, Fig. 7a, Supplementary Fig. 14a–c, and Table 2). We chose stage E9.5 because this is when *Tbx1* expression is the highest and when the OFT is elongating[58]. The ATAC-seq peaks were separated into commonly accessible regions (CARs) or differentially accessible regions (DARs; FDR < 0.05, Fig. 7b and Supplementary Fig. 14d) between *Tbx1* Ctrl and *Tbx1* cKO mutant samples. Among 5872 DARs, 5859 decreased and 13 increased in chromatin accessibility in *Tbx1* cKO embryos (Supplementary Fig. 14d). To focus on the CPM, we next performed ATAC-seq on *Mesp1*$^{Cre/+}$ cells in two biological replicates and used the peaks to exclude CARs and DARs that were found only in *Tbx1* control and *Tbx1* cKO samples, to eliminate changes in non-mesoderm lineages (Fig. 7c, d, gray regions). The remaining CARs and DARs are thus referred to as CARs-Mesp1 and DARs-Mesp1 (Fig. 7c, d). In DARs-Mesp1, several transcription factors binding motifs related to the heart or BrM differentiation including T-box motifs were enriched (Fig. 7e and Supplementary Data 11). More peaks in DARs-Mesp1 were found in distal intergenic regions than in CARs-Mesp1 (Supplementary Fig. 13e, f), suggesting that *Tbx1* inactivation has a large effect on the regulation of genes through putative enhancer regions.

Next, we annotated DARs-Mesp1 (2185 peaks) to predicted target genes (2652 genes; Fig. 7f and Supplementary Data 12). A total of 160 of the 468 DEGs that were downregulated (down in cKO) in MLPs (Fig. 6b) were associated with DARs-Mesp1 (Fig. 7f, *P* < 1e-16, chi-square test). GO analysis of the 160 genes indicated that they are involved in cell migration and muscle development, important for the function of *Tbx1* (Fig. 7g). On the other hand, those without DARs-Mesp1 that were reduced in expression in mutant embryos were associated with the MAPK pathway and other functions (Fig. 7h). Of interest, *Tbx1* might indirectly affect MAPK signaling in the MLPs that were not detected by ATAC-seq analysis and this is potentially mediated by well-known *Tbx1*-dependent FGF signaling[47,48].

To determine which genes with DARs could be direct target genes of TBX1, we performed ChIP-seq with our new Avi-tagged *Tbx1* mouse line and created double homozygous mice harboring the biotin ligase (BirA) gene (*Tbx1*$^{3'-Avi}$;*BirA*, Supplementary Fig. 15). We used input as background (sequencing depth replicate 1 is 26,318,633; replicate 2 is 29,821,823; replicate 3 is 33,965,309) to compare with peaks found by ChIP (sequencing depth replicate 1 is 28,070,687; replicate 2 is 32,561,521; replicate 3 is 32,907,512). We considered peaks found in at least two of the three replicates, each containing 20 embryos of 19–22 somites, with aligned peaks (replicate 1, 294; replicate 2, 128; replicate 3, 536 peaks) as high confidence TBX1-binding sites

(Supplementary Fig. 16a–c). Out of 255 peaks, 176 peaks had a T-box motif (Fig. 8a, Supplementary Fig. 16d, e, and Supplementary Data 13), supporting TBX1 occupancy; 104 peaks (41%) in the ChIP-seq data overlapped with DARs (Fig. 8a, b, *P* < 0.001, permutation test). Comparing *Tbx1* Ctrl and *Tbx1* cKO ATAC-seq data in 255 ChIP-seq peak regions, we found that these regions were mostly open and accessible in Ctrl embryos but closed in *Tbx1* cKO embryos (Fig. 8c and Supplementary Fig. 16f). Of the 255 ChIP-seq peaks, 151 (59%) did not show significant chromatin accessibility changes (i.e., overlapping with DARs), which include some TBX1-binding sites in closed regions that did not change in accessibility in the mutant data (Fig. 8c), suggesting a diverse role of TBX1 in promoting chromatin remodeling.

ChIP-seq peaks were annotated to predicted direct transcriptional target genes (443 genes, 470 peaks, Supplementary Data 14). We then intersected the DEGs reduced in MLPs, the annotated genes from Mesp1-DARs and TBX1 ChIP-seq targets (Fig. 8b). We found 21 known genes (Fig. 8b and Supplementary Data 14) common in all three datasets (*P* < 0.001; permutation test). Among them, eight had a DAR that overlapped with a TBX1 ChIP-seq peak (*Slit1*-intron, *Crybg3*-intron, *Nrg1*-upstream, *Trps1*-downstream, *Sox9*-upstream, *Trmt9b*-downstream, *Fn1*-upstream, and *Crtc2*-promoter region). Data is consistent with TBX1 binding to accessible chromatin in control embryos, but the interval is not accessible when *Tbx1* is inactivated. The rest had a DAR that did not overlap with a TBX1-binding site (*Fgf1*, *Aplnr*, *Tshz3*, *Rcsd1*, *B3galnt12*, *Ppml1*, *Spon1*, *Mpped2*, *Tbx18*, *Daam1*, *Parvb*); perhaps regulation is by long-range chromatin interaction with TBX1 binding. We show two representative examples, *Aplnr* and *Nrg1*, which are MLP enriched genes that are TBX1 direct transcriptional targets (Fig. 8d). In the *Nrg1* locus, the TBX1-binding region was closed in *Tbx1* cKO embryos. We examined the ENCODE ChIP-seq tracks in the UCSC genome browser tracks[49]. For *Nrg1*, the co-localized peak is within an ENCODE cis-regulatory element (cCRE) that is a poised enhancer in mouse embryonic heart[49]. For the *Aplnr* gene region, the TBX1-binding site that is just downstream of the 3'UTR (Fig. 8d), is in an ENCODE cCRE that is a poised enhancer in mouse embryonic heart, but it was not in a DAR found in our data. The DAR that was identified in the *Aplnr* locus, is in an ENCODE cCRE (E0701748/enhD) and is an ATAC-seq peak region in the embryonic heart[49]. Overall, the regions found appear to be regulatory regions, but TBX1 might not always affect chromatin accessibility, indicating that multiple mechanisms of regulation occur.

Taking the results from the three types of functional genomic data in this report, we can generate a putative gene regulatory network for TBX1 function in the MLPs as summarized in Fig. 8e. Here, we distinguish four categories of genes potentially regulated by TBX1: (1) Direct target genes with or (2) without chromatin changes, and

**a**

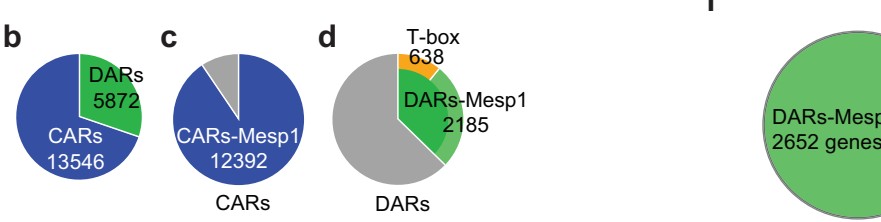

**e** DARs-Mesp1

| Rank | Motif | Transcription factor | p-value | Background | Targets |
|---|---|---|---|---|---|
| 1 | | Pitx1:Ebox(Homeobox,bHLH) | 1e-119 | 0.88% | 8.70% |
| 2 | | Six2(Homeobox) | 1e-80 | 6.91% | 19.32% |
| 3 | | Tbx20(T-box) | 1e-54 | 2.18% | 8.61% |
| 4 | | NFY(CCAAT) | 1e-48 | 8.18% | 18.04% |
| 5 | | Six1(Homeobox) | 1e-47 | 1.66% | 6.96% |
| 6 | | FOXA1(Forkhead) | 1e-35 | 6.71% | 14.38% |
| 15 | | Tcf21(bHLH) | 1e-27 | 12.22% | 20.60% |
| 66 | | Isl1(Homeobox) | 1e-13 | 14.14% | 20.01% |

**b** DARs 5872 / CARs 13546

**c** CARs-Mesp1 12392 / CARs

**d** T-box 638 / DARs-Mesp1 2185 / DARs

**f** DARs-Mesp1 2652 genes — Down in cKO 160 | 308 — DEG in MLPs — 46 | 137 — Up in cKO

**g** DARs-Mesp1-DEGs down in cKO (160 genes)

| GO pathway | Genes |
|---|---|
| ameboidal-type cell migration | *Akap12, Angpt2, Arsb, Bmp7, Efnb1, Efnb2, Ets1, Fgf1, Gbx2, Grem1, Kdr, Mef2c, Pak3, Plk2, Prkd1, Rhoj, Sox9, Stc1* |
| tissue migration | *Angpt2, Arsb, Efnb2, Ets1, Fgf1, Foxf1, Grem1, Kdr, Mef2c, Plk2, Prkd1, Rhoj, Sox9, Stc1* |
| muscle tissue development | *Bmp7, Csrp1, Edn1, Efnb2, Fgf1, Fhl2, Grem1, Hey1, Mef2c, Nrg1, Rarb, Smad3, Sox9, Tbx18, Tenm4, Tshz3, Vamp5* |

-Log (adj. P value)

**h** no DARs-Mesp1-DEGs down in cKO (308 genes)

| GO pathway | Genes |
|---|---|
| regulation of ERK1 and ERK2 cascade | *Ezr, Fgf10, Errfi1, Pdgfa, Timp3, Sirpa, Mfhas1, Flcn, Pdgfc, Prkcz, Dusp1, Fgf8, Dusp3, Apoe, Prkd2, Spry4, Spred3, Tnip* |
| negative regulation of MAPK cascade | *Ezr, Errfi1, Timp3, Sirpa, Flcn, Dusp1, Dusp3, Sh2b3, Apoe, Spry4, Rgs2, Spred3, Tnip1* |
| negative regulation of protein phosphorylation | *Ezr, Errfi1, Rb1, Timp3, Sirpa, Flcn, Prkcz, Hgf, Dusp1, Dusp3, Sh2b3, Apoe, Spry4, Rgs2, Spred3, Irf1, Tnip1, Dgkq, Mvp, Gpd1l* |

-Log (adj. P value)

**Fig. 7 *Tbx1* affects gene expression and chromatin accessibility of MLPs. a** Scheme of ATAC-seq data analysis. **b** Pie chart of ATAC-seq peaks of GFP + cells from *Tbx1^{Cre/+}; ROSA26-GFP^{f/+}* (Ctrl) and *Tbx1^{Cre/f}; ROSA26-GFP^{f/+}* (cKO) embryos at E9.5 (*n* = 3). A total of 13,546 peaks, defined as CARs, commonly accessible regions, were found in both *Tbx1^{Cre}* Ctrl and cKO datasets. A total of 5872 peaks, defined as DARs, differentially accessible regions, in which peak intensity was changed (FDR ≤ 0.05) between *Tbx1^{Cre}* Ctrl and cKO datasets. **c** Pie chart showing the intersection of CARs with ATAC-seq peaks of GFP + cells from *Mesp1^{Cre/+}; ROSA26-GFP^{f/+}* embryos at E9.5 (*n* = 3). Among 13,546 CARs, a total of 12,392 peaks were also found to open in *Mesp1^{Cre}* lineages and thus referred to as CARs-Mesp1. **d** Pie chart showing the intersection of DARs with ATAC-seq peaks of GFP + cells from *Mesp1^{Cre/+};ROSA26-GFP^{f/+}* embryos at E9.5 (*n* = 3). Among 5872 DARs, 2185 were also found to open in *Mesp1^{Cre}* lineages and thus referred as DARs-Mesp1. Of the DARs-Mesp1 regions, 638 had a T-box motif. **e** The motifs found in DARs-Mesp1 regions by Homer (hypergeometric optimization of motif enrichment) with default settings. **f** Venn diagram of genes with ATAC-seq DARs-Mesp1 regions and DEGs in MLPs from scRNA-seq. Of the 770 DEGs in the MLPs, 160 genes showed decreased (*P* value < 2.2e-16) and 46 genes showed increased were found as DARs-Mesp1 (*P* value < 2.2e-16). **g** Enriched biological processes for the 160 genes with both DARs-Mesp1 and reduced expressed in *Tbx1* cKO MLPs. *X* axis indicates adjusted *P* values. **h** Enriched biological processes for the 308 genes with reduced expression in *Tbx1* cKO but without DARs-Mesp1. *X* axis indicates adjusted *P* values.

indirect target genes, (3) with chromatin changes, that contain transcription factor binding sites, and (4) without chromatin changes. Overall, we suggest that TBX1 with *Isl1, Fox, Six, Pitx*, and E-box proteins such as *Tcf21*, (Fig. 8e), act together to regulate the progression of MLPs to more differentiated states in the CPM.

## Discussion

In this report, we discovered a progenitor cell population within the CPM that we term MLP. We discovered that MLPs are localized bilaterally within the posterior nascent mesoderm of the pharyngeal apparatus. The MLPs function to both maintain a

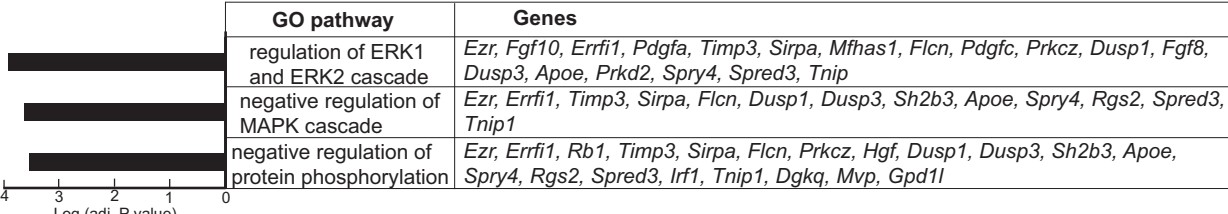

**Table 2 Summary of ATAC-seq samples.**

| | No. of embryos | Somites | Cell number GFP(+) | Aligned peaks |
|---|---|---|---|---|
| Mesp1-Cre; GFPf/+ | | | | |
| E9.5-1 | 3 | 20–21 | 10,000 | 25,485 |
| E9.5-2 | 2 | 19–21 | 10,000 | 30,535 |
| E9.5-3 | 5 | 21–22 | 10,000 | 21,161 |
| Tbx1-Cre;GFPf/+ | | | | |
| E9.5-1 | 5 | 19–22 | 10,000 | 49,490 |
| E9.5-2 | 5 | 19–21 | 10,000 | 43,298 |
| E9.5-3 | 5 | 19–21 | 10,000 | 47,013 |
| Tbx1-Cre/f; GFPf/+ | | | | |
| E9.5-1 | 4 | 19–22 | 10,000 | 29,251 |
| E9.5-2 | 3 | 21–22 | 10,000 | 24,042 |
| E9.5-3 | 2 | 21 | 10,000 | 19,772 |

progenitor state and promote differentiation to derivative cell types as shown in the model in Fig. 8f. *Tbx1* is required for the MLPs to express critical genes for cardiac and BrM development, and to prevent ectopic expression of non-mesodermal genes, needed for maturation of the CPM-derivative cells (Fig. 8f). This is mediated by direct and indirect transcriptional regulation and chromatin accessibility allowing for cell state progression.

Previous work showed that the aSHF, pSHF, and BrM cells comprising the CPM, derive from a relatively small number of *Mesp1* expressing progenitor cells at gastrulation[8]. Based upon the work presented in this report, it is most likely that not all cells have committed to final CPM fates at gastrulation and MLPs provide a source of progenitors as the pharyngeal arches form. Retrospective clonal analysis has shown that there is a direct clonal relationship between progenitor cells that form the muscles of mastication and right ventricle, which are derived from the first pharyngeal arch, with distinct clones that forms both the OFT and facial expression muscles, from the second arch, while separate clones form the neck muscles and venous pole of the caudal pharyngeal arches, 3–6[9,30]. This suggests that different clones contribute to different arches. It is possible that MLPs are comprised of heterogenous progenitor cells that are allotted to individual arches and/or that they are exposed to different extracellular signals during development, conferring pharyngeal arch-specific fates.

The heart tube elongates from E8 to 10.5 by the deployment of progenitor cells to the OFT. It is known that cells deployed to the cardiac poles first arrive at the dorsal pericardial wall (DPW), just behind the heart tube[50]. Mesodermal cells lateral and behind the DPW are thought to be incorporated to the DPW and then to the poles of the heart[51]. The deployment of mesoderm cells to the DPW provides a pushing force as the epithelial transitioned cells move to the poles of the heart[51,52]. In *Tbx1* null mutant embryos, there are fewer cells in the DPW resulting in a shortened cardiac OFT[35,43,53]. We propose that MLPs comprise the dorsal population of mesoderm progenitor cells that are needed to allocate cells to the DPW. This is consistent with the reduction of *Wnt5a* expression in our data, as *Wnt5a* is a key downstream gene of *Tbx1*, required for their deployment[51,52].

An anterior–posterior border is established in the DPW between *Tbx1* and *Tbx5* expressing cells, respectively, that provides cells to the poles of the heart[34]. Consistent with this, we found that there are few *Tbx5* expressing cells in the *Tbx1* lineage. Understanding the molecular mechanisms of how the anterior–posterior border is established is an active area of research[5,34]. We previously found that global inactivation of *Tbx1* results in increased expression of caudal pSHF genes such as *Tbx5* and cardiac muscle genes[20,54]. Data presented in this report and

recently[34], indicate that rather than changes in expression, there are instead cell population changes in *Tbx1* mutant embryos. Therefore, this scRNA-seq study discerns better between expression versus population changes depending on *Tbx1*, which is often a key challenge in interpreting developmental phenotypes. More work needs to be done in the future to better understand how these borders are formed and maintained.

In addition to deploying cells to the DPW and then to the heart, the MLPs express genes required for BrM formation in each arch. The BrMs form segmentally in a rostral to the caudal manner, in which the muscles of mastication form first from the first arch and the other muscles of the face and neck form thereafter from more caudal arches. The BrMs express myogenic regulatory transcription factors, including *Tcf21*, *Msc*, *Myf5*, and later *Myod1*[55]. In addition, transcription factor genes such as *Isl1*[11], *Pitx2*[56], *Tbx1*[19], *Lhx2*[22], and *Ebf* genes (*Ciona*[57]) are expressed in the CPM and are required for BrM formation. A subset of MLPs expressing *Tbx1* will later express BrM genes as they migrate to the core of the pharyngeal arches. These cells progressively express myogenic transcription factor genes as they become restricted to form BrM skeletal muscle cells. Besides the MLPs, *Tbx1* is also expressed in BrM progenitor cells, and therefore, some of the gene expression changes we observed in these cells might be due to *Tbx1* expression in the BrM progenitor cells themselves.

Although we focus on the CPM as it relates to cardiac and skeletal muscle development, it was shown that the CPM also contributes to mesenchyme of connective tissue, including cartilage in the neck[58]. Thus, it is possible that the MLPs could promote connective tissue fates in the craniofacial region that are dependent on *Tbx1*. Further work will be needed to assess their lineage relationships. Given that most patients with 22q11.2DS have craniofacial malformations in the face and neck, it is important to understand the developmental trajectories of *Tbx1*-dependent connective tissue progenitor cells.

A strength of this study is that we used two different *Cre* lines (*Mesp1*[Cre] nor *Tbx1*[Cre]) and analyzed genes altered in both lines. This is because neither *Mesp1*[Cre] nor *Tbx1*[Cre] is sufficient to define the CPM, and further, both label non-relevant cell lineages. Since the cell number isolated from these embryos was relatively small, it is also possible that rare populations would be missed, so that using this strategy reduces the concern of under sampling. Further, there are limitations of using the *Tbx1*[Cre] allele to compare with gene expression changes in *Tbx1*[Cre] conditional null mutant embryos. In *Tbx1*[Cre/+] control embryos, one copy of *Tbx1* is inactivated. However, we did not observe cardiac defects in *Tbx1*[Cre] heterozygous embryos or *Tbx1*[+/−] embryos in which a different region of *Tbx1* was inactivated, as maintained in the SwissWebster background. This is in comparison to *Tbx1* heterozygous mice as maintained in C57Bl/6, in which heterozygous mice have pharyngeal arch artery defects at reduced penetrance[16–18]. This suggests that the mild phenotype observed is not due to localized genetic modifiers in the *Tbx1* locus itself but rather the genetic background.

The effect of genetic background upon the pharyngeal arch artery phenotype of *Tbx1* heterozygous mice has been known for many years. Early work noted that the original 129 strain, in which the *Tbx1*[Cre], *Tbx1*[f/+], and *Tbx1*[+/−] mutations were generated, has a suppressive effect upon the penetrance of such defects (similarly to SwissWebster), while the C57Bl/6 background has an enhancing effect, and research excluded that this modifying effect was due to modifications in the localized genomic region[59]. Further, we have not observed the presence of novel or unusual transcripts in the *Tbx1* locus in different alleles in our scRNA-seq data. Finally, this work uses the null phenotype to draw conclusions about the role of *Tbx1* in development, not the

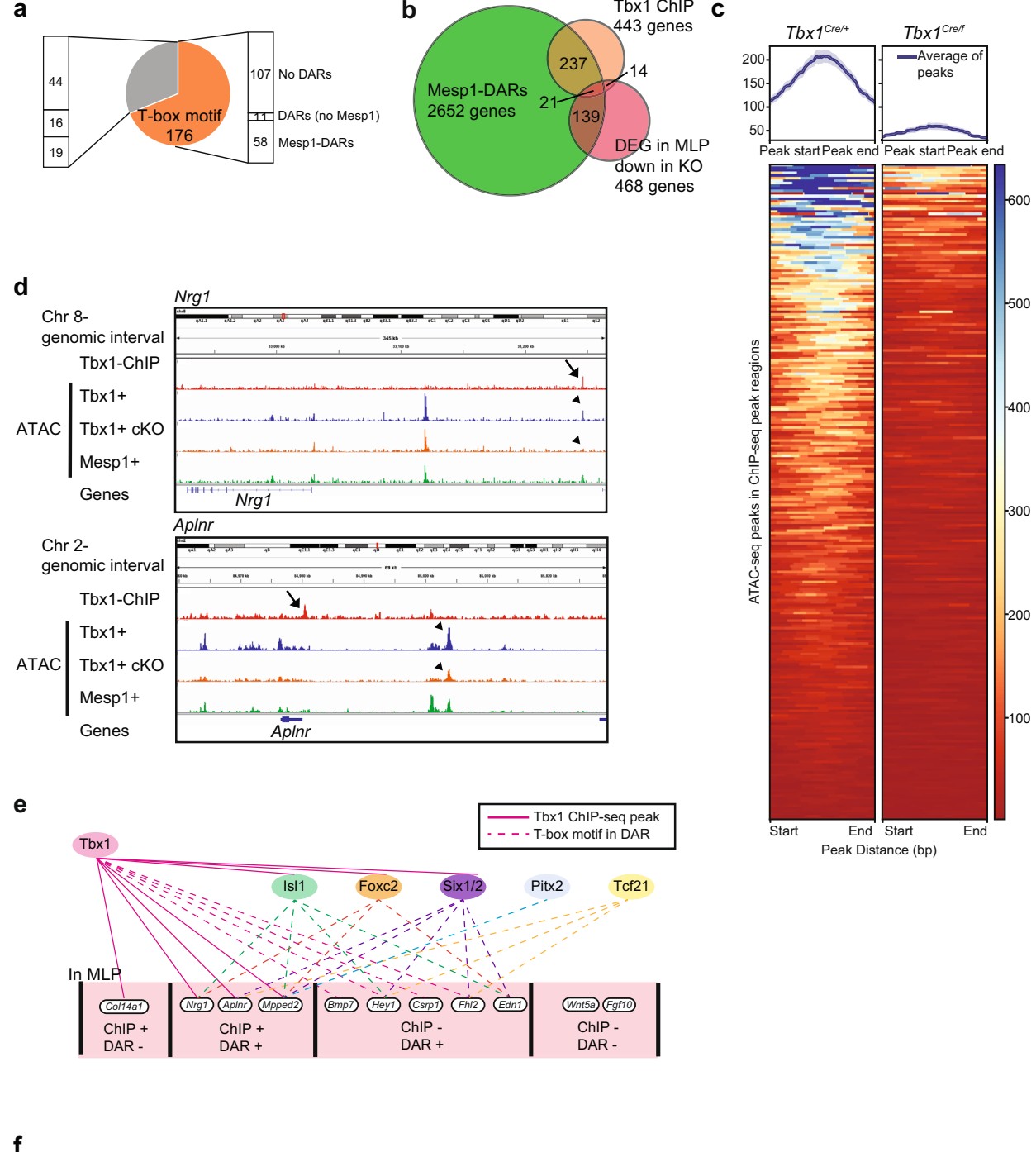

heterozygous phenotype. The null phenotype, as two decades of publications indicate, is remarkably similar across multiple genetic backgrounds. Therefore, the genetic background-dependence of the heterozygous phenotype would not affect the major conclusions of the manuscript.

Although there are likely some gene expression changes in $Tbx1^{Cre/+}$ embryos versus wild-type embryos, we obtained

similar findings when we directly compared scRNA-seq results from $Mesp1^{Cre}$ versus $Tbx1^{Cre}$ experiments. When taken together, using both $Mesp1^{Cre}$ and $Tbx1^{Cre}$ alleles and investigating changes that occurred only in both, allowed for a more complete analysis of the CPM with respect to $Tbx1$. We also performed a scRNA-seq experiment using wild-type versus $Tbx1^{-/-}$ embryos at E9.5 and obtained comparable results.

**Fig. 8 Open chromatin status in TBX1-binding regions in vivo was reduced in *Tbx1* cKO embryos. a** Pie chart of TBX1 ChIP-seq peaks from three biological replicates. Among 255 ChIP-seq peaks, 176 had a T-box motif. The bar chart indicates peak intersection with ATAC-seq data. **b** Venn diagram for overlap of genes with ChIP-seq peaks or ATAC-seq DARs-Mesp1 regions or downregulated expression in *Tbx1* cKO MLPs (*P* value = 0.3924). **c** Average chromatin accessibility at ChIP-seq peaks. Left, ATAC-seq data from *Tbx1^Cre^* Ctrl; right, ATAC-seq data from *Tbx1^Cre^* cKO. Top, average ATAC-seq signal in TBX1 ChIP-seq peak regions; Bottom, heatmap of ATAC-seq read densities in TBX1 ChIP-seq peaks. Statistical significance (adj. *P* < 0.0009) of ChIP-seq peaks on ATAC-seq signal has been tested using enrichPeakOverlap present in ChIPseeker. **d** Genome browser snapshots including results from TBX1 ChIP-seq, ATAC-seq of *Tbx1^Cre^* Ctrl, *Tbx1^Cre^* cKO, and *Mesp^Cre^* lineages, at *Nrg1* (upper), *Aplnr* (lower) loci. Arrow indicates TBX1 ChIP-seq peak. Arrowhead indicates DARs. **e** A cartoon of the TBX1 regulation network in the MLPs and derivatives. Pink lines indicate TBX1 direct target genes from in vivo ChIP-seq. Dotted lines indicate genes with the TBX1 motif harboring DARs-Mesp1 from ATAC-seq analysis. The transcription factor genes indicated in different colors (*Isl1, Foxc2, Six1/2, Pitx2, Tcf21*) are representative family members sharing the same binding motif enriched in DARs-Mesp1. **f** A working hypothesis of MLP function with respect to *Tbx1*. In normal embryogenesis, MLPs provide progenitors that can form cardiac and BrM skeletal muscle. In *Tbx1* conditional null embryos, the cellular state of MLPs is disrupted by the loss of direct and indirect transcriptional regulation (shown as flower shape). This leads to failed formation of CPM derivatives, which causes BrM and cardiac OFT defects.

To understand the molecular mechanisms of *Tbx1* function we performed ATAC-seq and TBX1 ChIP-seq. We generated a gene regulatory pathway downstream of *Tbx1* in MLPs that is required for the maturation of cells to CPM derivatives. We obtained robust data from the ATAC-seq experiments and found alteration of chromatin accessibility when *Tbx1* was inactivated, of which some harbored putative T-box-binding sites. The TBX1 ChIP-seq provided hundreds of direct target genes, of which some are also reduced in expression in mutant embryos and show a change in chromatin accessibility. One note is that we identified hundreds rather than thousands of direct transcriptional targets that were expected based upon studies of other transcription factors. This could be because we used whole embryos for the ChIP-seq with lower tissue yield than microdissection. Nonetheless, the ChIP-seq data supports the ATAC-seq findings. Further, this work shows that both direct and indirect regulation occurs downstream from TBX1 because not all differentially accessible sites have T-sites, and not all differentially expressed genes have either differentially accessible sites or ChIP-seq peaks. Additionally, TBX1 protein can regulate the protein level of serum response factor, SRF without changing the expression level of RNA, thus there are possible other functions of TBX1 that do not involve binding to DNA[22]. We identified MAPK pathway genes are altered indirectly downstream of *Tbx1*, suggesting that this is a result of the alteration of well-known *Tbx1*-dependent FGF signaling[47,48]. A subset of genes found with differentially accessible regions in mutant embryos was expressed in MLPs as well as derivative cell types. This is particularly true for the BrM progenitor cells because this is where *Tbx1* is actively expressed.

By performing these multi-omic studies, we identified *Aplnr* and *Nrg1* among the genes enriched in expression in MLPs. *Aplnr* encodes the APJ/Apelin G-protein coupled receptor that binds Apelin or Elabela/Toddler peptide ligands that have many embryonic and adult functions[60]. In zebrafish, knockdown of *Aplnr* disrupts normal migration of cells during gastrulation including that of cardiac progenitors resulting in severe defects[61,62]. Unexpectedly, their role in early embryogenesis is not recapitulated fully in mouse models, implicating perhaps functional redundancy with other G-protein-coupled receptors or ligands[63]. *Aplnr* is expressed in the CPM, and from stem cell studies, has a role in cardiomyocyte development[27]. *Nrg1* is also of particular interest. In contrast to *Aplnr*, *Nrg1* is not expressed in the DPW. *Nrg1* encodes an EGF family ligand that binds to ErbB receptor tyrosine kinases and has multiple roles in cardiac development and function[28,64]. Interestingly, both *Aplnr* and *Nrg1* are direct target genes of TBX1 based on our ATAC-seq and ChIP-seq results, however, more work will need to be done to know whether these genes have functional importance in MLPs or in relation to *Tbx1*.

Among the genes that were differentially expressed and differentially accessible in *Tbx1* conditional null embryos are

*Isl1*[20,65], *Foxc2*[66,67] *Six1/Six2*[48], *Pitx2*[56,68], and *Tcf21*[22]. *Isl1, Foxc2*, and *Six2* may be direct transcriptional target genes of TBX1 based upon ChIP-seq analysis. Some known downstream genes of *Tbx1* were not identified in the multi-omic data, such as *Wnt5a*[69], *Fgf10*[70,71], and *Nkx2-5*, possibly due to low transcript abundance, incomplete set of TBX1 target genes from ChIP-seq, or non-autonomous functions in neighboring CPM cells.

The distal pharyngeal apparatus is hypoplastic when *Tbx1* is inactivated in the mesoderm[72]. This is in part because the loss of *Tbx1* severely affects pharyngeal endoderm-mediated segmentation[36] affecting neighboring neural crest cell populations[72]. These functions are non-autonomous between the CPM and neural crest cells, given that *Tbx1* is not expressed in neural crest cells that contribute to OFT septation[73]. It is possible that altered signaling from affected pharyngeal endoderm cells or lack of neural crest cells could influence MLP or CPM differentiation, besides cell or tissue autonomous effects.

Global inactivation of *Tbx1* results in a persistent truncus arteriosus and hypoplastic muscles of mastication and failure to form the facial/neck muscles[16,19,74]. Here, we report the existence of a mesodermal cell population termed MLPs. We show that inactivation of *Tbx1* results in dysregulation of gene expression in MLPs affecting their cellular state. Once this occurs, differentiation is affected as well as signaling to adjacent cells thereby altering migration and/or survival of derivative CPM cells leading to the observed phenotypes. The MLPs are thus a continuous but evolving source of CPM cells that is maintained during the development of the pharyngeal apparatus and plays a critical role in craniofacial and cardiac development as well as pathology.

## Methods

**Mice.** All experiments using mice were carried out according to regulatory standards defined by the National Institutes of Health and the Institute for Animal Studies, Albert Einstein College of Medicine (https://www.einsteinmed.org/administration/animal-studies/), IACUC protocol is #0000-1034.

The following mouse mutant alleles used in this study are: *Mesp1^Cre^*[7], *Tbx1^Cre^*[75], *Tbx1^f/f^*[76], *Tbx1^+/-^*[76], *ROSA26-GFP^f/f^* (RCE: loxP)[77]. *Tbx1^Cre/+^*, *Tbx1^f/+^*, *Mesp1^Cre/+^*, and *ROSA26-GFP^f/+^*, single heterozygous mice have been backcrossed in SwissWebster strain for over 15 generations. The *Tbx1^f/f^*, *ROSA26-GFP^f/f^*, and *ROSA26-GFP^f/f^;Tbx1^f/f^* homozygous mice were inter-crossed in brother × sister crosses for over 20 generations and constitute inbred lines. The breeding strategies for all the experiments are illustrated in Supplementary Fig. 1. For lineage tracing in *Tbx1* conditional knockout embryos, *ROSA26-GFP^f/f^;Tbx1^f/f^* and *ROSA26-GFP^f/f^;Tbx1^f/+^* mice were generated by inter-crossing *Tbx1^f/f^* and *ROSA26-GFP^f/f^* mice. To generate *Mesp1^Cre/+^;ROSA26-GFP^f/+^* embryos, *Mesp1^Cre/+^* male mice were crossed with *ROSA26-GFP^f/f^* female mice (Supplementary Fig. 1a). To generate *Mesp1^Cre/+^;ROSA26-GFP^f/+^;Tbx1^f/f^* mutant embryos, *Mesp1^Cre/+^* knock-in mice were crossed with *Tbx1^f/f^* mice to obtain *Mesp1^Cre/+^;Tbx1^f/+^* male mice, then crossed with *ROSA26-GFP^f/f^;Tbx1^f/f^* female mice (Supplementary Fig. 1b). The littermates with *Mesp1^Cre/+^; ROSA26-GFP^f/+^;Tbx1^f/+^* were used as the Ctrl in RNAscope experiments. To generate *Tbx1^Cre/+^;ROSA26-GFP^f/+^* and *Tbx1^Cre/f^;ROSA26-GFP^f/+^* embryos for scRNA-seq and ATAC-seq, *ROSA26-GFP^f/f^* female mice and *ROSA26-GFP^f/f^;Tbx1^f/f^* female mice were crossed with *Tbx1^Cre/+^* male mice, respectively (Supplementary Fig. 1c, d). For RNAscope analysis, littermates were used from a cross of *Tbx1^Cre/+^* male mice and *ROSA26-GFP^f/f^;Tbx1^f/+^*; female

mice (Supplementary Fig. 1e). All the mice are maintained on the SwissWebster genetic background. The PCR strategies for mouse genotyping have been described in the original papers and are available upon request.

To generate *Tbx1³'-Avi* mice, Avidin (Avi) was inserted using CRISPR/Cas9 genomic engineering in the Gene Modification Facility of Albert Einstein College of Medicine. A guide RNA (gRNA) targeting to the 3' last exon of *Tbx1* (5'-gcgcgcgggcgcactatctgggg-3') was designed by Guide Design Resources (http://crispr.mit.edu/) and generated by in vitro transcription[77]. Cas9 mRNA was purchased from SBI. *Tbx1³'-Avi* homology directed repair (HDR_ vector containing the 60 nt homology arms (5'-ggcggccgcgccgcccggtgcctacgactactgcccagaGGTGG AAGTggcctgacgacatcttcgaggctcagaaaatcgaatggcacgaatagtgcgcccgcgcgccgaccccgagg gccatccaaggacgcgctccc-3') at each side surrounding the Avi tag was synthesized chemically from IDT. Super-ovulated female C57BL6 mice (3–4 weeks old) were crossed with C57BL/6 males, and fertilized embryos were collected from oviducts. The gRNA, Cas9 mRNA and *Tbx1³'-Avi* HRD vectors were mixed and microinjected into the cytoplasm of fertilized eggs. The injected zygotes were transferred into pseudopregnant CD1 females, and the resulting pups were obtained. For genotyping, 311 bp (wild-type) and 365 bp (*Tbx1³'-Avi*) bands were observed with *Tbx1³'-Avi* F4: 5'-gcagccaacgtgtactcgtc-3' and *Tbx1³'-Avi* R2: 5'-gccggtgcagtatctacagt-3' primer pair. After several backcrosses with Swiss Webster mice, *Tbx1³'-Avi* mice were crossed with *FVB;129P2-Gt(ROSA)26Sor^tm1.1(BirA)Mejr/J* (*BirA*) mice from the Jackson laboratory to obtain *Tbx1³'-Avi;BirA* double homozygous mice.

The embryos at E8.0–E10.5 determined by standard somite counts, were used for all the experiments. We did not genotype for sex and we used both male and female embryos for all the experiments.

**scRNA-seq.** The embryos at E8.0–10.5 were isolated and GFP-positive embryos were selected under a SteREO Discovery.V12 microscope (Carl Zeiss, Jena, Germany) in ice-cold DPBS with Ca²⁺ and Mg²⁺ (GIBCO, Cat# 14040-133). The rostral half of the embryos were collected at E8.0, E8.25, and E8.5. The pharyngeal apparatus with heart was collected at E9.5 and E10.5, as shown in Fig. 1b. The microdissected tissues were kept on ice and pooled in DMEM (4 °C, GIBCO, Cat# 11885-084) until all the dissections were completed. Following centrifugation (4 °C, 100 × g, 5'; ' is minutes) and removal of DMEM, tissues were incubated with 0.25% Trypsin-EDTA (GIBCO, Cat# 25200-056) with Dnase I (50 U/ml) (Millipore, Cat# 260913-10MU), 10' at RT. Then FBS (heat-inactivated, ATCC, Cat# 30-2021) was added to stop the reaction. After centrifugation (4 °C, 300 × g, 5'), the cells were resuspended in PBS w/o Ca²⁺ and Mg²⁺ (Corning, Cat# 21-031-cv) with 10% FBS and passed through the 100-μm cell strainer. DAPI (Thermo Fisher Scientific, Cat# D3571) was added before cell sorting. The GFP + DAPI− cells were sorted with the BD FACSAria II system (Becton, Dickinson Biosciences, Franklin Lakes, NJ) with BD FACSDiva 8.0.1 software (Becton, Dickinson Biosciences). The sorted GFP-positive cells were centrifuged (4 °C, 300 × g, 5'), and resuspended in 50 μl PBS w/o Ca²⁺ and Mg²⁺ with 10% FBS. After measuring cell number and cell viability, the cells were loaded in a 10x Chromium instrument (10x Genomics, Pleasanton, CA) using Chromium Single Cell 3' Library & Gel Bead Kit v2 (10x Genomics, Cat# PN-120237) or Chromium Next GEM Single Cell 3' GEM, Library & Gel Bead Kit v3.1 (10x Genomics, Cat# PN-1000121) according to the manufacturer's instructions (Genomics Core Facility). The concentrations of the libraries were measured with Qubit®2.0 Fluorometer (Thermo Fisher Scientific, Waltham, MA) using Qubit dsDNA HS Assay Kit (Thermo Fisher Scientific, Cat# Q32851). The details of each sample are provided in Table 1.

**ATAC-seq.** The ATAC-seq method has been previously described[78]; however, we provide details in this section. The embryos at E9.5 were isolated from a euthanized mother. GFP-positive embryos were selected under a SteREO Discovery.V12 microscope in ice-cold DPBS with Ca²⁺ and Mg²⁺ (GIBCO, Cat# 14040-133). The pharyngeal apparatus without the heart was collected and pooled in DMEM (GIBCO, Cat# 11885-084) until all the dissections were completed. Following centrifugation (4 °C, 100 × g, 5') and removal of DMEM, tissues were incubated with 0.25% Trypsin-EDTA (GIBCO, Cat# 25200-056) with Dnase I (50 U/ml) (Millipore, Cat# 260913-10MU), 10' at RT. Then FBS (ATCC, Cat# 30-2021) was added to stop the reaction. After centrifugation (4 °C, 300 × g, 5'), the cells were resuspended in PBS w/o Ca²⁺ and Mg²⁺ (Corning, Cat# 21-031-cv) with 3% FBS and passed through the 100-μm cell strainer. DAPI (Thermo Fisher Scientific, Cat# D3571) was added before the cell sorting. The GFP + DAPI- cells were sorted with the BD FACSAria II system (Becton, Dickinson and Company, Franklin Lakes, NJ). The sorted cells were centrifuged (4 °C, 800 × g, 5'), and washed with PBS w/o Ca²⁺ and Mg²⁺. After centrifugation (4 °C, 800 × g, 5'), cells were resuspended with ice-cold Lysis buffer which contained 10 mM Tris-HCl pH 7.4 (Thermo Fisher Scientific, Cat# BP152-1), 10 mM NaCl (Thermo Fisher Scientific, Cat# AM9760G), 3 mM MgCl₂ (MilliporeSigma, Cat# M9272) and 0.1% Igepal CA-630 (MilliporeSigma, Cat# I8896). Following centrifugation (4 °C, 1000 × g, 5'), nuclear pellets were incubated with Tn5 transpose from the Nextera DNA Sample Preparation Kit (Illumina, Cat# FC-121-1030) for 37 °C, 30'. Transposed DNA was purified using the MinElute PCR purification kit (QIAGEN, Cat# 28004), according to the manufacturer's instructions. The transposed DNA was amplified with NEB Net High Fidelity 2× PCR master mix, PCR primer cocktail, and two index primers from the Nextera Index Kit (Illumina, Cat# FC-121-1011). PCR

conditions for amplification is one cycle 5' 72 °C, 30" (seconds) 98 °C, 12 cycle 10", 98 °C, 30", 63 °C, 1', 72 °C. PCR products were purified using the MinElute PCR purification kit. DNA concentration was measured with a Qubit®2.0 Fluorometer using Qubit dsDNA HS Assay Kit. DNA qualities were analyzed with a Bioanalyzer (Agilent Technologies, Santa Clara, CA).

**ChIP-seq.** Whole embryos from *Tbx1³'-Avi;BirA* double homozygous embryos or *BirA* homozygous embryos (Ctrl) at E9.5 were collected and minced in ice-cold PBS. After centrifugation (4 °C, 200 × g, 5'), tissues were cross-linked with 1% formaldehyde (Thermo Fisher Scientific, Cat# 28906), 30' at RT. A total of 2.5 M glycine (MilliporeSigma, Cat# G8898) was added at a final concentration 0.125 M to stop the reaction. After the tissues were washed with PBS, and centrifuged (4 °C, 200×g, 5'), the pellets were frozen in dry ice and stored at −80 °C. We used 20 embryos for one sample and performed three biological replicates. The frozen tissues were homogenized by BeadBug Microtube Homogenizer (Benchmark Scientific, Edison, NJ) in Lysis buffer, which contained 50 mM Hepes pH 7.5 (MilliporeSigma, Cat# H0887), 140 mM NaCl, 1 mM EDTA (MilliporeSigma, Cat# E7889), 10% Glycerol (MilliporeSigma, Cat# 4750-OP), 0.5% IGEPAL® CA-630 (MilliporeSigma, Cat# I8896), 0.25% TritonX-100 (MilliporeSigma, Cat# X100) and Protease Inhibitor cocktail (MilliporeSigma, Cat# P8340). Samples were incubated on ice for 10'. After centrifugation (4 °C, 2000 × g, 5'), extracted nuclei were washed in Wash buffer containing 10 mM Tris-HCl pH 8.0 (MilliporeSigma, Cat# T2694), 200 mM NaCl, 1 mM EDTA pH 8.0, 0.5 mM EGTA pH 8.0 (MilliporeSigma, Cat# E3889) and Protease Inhibitor Cocktail. Then the nuclei were resuspended in Shearing buffer containing 10 mM Tris-HCl pH 8.0, 1 mM EDTA pH 8.0, 0.1% SDS (BIO-RAD, Cat# 1610418), and Protease Inhibitor Cocktail. Resuspended nuclei were sonicated with a S2 Focused-ultrasonicator (Covaris, Inc., Woburn, MA). After sonication, 10% TritonX-100 and 5 M NaCl were added to sheared chromatin in a final concentration of 1% TritonX-100 and 150 mM NaCl. Dynabeads™ MyOne™ Streptavidin T1 (Thermo Fisher Scientific, Cat# 65602) were blocked with SEA BLOCK Blocking Buffer (Thermo Fisher Scientific, Cat# 37527), and sheared chromatin was blocked with Dynabeads™ Protein A for immunoprecipitation (Thermo Fisher Scientific, Cat# 10002D) and Dynabeads™ Protein G for immunoprecipitation (Thermo Fisher Scientific, Cat# 10004D) at 4 °C, 1 h on a rotator. Following washing with Shearing buffer, blocked streptavidin magnetic beads were added to the sheared chromatin, then incubated 4 °C overnight on a rotator. Streptavidin magnetic beads were washed with low-salt wash buffer (20 mM Tris-HCl pH 8.0, 45 mM NaCl, 0.1% SDS, 1% TritonX-100, 2 mM EDTA), high-salt wash buffer (20 mM Tris-HCl pH 8.0, 150 mM NaCl, 0.1% SDS, 1% TritonX-100, 2 mM EDTA), LiCl wash buffer containing 10 mM Tris-HCl pH 8.0, 0.25 M LiCl (MilliporeSigma, Cat# L7026), 1% IGEPAL® CA-630,1% sodium deoxycholate (MilliporeSigma, Cat# D6750), 1 mM EDTA. After a wash with TE buffer, DNA was eluted in Elution buffer (10 mM Tris-HCl pH 8.0, 1 mM EDTA, 1% SDS) with Proteinase K (Promega, Cat# V3021) at 65 °C, 5 h. Eluted DNA was purified with the MinElute PCR purification kit. After measuring the DNA concentration with the Qubit®2.0 Fluorometer using Qubit dsDNA HS Assay Kit, the libraries were prepared using Accel-NGS 2S Plus DNA Library Kit (Swift Bioscience, Cat# 21024) and 2S Set A Indexing Kit (Swift Bioscience, Cat# 26148); then the DNA concentration was measured with the Qubit®2.0 Fluorometer using Qubit dsDNA HS assay kit.

**Sequencing.** The DNA libraries were sequenced using an Illumina HiSeq2500 system (at Einstein Epigenomics Core Facility), Illumina HiSeq4000 system (at Genewiz, South Plainfield, NJ) or NovaSeq6000 system (at Novogene, Sacramento, CA), with paired-end, 100 bp read length.

**RNAscope.** *RNAscope on tissue sections:* After incubation of 4% PFA (Alfa Aesar, Cat# 43368) at 4 °C for 4 h, embryos were dehydrated with 70%, 90%, and 100% Ethanol (Thermo Fisher Scientific, Cat# A405P-4). Incubation with xylene (Thermo Fisher Scientific, Cat# X3S-4) was done at RT, an hour twice followed by incubation of paraffin (Thermo Fisher Scientific, Cat# T555) at 65 °C, an hour twice. Embryos were embedded and stored at 4 °C.

Embryos were sectioned at 7-μm thickness and dried at RT. The sections were processed with RNAscope® Multiplex Fluorescent v2 reagents (Advanced Cell Diagnostics, Cat# 323100), according to the manufacturer's instructions. Briefly, sections were deparaffinized in xylene and dehydrated in graded ethanol, then incubated with hydrogen peroxide at RT, in 10'. The sections were incubated in boiled 1× Target retrieval reagent, for 15'. After rinsing with distilled water and 100% ethanol, the sections were dried at RT, for 5'. The sections were incubated with Protease Plus at RT, for 3' then incubated with mixed RNAscope probes of C1, C2, and C3 channels (Supplementary Table 2), at 40 °C, overnight. Following AMP1, AMP2, and AMP3 treatment, the sections were incubated with Opal 570 (AKOYA BIOSCIENCES, Cat# FP1488001KT) for C1, Opal 620 (AKOYA BIOSCIENCES, Cat# FP1495001KT) for C2, and Opal 520 (AKOYA BIOSCIENCES, Cat# FP1487001KT) for C3 in 1:1000 dilution at RT, for 15'. Then slides were mounted with VECTASHIELD® HardSet™ Antifade Mounting Medium with DAPI (Maravai LifeSciences, Cat# H1500). Images were then captured using a Zeiss Axio Observer microscope with an ApoTome (Carl Zeiss Corp.).

*RNAscope on whole embryos:* Embryos were fixed in 4% PFA at 4 °C overnight, and dehydrated in 25%, 50%, 75%, and 100% Methanol (Thermo Fisher scientific, A412-4). Embryos were stored at −20 °C until needed. Embryos were rehydrated in

75%, 50%, 25% methanol, and PBS + 0.1% Tween (Sigma, Cat# P7949). The embryos were permeabilized using Protease III (Advanced Cell Diagnostics, Cat# 323100) for 20' followed by a wash with PBS + 0.01% Tween. The C1, C2, and C3 probes chosen for the experiments were warmed at 40 °C for 10 min. Embryos were incubated with 100 μL of mixed RNAscope probes of C1, C2, and C3 (50:1:1, respectively) channels (Supplementary Table 2), at 40 °C, overnight. Three washes using 0.2×SSC + 0.01%Tween at RT were used in between all future steps. After washing, embryos were fixed for 10' in 4% PFA at RT. Embryos were incubated in AMP1 for 30', AMP2 for 30 min, and AMP3 for 15 min at 40 °C with washes in between. Tyramide Signal Amplification (TSA) was prepared at 1:2000 for TSA-CY3 (Akoya Biosciences, Cat# NEL744001KT), 1:1000 for TSA-CY5 (Akoya Biosciences, Cat# NEL745001KT), and 1:500 for TSA-Fluorescein (Akoya Biosciences, Cat# NEL741001KT). Embryos were incubated in HRP-C1 for 15' at 40 °C followed by 30' in their chosen TSA for C1. Amplification was blocked in HRP-Blocker for 15'. The previous two steps were then repeated for HRP-C2 for C2 probes and HRP-C3 for C3 probes. After the final washes, embryos were incubated in DAPI (Advanced Cell Diagnostics, Cat# 323100) for 2 days then washed and stored at 4 °C until ready for imaging. Imaging was conducted with a confocal Leica SP5.

**scRNA-seq data analysis.** We utilized Cell Ranger (v 3.1.0, from 10x Genomics) to align reads of scRNA-seq data to the mouse reference genome (assembly, mm10). All the samples passed quality control measures for Cell Ranger (Table 1), and the filtered gene-barcode matrices were used for the following analyses. For *Mesp1Cre* four time point dataset analyses, Scran v1.10.2 was used to normalize the individual datasets by using the computeSumFactors function with the deconvolution method for scaling normalization[79]. The cells were clustered by densityClust v0.3 with the density peak clustering algorithm[80]. We found batch effects existed in the scRNA-seq datasets from different time points (*Mesp1Cre* data) or experimental perturbations (*Mesp1Cre* or *Tbx1Cre* Ctrl versus cKO). Therefore, we performed batch corrections before we comprehensively analyzed gene expression values across these scRNA-seq datasets. We employed the MNN (mutual nearest neighbors) method to identify shared cell types across datasets, and corrected batches, according to the shared cell types, by the MNN batch correction method[81]. More specifically, we removed batch effects by fastMNN function of the Scran package, with the input of the normalized counts from individual datasets. The *Mesp1Cre* Ctrl data from different stages are comprised of homogeneous cell types (transcriptomes of each cell type are concordant across datasets), but the *Mesp1Cre* Ctrl with cKO and then separately, the *Tbx1Cre* Ctrl versus cKO, are comprised of heterogeneous cell types (the same cell types from different conditions, Ctrl vs cKO, with dissimilar gene expression). Thus, we aligned gene expression values for Ctrl vs cKO data using the RPCI (reference principal component integration) method in RISC, which utilizes the global gene reference to calibrate the gene expression changes of heterogeneous cell types[37]. In detail, we combined individual datasets by the scMultiIntegrate function of the RISC package and outputted the corrected gene expression values after the clustering by Seurat 3.1.5[82]. Then, the corrected datasets were processed on Scanpy v1.4.3[83] for cell trajectory analysis, with the PAGA approach[26]. The RISC software[37] was also used to identify differentially expressed genes of *Tbx1* Ctrl vs cKO embryos in each cluster by a Negative Binomial generalized linear model[84] (scDEG function of RISC package), with adjustive P values < 0.05 and logFC (log fold change) >0.25 or < −0.25. The clusterProfiler v3.10.1[85] was used for Gene Ontology pathway analysis, with adjustive P values < 0.05.

**ATAC-seq data analysis.** ATAC-seq analysis pipeline has been described previously[86]; however, we describe the methods here as well. We removed Nextera Transposase Sequence primers[87] in the range 33–47 bp using cutadapt[88] with the following option -a CTGTCTCTTATACACATCTCCGAGCCCACGAGAC -A CTGTCTCTTATACACATCTGACGCTGCCGACGA. Sequences were then aligned to the mouse genome (mm9) using Bowtie2 2.3.4.3[89] with default parameters. Only uniquely mappable reads were retained. We removed reads with mates mapping to different chromosomes, or with discordant pairs orientation, or with a mate-pair distance >2 kb, or PCR duplicates (defined as when both mates are aligned to the same genomic coordinates). Reads mapping to the mitochondrial genome were also removed. ATAC peaks in each sample were identified using MACS2 2.1.2.1[90] with the option -- nomodel --shif100 --extsize 200. The differentially enriched regions (DARs) of *Tbx1Cre* Ctrl vs cKO were obtained using DiffBind 2.14.0[91] by loading all the MACS2 peaks, default parameters were used except for dba.count method where summits have been set to 250. After that, we formed a consensus list of enriched regions for each condition using the intersectBed function from the BEDTools 2.29[92] with the default minimum overlap and retaining only the peak regions common to all three replicates. We defined common peaks the regions that were common to the consensus peaks of both *Tbx1Cre* Ctrl and cKO (using again the intersectBed function). The common peaks in *Tbx1Cre* Ctrl vs cKO data that did not intersect the DARs were defined as common accessible regions (CARs). Both DARs and CARs were filtered out by removing peaks intersecting blacklist regions (Encode mm9 blackregions Version 2) using findOverlapsOfPeaks of ChIPpeakAnno v3.22.4[93] before any further analysis. Then DARs or CARs were filtered with the consensus *Mesp1Cre* ATAC-seq peaks using intersectBed. Transcription factor binding motifs were obtained using the findMotifsGenome program of the HOMER suite (v4.10.3)[94]. For peak

annotation as *cis*-regulatory regions, GREAT[95] was used with default settings with mm9. The comparison of the gene list from DARs and DEGs was performed using standard R-scripts. The clusterProfiler v3.10.1 was used for Gene Ontology pathway analysis. Coverage heatmaps and average enrichment profiles (TSS +/− 10 Kb) in each experimental condition were obtained using ngs.plot[96] or deepTools[97]. We applied statistical analysis in the processes of all the software and packages with default settings.

**ChIP-seq data analysis.** We removed adapters using cutadapt[88] with -a option and a set of adapters detected with FASTQC[98]. Sequences were then trimmed using TrimGalore with option –length 0. Then sequences were aligned to the mouse genome (mm9) using Bowtie2 2.3.4. with default parameters. Reads were removed with mates mapping to different chromosomes, or with discordant pairs orientation, or with a mate-pair distance >2 kb, or PCR duplicates (defined as when both mates are aligned to the same genomic coordinate). Only uniquely mappable reads were retained. ChIP-seq peaks in each sample were identified using MACS2 2.1.2.1 with default parameters. Then, a consensus list of enriched regions was obtained using the intersectBed function from the BEDTools 2.29 with the default minimum overlap and retaining only the peak regions common to at least two out of the three replicates. Peaks were filtered by removing those overlapping with blacklist regions (Encode mm9 black regions Version 2) using findOverlapsOfPeaks of ChIPpeakAnno. The transcription factor binding motifs were obtained using the findMotifsGenome program with -size given parameter of the HOMER suite. For peak annotation as *cis*-regulatory regions, GREAT was used with the default settings mm9. The comparison of the gene lists of DAR, DEG, and ChIP regions was performed using the dplyr package in R. IGV 2.4.8[97] was used for peak visualization. Coverage heatmaps and average enrichment profiles (TSS + /- 10Kb) in each experimental condition were obtained using ngs.plot or deepTools2. The significance of the overlap between the two lists of peaks was evaluated using the ChIPseeker enrichPeakOverlap using mm9 annotation. We applied statistical analysis in the processes of all the software and packages with default settings.

**Quantification and statistical analysis.** Besides the methods for scRNA-seq analysis, specific statistical tests were described in "Results" and Figure legends. Briefly, to determine the relative proportions of the CPM populations, a two proportion Z test was performed in each cluster with 95% confidence interval. For ATAC-seq and TBX1 ChIP-seq, default statistical analysis was used for DiffBind, HOMER, and GREAT. The heatmap of ATAC-seq read densities in TBX1 ChIP-seq peaks on ATAC-seq signals were determined using enrichPeakOverlap present in ChIPseeker. For Venn diagrams, Pearson's Chi-squared test was performed.

**Reporting summary.** Further information on research design is available in the Nature Research Reporting Summary linked to this article.

## Data availability
The datasets generated during this study are available at the GEO repository under the accession numbers: scRNA-seq "GSE167493", scRNA-seq "GSE167491", ATAC-seq "GSE173700", and ChIP-seq "GSE173521". The scRNA-seq data can be viewed at https://scviewer.shinyapps.io/heartMLP. All other relevant data supporting the key findings of this study are available within the article and its Supplementary Information files or from the corresponding author upon reasonable request.

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

## Acknowledgements

We thank Genomics core, especially David Reynolds, Director of the Genomics Core, Shahina Maqbool, Director of the Epigenomics core as well as the Flow Cytometry and Gene modification facilities at Einstein. We appreciate all the help in data analysis and training by Masako Suzuki at Einstein. We thank Drs. Bin Zhou, Cedric Blanpain, and Peter Scambler for reading this manuscript and providing helpful suggestions. We also thank NYU Center for Genomics and Systems Biology Genomics Core; NYU Langone's Genome Technology Center is partially supported by the Cancer Center Support Grant P30CA016087 at the Laura and Isaac Perlmutter Cancer Center, especially Peter Meyn. This work was supported by grants from the National Institutes of Health P01HD070454 (B.E.M. and D.Z.), R01HL153920 (B.E.M. and D.Z.), R01HD096770 (L.C.), and R01HL108643 (L.C.). The work was also supported by a grant from the Foundation Leducq (Transatlantic Network of Excellence 15CVD01; B.E.M., L.C., R.G.K., and A.B.) and the Agence Nationale de la Recherche Heartbox and Myohead projects (RGK). Dr. Nomaru was supported by an American Association Grant (19POST34380281).

## Author contributions

H.N., L.C., R.G.K., A.B., D.Z., and B.E.M. designed the studies. H.N., Y.L., C.D.B., D.R., A.C., W.W., H.S., S.E.R., A.G.D., C.A., R.G.K., A.B., D.Z., and B.E.M. performed and analyzed the experiments. H.N. and B.E.M. wrote the paper. Y.L. and D.Z. provided expert bioinformatics knowledge and guidance with the intellectual contribution of biological interpretation from C.D.B. and R.G.K. C.A. provided expert knowledge for ATAC-seq and ChIP-seq with the consultation of D.Z. L.Z. and C.-L.C. provided the conceptional knowledge and helped generate *Tbx1-Avi* mice used for ChIP-seq. H.N. with H.S. performed RNAscope experiments and analysis with intellectual contribution by C.D.B. All authors reviewed the results, helped edit the manuscript, and approved the paper.

## Competing interests

The authors declare no competing interests.

## Additional information

**Peer review information** *Nature Communications* thanks Simon Bamforth and the other anonymous reviewer(s) for their contribution to the peer review this work. Peer reviewer reports are available.

