## [Peer Review File · Nature Communications]

Reviewers' Comments:

Reviewer #1:

Remarks to the Author:

The authors perform single cell RNA-seq of *Mesp1*-cre traced cells from the rostral part of the embryo proper to identify CPM cells and their derivatives. From this study, the authors make a number of interesting observations, including that a population of undifferentiated, multipotent CPM is maintained over the time period of study along the lateral aspect of the pharyngeal apparatus, from where it contributes differentiating cells into the heart and BrM. They further show, by evaluating the same cells in knockout embryos, that *Tbx1*, a gene whose reduction in copy number gives rise to congenital heart defects in humans, is an important regulatory factor, promoting differentiation of CPM. They go on to perform ATAC-seq in *Tbx1* mutants and ChIP-seq for *Tbx1* at E9.5 to identify genes that are directly regulated by *Tbx1* within the CPM. This provides important mechanistic insight into the function of *Tbx1* within CPM and the teratogenic effects of the 22q11.2 deletion in humans. The study is comprehensive, well written, and fairly interpreted, and will be of interest to a wide audience.

I have only one major concern:

1. The authors mention that on the Swiss Webster background, *Tbx1* heterozygotes do not manifest haploinsufficiency for cardiac development, and that (presumably for this reason) all mice are kept on the Swiss Webster background. However, the methods do not explain how this was done, as the genetic tools were not all generated on the Swiss Webster background. Were they independently backcrossed, and if so, for how many generations? Was whole-genome SNP analysis performed to ascertain the degree of homozygosity on the Swiss Webster background? One significant concern when backcrossing from the strain in which the genetic modification was done onto another is that the region proximal to the modification will not be converted to the new strain. If modifiers happen to exist in this region, their effects will be unknowingly assigned to the modification. I.e., is an effect, such as a gene expression difference or loss of a set of ATAC-seq peaks, due to the presence of the cre, or to a linked modifier brought over from the previous strain? This problem becomes smaller with greater backcrossing, and more defined with determination of congenic breakpoints. This is a particular concern because evidently there are modifiers of the *Tbx1* effect on CPM development that distinguish Swiss Webster from the strains in which the genetic tools were made (probably 129). The authors should provide as much detail as possible regarding the backcrossing and check the size of congenic domains by whole genome SNP analysis in the methods section, perhaps mention the backcross N in the results section, and briefly address the concern that modifiers may be confounding their interpretations.

Additional comments:

2. A number of genes fall out of the pseudotime analysis as changing within the undifferentiated population of CPM over time. Can the authors confirm these predicted changes using RNA-scope for specific examples, for example by showing E10.5 RNA-scope data for *Apln4* or *Sema3c* side by side with E8.5?

3. In comparing the *Mesp1*-cre with the *Tbx1*-cre, the authors state that the *Mesp1* lineage contributes more broadly to embryonic mesoderm, while *Tbx1* contributes to pharyngeal endoderm and ectoderm. They provide two references for the extra-CPM activity of *Tbx1*, but none for that of *Mesp1*.

Reviewer #2:

Remarks to the Author:

General comments

This paper investigates the role of a multilineage-primed population (MLP) within the cardiopharyngeal mesoderm (CPM) that contributes to cardiac and skeletal cell differentiation. It specifically focusses on *Tbx1*, the gene central to complications in 22q11 deletion syndrome and identifies a mechanism, through single cell-based technologies, as to how *TBX1* directs the development of the pharyngeal apparatus by marking the MLP population.

Major concerns

The use of two different Cre lines to conditionally delete Tbx1 from the developing embryo, particularly for the RNA-seq experiments, is very confusing. Firstly, only one Mesp1;Tbx1 cKO embryo was used for this experiment, and it also appears that only one pool of Tbx1Cre;Tbx1 cKO embryos was used. Not only does this mean that only one biological replicate was used for each Cre line for the whole RNA-seq study, these two distinct Cre lines were also pooled together as a general cKO line. This does not seem appropriate to me as each Cre works in different tissues where Tbx1 is active and there is no ability to verify the accuracy of the sequencing with data from a separate biological replicate from each Cre line.

The authors state: "In Tbx1 cKO embryos at E9.5, the BrM and CM/OFT populations were significantly smaller than in controls". This is not surprising as the caudal pharyngeal arches do not form in the Tbx1 cKO embryos, so there is relatively less tissue present. Have the authors compensated for this change in anatomy with the single cell analysis? Has this, for example, been applied to the data in Figures 4 and 5?

There are also issues with the low number of cells collected. The data in Table 1 shows that a very low number of cells were captured overall, and particularly for the Tbx1Cre genotypes at E8.5. Sequencing a low number of cells is likely to result in the loss of rare cell populations in the analysis. Can the authors please comment on this?

The use of different time points (e.g. E8.0/E8.25 in some figures but E8.5 in others) needs to be justified. Also, the use of control lines needs more clarification - there are some experiments where Tbx1 hets and Tbx1 WTs are grouped together for a control group, but others where the Tbx1 hets and cKOs are compared. All experiments should be done with appropriate and consistent controls (e.g. line-specific controls) at appropriate and consistent time points.

The overall structure of the paper, which includes many multi-panel figures (in the main paper and supplemental) and several supplemental tables, is confusing to read. For example, there are several parts in the text where the reader is referred to four different panels or figures within the span of a couple of sentences, with some genes being demonstrated in the main figure and others being in the supplementary figures. Moreover, the panels are not necessarily referred to in the correct order (e.g. A, B, C), and there are a couple of occasions where a results section will discuss, for example, 3/4 of a figure, then the next section will discuss the remaining panels on the previous figure plus some other panels in the next figure, plus several supplementary figures/panels.

The Supplementary Tables could not confidently be identified in the Suppl files due to the uninformative labelling of each Excel file.

The manuscript as whole therefore needs to be restructured to make the data more accessible to the reader.

Specific comments

Methods

It is not at all clear why the Tbx1 cKO embryos analysed are a mixture of Mesp1Cre and Tbx1Cre deletions. Although the Tbx1 null phenotype is achieved with both lines, the Tbx1Cre allele (which is heterozygous for Tbx1) will almost certainly affect Tbx1 expression in the pharyngeal ectoderm and endoderm and this will change the gene expression profile compared to the Mesp1Cre allele (even non-cell autonomously). Please justify this in the results section.

It is stated that "In the Swiss Webster background, Tbx1Cre/+ and Tbx1+/- heterozygous mice have no heart or aortic arch defects 33 and thus serve as controls". Although Tbx1- hets usually exhibit 4th PAA defects on other genetic backgrounds, even when they do not it is expected that there will still be transcriptome changes related to Tbx1 haploinsufficiency (as mentioned above). Have the authors demonstrated that there is no change in gene expression between Tbx1-hets and wild-type embryos, i.e. by comparing data between Mesp1Cre;Tbx1+/+ and Tbx1Cre/+ controls? The statement "Because of the difficulty to distinguish the sex at those time points, we used both male and female embryos for all the experiments" is not appropriate. Sex genotyping by PCR using Y chromosome specific primers is easily performed at the same time as routine genotyping (was this done? See comment below). The sentence should be modified to just state that embryo sex was not determined.

"The microdissected tissues were pooled in DMEM (GIBCO, Cat# 11885-084) until all the dissections were completed." At what temperature were the samples maintained at during the dissections?

More importantly it is not clear how many biological replicates have been analysed in this study. Table 1 shows that multiple embryos were collected from most genotypes, but the methods

section implies that the embryo material was pooled at the time of dissection. This can only be possible if the embryos do not need genotyping, but the breeding scheme describes that a wild-type *Tbx1* allele is present in the male for *Mesp1Cre* (*Mesp1Cre/+;Tbx1flox/+* male mice, crossed with *ROSA26-GFPflox/flox;Tbx1flox/flox* female mice). So the embryos in each litter will be a mix of *Tbx1+/-* and *Tbx1f/f*. This is also the case for the *Tbx1Cre* line (*ROSA26-GFPflox/flox;Tbx1flox/flox* female mice were crossed with *Tbx1Cre/+* male mice), so the embryos in each litter will be a mix of *Tbx1+/-* and *Tbx1f/Cre*. Please clarify how embryos of each genotype were collected and identified.

The breeding scheme explicitly states that *Tbx1Cre* was used for RNA-seq and ATAC-seq, but this is not stated for the *Mesp1Cre* line, although the results section says that these two lines were analysed together as the *Tbx1* cKO.

Does 10' mean 10 mins?

The statistical tests used should be detailed in the methods section.

Why were some experiments done at E8.0 and E8.25 but others done at E8.5?

Why was ATAC-seq / ChIP experiments only performed at E9.5?

Results

Table 1: The data shows that only one *Mesp1Cre;ROSA26-GFPflox/+;Tbx1flox/flox* was collected at E9.5. Is this the reason that the cKO group contains *Tbx1*-null embryos from both *Cre* alleles? Can sufficient analysis be performed when only one biological replicate is present for the *Mesp1Cre* deletion? Is the *Tbx1Cre* Cko deletion in fact also one biological replicate? (see above). If so, how can any analysis be confidently carried out without any replicates? Also, why were more cells submitted and captured from this one sample (13,788 and 5,157) when only 12,000 and 4,055 were acquired from six *Mesp1Cre; ROSA26-GFPflox/+* embryos? Further analysis includes comparing *Mesp1/Tbx1* cKO at E9.5 with only the *Tbx1* cKO at E8.5. As these are two different models can they be directly compared?

Figure 1 shows data analysis from the four developmental stages selected, where mesoderm has been FACS as GFP labelled cells from the *Mesp1Cre* allele. On page 4 it states "To better understand the developmental connection, we integrated the four time point datasets..." but it is not clear how this developmental connection is being shown. Clearly there is a lot of anatomical changes occurring between E8.5 and E10.5 in pharyngeal arch development. The concluding sentence states "that CPM progenitors can be distinguished from more mature CPM states by their multilineage primed gene signatures" but it is not clear from Figure 1 how this was determined from a mixture of embryonic stages. Overall this section therefore needs some clarification and more explanation as to why this experiment was done and what it actually shows.

Figure 1B: include scale bars.

Figure 1F. Consider indicating the CPM, MLP and BrM genes within the figure for clarity. It would also be useful to clarify in the main figure which genes are also in other groups e.g. C3, C1, C18 to reduce jumping between figures and text.

Page 6. The 4th pharyngeal arch is mentioned twice in this section but is not labelled on the figure panels. This will help with identifying the structure being referred to.

The term "newly appreciated" for *Aplnr* and *Nrg1* should be reconsidered as these genes are already known and described.

Figure 2A legend. Do you mean that cluster numbers are consistent with 1D? Figure legend should be stand-alone – define clusters in legend again instead of referring to Figure 1. Please summarise in the text what the 6 main branches are.

Figure 2B and 2C are referred to, alongside Supplementary Figure 2B and 2A. The supplementary panels should be switched around so that the authors are referring to the appropriate panels.

Figure 2I/J/K/L/M. Include scale bar on all images

Figure 2H-M. Why is E8.5 used in this figure while E8.0/E8.25 was used in previous figures? Please clarify why time points are not the same for each experiment.

Figure 2H/I/J. What line/genotype are the WTs from?

Heat map in Figure 3B and Figure 3C don't match up with the description of stages in the legend. B legend and text says E8, E8.25 while C legend says E9.5, E10.5, but all 4 time points are illustrated in the figure. Please correct this. Also clarify colour code in the figure legend for E8-10.5.

Please refer to panels in order – A&D, B&E, C&F should be moved around so that they are referred to appropriately within the text.

Figure 4A/B: add scale bars. Why are the experiments done in the *Mesp1Cre* line only at E9.5 but

at E8.5 and E9.5 in the Tbx1Cre line? Why was the E8.5 used here instead of E8.0 and E8.25? Is there a statistical test to compare CTR/KO at E8.5 and E9.5?

This is another example of not referring to panels in order – please move around so that panels are referred to in the correct order in the text.

Figure 5: Please clarify why the Mesp1Cre line was used as the control when both Mesp1Cre and Tbx1Cre cKO lines were used.

Fig 5G, H. Panels should specify which Cre mutant is being shown. Include scale bars on all images.

The authors say in the text that the affected genes are involved in cell differentiation or cell signalling. From the GO terms there is muscle cell differentiation but presumably a lot of genes will be involved in cell signalling – what signalling are they involved in?

When referring to Figure 5 in the text, the authors refer to Tbx1 cKO (for the RNAScope staining) instead of clarifying whether the embryos are Mesp1-Cre or Tbx1-Cre. It would be useful to refer to each one specifically in the text, and to indicate this on the figure panels.

Figure 6: What do the grey regions in the pie charts represent?

Page 10. It is not clear which genotypes were analysed for ATAC-seq and ChIP-seq experiments, only “control versus Tbx1 mutant embryos” is stated. Figure 6A shows that only Tbx1Cre cKO embryos were used for ATAC-seq but this should be confirmed in the text. Why was only this line used whereas the Mesp1 cKO was also used in the RNA-seq analysis?

In this section replicates are mentioned. How were these prepared? as it is not mentioned in the ATAC-seq methods section. For ChIP-seq it states that the Tbx13'-Avi;BirA embryos were collected and 20 pooled for one sample. How many pools/samples were used in the experiment? For the RNA-seq embryos a detailed table was provided. There is no equivalent details for the other experiments.

Figure 7: Why were Tbx1 hets assessed separately here when a previous results section stated that the hets and WTs were pooled?

Discussion

First sentence is very vague. Please re-write so it puts the study results in context.

The model proposed in Fig7F shows that Tbx1 deficiency in the CPM causes caudal pharyngeal hypoplasia and the developmental consequences of this. As the caudal arches do not form properly it is important to know that this change in tissue architecture is not the reason for the differential gene expression in cKO embryos compared to controls with normal arch morphogenesis.

Can the authors explain how the Tbx1-null mouse model relates to the gene regulation events in 22q11DS patients, who are hemizygous for TBX1? These patients show a range of clinical phenotypes despite all lacking one allele of TBX1. How does your data fit with arch artery defects such as IAA?

Reviewer #3:

Remarks to the Author:

Review of Nomaru et al.

Summary

The work presented in this manuscript defines a population of Mesp1-derived multi-lineage progenitor cells in the cardiopharyngeal mesoderm (CPM) that give rise to second heart field (SHF) and branchiomeric muscle (BrM), and attempts to ascribe a specific functional role to Tbx1 within this population. The work builds on clonal lineage studies and Tbx1 loss of function studies from several groups that demonstrated that SHF cells and BrM cells have a common progenitor in the CPM and that Tbx1 is required for CPM to differentiate into SHF and BrM. The novelty of the present study is using clustering analysis on scRNA-seq from Mesp1-derived CPM cells to molecularly and spatially define a specific progenitor population at several timepoints during CPM expansion and differentiation that is dubbed a “multilineage primed progenitor” (MLP). MLPs express both BrM and SHF genes and are located bilaterally in the lateral part of the caudal pharyngeal apparatus. To define the role of Tbx1 in this population, scRNA-seq was performed in control and conditional Tbx1 knockout embryos using Tbx1-Cre and Mesp1-Cre driver lines. In the absence of Tbx1, both models demonstrated that MLPs were present, but failed to differentiate as

efficiently towards SHF and BrM fates, leading to MLP accumulation. Additionally, within the MLP population, genes involved in cell differentiation and signaling decreased, and expression of some genes associated with non-mesodermal lineages was observed. Specifically, *Aplnr* (MLP gene) was downregulated while *Pax8* (non-mesodermal gene) was upregulated. To identify target genes of *Tbx1* in the MLP, the authors use ATAC-seq in control and conditional KO embryos, as well as *Tbx1* ChIP-seq using a novel Avi-tagged knock-in *Tbx1* allele. The authors define a set of differentially accessible chromatin regions (DARs) between control and knockout embryos, the vast majority of which demonstrated reduced accessibility in the knockout. Intersecting ATAC-seq DARs with MLP-enriched genes and *Tbx1*-bound regions identified with ChIP-seq defined 8 putative direct *Tbx1* targets in the MLP, including *Aplnr* and *Nrg1* as well as *Isl1* and other genes. Based on the results of these experiments, the authors advance the following conclusions:

- 1) A multilineage primed progenitor (MLP) population in the CPM can be molecularly identified through co-expression of *Aplnr*, *Nrg1*, and *Tbx1*.
- 2) MLPs are maintained as a source of progenitor cells in the CPM, deploying cells to the heart and branchiomeric muscles during development.
- 3) *Tbx1* promotes progression of MLPs to more differentiated cell states by direct and indirect regulation of a set of defined transcriptional targets.

General Comments

This manuscript addresses an important problem using a powerful combination of computational and genetic tools, and along the way contributes several new and useful datasets to the field. The datasets are generally consistent with one another and mutually reinforcing, with some minor exceptions noted below. The work may serve as a springboard for future studies on more detailed mechanisms that regulate differentiation of the MLP population. I do have some concerns, however, about how some of the experiments are interpreted, and whether alternative explanations for the findings presented are adequately considered or excluded. While the data do support qualified versions of the conclusions enumerated above, the manuscript could be strengthened by acknowledging areas where the data are insufficient to support broad conclusions and also by more clearly delineating how this work adds to the body of literature already published on the role of *Tbx1* in the CPM.

Specific Comments

1. In the introduction, the authors state, "Although there are many studies of *Tbx1*, we do not yet understand its functions on a single cell level, which is needed to elucidate the true molecular pathogenesis of 22q11.2DS" Since the role of *Tbx1* in the CPM has been explored previously and it is known that *Tbx1* is required for differentiation from CPM to BrM and SHF, a clearer statement in the introduction about what has been established/hypothesized about the role of *Tbx1* in the CPM, along with a clearer statement of why scRNA-seq is needed to fill these knowledge gaps would be very helpful for readers who may have less familiarity with the literature on CPM. Similarly, a more thorough discussion of the previous literature on *Tbx1* function in the CPM in the Discussion would be helpful to better highlight the specific ways in which the present work advances the field.

2. The existence of a multipotent progenitor in the CPM has been established with clonal fate mapping in prior studies, but the molecular identity of that population has not been established. In the present work, the MLP is defined through a clustering and pseudotime/PAGA analysis rather than by clonal fate mapping. The authors provide several lines of evidence that support identifying the MLP with the multipotent progenitor defined in prior work, but without clonal fate mapping of the MLP cells *in vivo*, one cannot conclude that individual MLP cells are actually multipotent. Therefore, a statement in the Discussion clarifying this point would be helpful and would limit the potential for over-interpretation.

3. Related to the latter point, can the authors comment on the presence of the MLPs at E10.5? Given what is known about branchial arch development and timing of addition of cells to the SHF, is it expected to see MLPs this late in development or is it likely that these cells are already specified to either BRM or SHF or other MLP derivatives?

4. A strength of the work is the authors' use of 2 genetic models to knock out *Tbx1* with comparison of the results. However, since *Tbx1* is expressed in other *Mesp1+* populations beyond

the MLP, it's possible that some of the observed changes in percentages of different cell populations as well as some of the gene expression changes within cell populations, are non-cell-autonomous, and this possibility should be addressed when discussing the results of Tbx1 knockout experiments. The authors conclude that Tbx1 plays a specific role in MLP through regulation of defined targets but cannot exclude the possibility that Tbx1 plays a role in other cell types that is relevant to what happens to the MLP.

5. Tbx13'-Avi mouse line. Since these mice survive and breed as homozygotes, the presumption is that there is no deleterious phenotype associated with the knock-in and that this allele is not hypomorphic, but it would be ideal to state this explicitly as if affects interpretation of the ChIP-seq data. Related to this, can the authors comment on the relatively small number of ChIP-seq peaks identified despite a very large number of DARs in the knockout with reduced chromatin accessibility? Is this because there are actually relatively few binding sites for Tbx1 and many effects on chromatin are indirect, because of low expression of Tbx1 in this population, low cell numbers for the experiment, or because Tbx1 might have other functions that are not related to DNA binding as has been observed by other researchers?

6. The statement in the results (p.10), "We suggest that Tbx1 provides a balance of specific gene expression required for MLP function" is vague and should ideally be rephrased since It's not clear what is being "balanced".

7. The authors define 8 direct transcriptional targets of interest for Tbx1 in the MLP and present ChIP-seq browser tracks for 2. While the peak for Nrg1 overlaps a Tbx1-dependent ATAC-seq peak the Aplnr peak appears to bind closed chromatin that is far from the DAR, making the mechanism whereby this Tbx1 binding event activates Aplnr in the MLP somewhat unclear. Can the authors identify any other datasets (e.g., ENCODE or other embryonic dataset) that suggest the presence of regulatory element at this site?

8. Previous work defined a role for Tbx1 in regulating epithelial properties of the SHF. Were DEGs related to cell polarity or tissue morphogenesis observed in the knockout or as direct targets from the ChIP-seq?

9. The Discussion states, "Interestingly, both Aplnr and Nrg1 are direct target genes of TBX1 based on our ATAC-seq and ChIP-seq results, suggesting that these genes are mediators of TBX1 function in MLPs." The data presented don't define any specific role for Aplnr and Nrg1 in the MLP population beyond their expression as marker genes, so this statement is overinterpretation.

10. Venn diagrams showing overlap of different gene sets between different experiments (e.g, Figures 6F and 7B) look highly significant but should still have accompanying chi squared tests and P values presented alongside the data.

August 26, 2021

Here is a summary of our experiments and changes:

- 1) We originally performed one scRNA-seq experiment of *Mesp1^{Cre}* (control) versus *Mesp1^{Cre};Tbx1^{fl/fl}* (conditional knockout) at E9.5. We now performed a replicate experiment and obtained similar gene expression results that further strengthen our main conclusions (updated Figs. 5-8). One of the secondary conclusions of the study with one replicate is that the MLP population expands while certain other populations in the CPM are relatively reduced. In the new analysis, we did not find consistent results supporting the conclusion and we modified the text accordingly.
- 2) We performed a new scRNA-seq experiment on wildtype versus *Tbx1^{-/-}* embryos. We obtained similar results for gene expression changes as from using *Mesp1^{Cre}* and *Tbx1^{Cre}* alleles. This is now included as supplemental data (Suppl. Fig. 9). Overall, this increases the confidence in the findings.
- 3) There were major questions about the *Tbx1* locus affected by the *Cre* allele with genetic background effects. We now explain in the text that our mice have been backcrossed over 15 generations in the SwissWebster background. We explain that we do not observe heart defects in either *Tbx1^{Cre}* or in a different *Tbx1^{+/-}* heterozygous allele, as maintained in the same background, as we have published (PMID: 31412026). We obtained similar caudal pharyngeal hypoplasia in *Tbx1* mutant embryos of different genotypes (*Tbx1^{Cre/fl}*, *Mesp1^{Cre};Tbx1^{fl/fl}* or *Tbx1^{-/-}*) and obtained similar scRNA-seq results using such genotypes as now included in the manuscript. The global and conditional null embryos in different genetic backgrounds and of different *Tbx1* mutant alleles in different labs have a fully penetrant persistent truncus arteriosus.

- 4) We include a new figure, Fig 4, in which we directly compared *Mesp1^{Cre}* and *Tbx1^{Cre}* control scRNA-seq data.
- 5) We did not explain how we compared mice of different control, heterozygous and conditional null phenotypes, which led to much confusion in reading the manuscript. We have now included a figure of our mating strategy (Supplementary Fig 1) and explain our experimental design, where we use *Mesp1^{Cre}* to filter data obtained using the *Tbx1^{Cre}* allele, to select the CPM populations, to increase the rigor of the findings.
- 6) We made a very serious attempt to determine the mechanism of *Aplnr* and *Nrg1* function in MLPs, but did not succeed. We obtained *AplnrCreERT2* mice from Dr. Red-Horse to test if the *Aplnr* lineage can migrate to the cardiac outflow tract. We noticed that *Aplnr* is also expressed in the dorsal pericardial wall harboring cells after they have left the MLP and migrating to the outflow tract. Therefore, we realized that even if we can find that the *Aplnr* lineage can migrate to the heart from the MLPs, expression in the dorsal pericardial wall, where *Tbx1* is also expressed, would making it difficult to interpret results, therefore we decided not to include these findings. We obtained *Nrg1* floxed mice. Unfortunately, the mice we received was a floxed allele in which loxP sites flanked the IG domain that would not result in a complete null. We crossed the mice using *Tbx1^{Cre}*, but we didn't find an abnormal phenotype. We obtained sperm from a different *Nrg1* floxed allele that should result in a complete null after recombination, however, none of the founders and none of the offspring from IVF have the floxed allele. Therefore, we must have received the wrong aliquot of sperm.

Referee #1. (comments in bold font, our response in regular font)

The authors perform single cell RNA-seq of *Mesp-Cre* traced cells from the rostral part of the embryo proper to identify CPM cells and their derivatives. From this study, the authors make a number of interesting observations, including that a population of undifferentiated, multipotent CPM is maintained over the time period of study along the lateral aspect of the pharyngeal apparatus, from where it contributes differentiating cells into the heart and BrM. They further show, by evaluating the same cells in knockout embryos, that *Tbx1*, a gene whose reduction in copy number gives rise to congenital heart defects in humans, is an important regulatory factor, promoting differentiation of CPM. They go on to perform ATAC-seq in *Tbx1* mutants and ChIP-seq for *Tbx1* at E9.5 to identify genes that are directly regulated by *Tbx1* within the CPM. This provides important mechanistic insight into the function of *Tbx1* within CPM and the teratogenic effects of the 22q11.2 deletion in humans. The study is comprehensive, well written, and fairly interpreted, and will be of interest to a wide audience.

I have only one major concern:

1. The authors mention that on the Swiss Webster background, *Tbx1* heterozygotes do not manifest haploinsufficiency for cardiac development, and that (presumably for this reason) all mice are kept on the Swiss Webster background. However, the methods do not explain how this was done, as the genetic tools were not all generated on the Swiss Webster background. Were they independently backcrossed, and if so, for how many generations?

Was whole-genome SNP analysis performed to ascertain the degree of homozygosity on the Swiss Webster background? One significant concern when backcrossing from the strain in which the genetic modification was done onto another is that the region proximal to the modification will not be converted to the new strain. If modifiers happen to exist in this region, their effects will be unknowingly assigned to the modification. I.e., is an effect, such as a gene expression difference or loss of a set of ATAC-seq peaks,

due to the presence of the Cre, or to a linked modifier brought over from the previous strain?

This problem becomes smaller with greater backcrossing, and more defined with determination of congenic breakpoints. This is a particular concern because evidently there are modifiers of the *Tbx1* effect on CPM development that distinguish Swiss Webster from the strains in which the genetic tools were made (probably 129). The authors should provide as much detail as possible regarding the backcrossing and check the size of congenic domains by whole genome SNP analysis in the methods section, perhaps mention the backcross N in the results section, and briefly address the concern that modifiers may be confounding their interpretations.

Response: We now explain in the text that our mice are backcrossed over 15 generations in the SwissWebster background. We do not observe heart defects in either *Tbx1^{Cre}* or in a different *Tbx1^{+/-}* heterozygous allele, yet we observe the same defects conditional and null mutant embryos. Both *Tbx1^{Cre/+}* and *Tbx1^{+/-}* mice have been used by us in our experiments. These two alleles were generated in different labs (*Tbx1^{+/-}* mice were generated in our lab; *Tbx1^{Cre}* in Dr. Baldini's lab) and were targeted in different regions of *Tbx1*, but they have the same phenotype in SwissWebster. They have no cardiac defects but they do have parathyroid gland malformations (PMID: 16452092). Specifically, *Tbx1^{+/-}* mice have a knockout of exons 2-3. *Tbx1^{Cre/+}* mice have a knock-in of *Cre* to exon 5 (PMID: 17610275). The lack of a cardiac phenotype was observed in the *Tbx1^{Cre/+}* and *Tbx1^{+/-}* mice, as we previously reported (PMID: 31412026). This suggests that the lack of a cardiac phenotype in *Tbx1^{Cre/+}* or *Tbx1^{+/-}* mice is due to the genetic background of SwissWebster. The *Tbx1^{Cre/f}* and *Tbx1^{-/-}* embryos have neonatal lethality with a persistent truncus arteriosus, cleft palate and absent thymus/parathyroid glands, which is the same as reported in the mice as maintained in C57Bl/6 (PMID:11242110). In the conditional null *Mesp1^{Cre};Tbx1^{flox/flox}*, *Tbx1^{Cre/flox}* and global null *Tbx1^{-/-}* embryos, pharyngeal arches 3-6 are hypoplastic.

We now include our mating strategy in the Methods section to include the backcrossing and intercrossing.

Lines 493-498 in the Methods section now states, "*Mesp1^{Cre}*⁷, *Tbx1^{Cre}*⁷², *Tbx1^{ff}*⁷³, *Tbx1^{+/-}*⁷³, *ROSA26-GFP^{ff}* (*RCE: loxP*)⁷⁴. *Tbx1^{Cre/+}*, *Tbx1^{ff/+}*, *Mesp1^{Cre/+}* and *ROSA26-GFP^{ff/+}*, single heterozygous mice have been backcrossed in SwissWebster strain for over 15 generations. The *Tbx1^{ff}*, *ROSA26-GFP^{ff}* and *ROSA26-GFP^{ff};Tbx1^{ff}* homozygous mice were inter-crossed in brother x sister crosses for over 20 generations and constitute inbred lines. The breeding strategies for all the experiments are illustrated in Supplementary Fig. 1."

Lines 186-187, in the Results section now states, "The *Tbx1^{Cre}* allele has a knockin of *Cre* that inactivates one copy of the *Tbx1* gene. The *Tbx1^{Cre}* mice do not have cardiac or aortic arch defects in the SwissWebster background³⁶. "

Lines 204-206, in the Results section now states, "*Mesp1^{Cre}* and *Tbx1^{Cre}* mediated *Tbx1* conditional null embryos have similar phenotypes including hypoplasia of the caudal pharyngeal apparatus and a fully penetrant persistent truncus arteriosus^{16,33}. "

Further, we used the *Mesp1^{Cre/+}* ATAC-seq data to compare with *Tbx1^{Cre/+}* (heterozygous) ATAC-seq data to identify common chromatin accessible regions to extract CPM open chromatin regions. We used these for further analysis to compare with conditional null mutant embryos. By using this filtering method, we reduce confounding effects that may be specific to different alleles or background effects.

Lines 419-426, In the Discussion now states, “ Further, there are limitations of using the *Tbx1^{Cre}* allele to compare with gene expression changes in *Tbx1^{Cre}* conditional null mutant embryos. In *Tbx1^{Cre/+}* control embryos, one copy of *Tbx1* is inactivated. However, we did not observe cardiac defects in *Tbx1^{Cre}* heterozygous embryos or *Tbx1^{+/-}* embryos in which a different region of *Tbx1* was inactivated, as maintained in the Swiss Webster background. Although there are likely some gene expression changes in *Tbx1^{Cre/+}* embryos versus wildtype embryos, we obtained similar findings when we directly compared scRNA-seq results from *Mesp1^{Cre}* versus *Tbx1^{Cre}* experiments. ”

Additional comments:

2. A number of genes fall out of the pseudotime analysis as changing within the undifferentiated population of CPM over time. Can the authors confirm these predicted changes using RNA-scope for specific examples, for example by showing E10.5 RNA-scope data for *Apln4* or *Sema3c* side by side with E8.5?

Response: We now updated Figure 3 with RNAScope data (new Figure 3g). We performed RNAScope analysis of 3 replicates of *Aplnr* (early gene), *Sema3c* (late gene), *Tbx1* (early gene), *Nkx2-5* (early and late) and *Isl1* (early and late) at E9.5 and E10.5. We selected E9.5 rather than E8/8.25, because both early and late genes are expressed at this time point. Further, *Sema3c* is not expressed in CPM at E8/8.25.

Lines 163-176 now states, “An important question is whether MLPs as CPM progenitors, maintain the same state based upon gene expression over time. To address this, we examined differentially expressed genes in MLPs versus other populations from E8-10.5. We identified core CPM genes that are expressed similarly at all time points, including *Isl1*, *Mef2c* and *Nkx2-5* (Fig. 3a). However, we also found that early expressing genes such as *Aplnr*, *Nrg1*, *Irx1-5*, *Fgf8/10* and *Tbx1* (Fig. 3b) are reduced over time, with increasing expression of cardiac developmental genes such as *Hand2*, *Gata3/5/6*, *Bmp4* and *Sema3c* (Fig. 3c). These differences are shown in violin plots as well (Fig. 3d-f). *Nkx2-5* and *Sema3c* are expressed in the caudal pharyngeal apparatus at E9.5, like that of *Isl1*, *Tbx1* and *Aplnr*, defining MLPs (Fig. 3g). The MLP region was reduced in size in the caudal pharyngeal apparatus at E10.5 (Fig. 3g). Further, expression of the early MLP marker *Aplnr* wasn't observed at E10.5, while co-expression of *Sema3c*, *Nkx2-5* and *Isl1* occurred strongly in the OFT (Fig. 3g). This is consistent with the model that the MLPs continuously allocate progenitor cells to BrMs and OFT-CMs, while showing maturation themselves by E10.5 (Fig. 3h). ”

3. In comparing the *Mesp1-Cre* with the *Tbx1-Cre*, the authors state that the *Mesp1* lineage contributes more broadly to embryonic mesoderm, while *Tbx1* contributes to pharyngeal endoderm and ectoderm. They provide two references for the extra-CPM activity of *Tbx1*, but none for that of *Mesp1*.

Response: *Mesp1^{Cre}* description is provided in references #7 and 8. We added a new figure, Fig. 4 to better explain the two different lineages. We directly compared *Mesp1^{Cre/+}* and *Tbx1^{Cre/+}* scRNA-seq data at E9.5, after data integration. In this figure, the different cell types that are unique to one or the other *Cre* allele, and the cell types that comprise the CPM is shown. We also include a diagram (Fig. 4f) that shows that the CPM shared between both alleles is further examined in this manuscript.

Lines 178-200 in the Results section now states,

“The intersection of the *Tbx1* and *Mesp1* lineages helps to identify the CPM

Tbx1 function in MLPs and roles in derivative CPM cells on a single cell level are unknown. We therefore examined the *Mesp1* and *Tbx1* lineages in control embryos to

understand how the CPM lineages compare in relation to *Tbx1*. The *Mesp1* lineage contributes more broadly to the embryonic mesoderm, while *Tbx1* is expressed in pharyngeal endoderm and distal pharyngeal ectoderm³², in addition to the CPM³³. Although *Tbx1* is strongly expressed in the CPM, it is not expressed in the heart, neither in the FHF nor the caudal and medial pSHF at the timepoints analyzed^{22,34,35}. The intersection of these two datasets defines the CPM more precisely.

The *Tbx1^{Cre}* allele has a knockin of *Cre* that inactivates one copy of the *Tbx1* gene. The *Tbx1^{Cre}* mice do not have cardiac or aortic arch defects in the SwissWebster background³⁶. We then performed scRNA-seq of the *Tbx1^{Cre}* lineage at E9.5 and integrated this with data from the *Mesp1^{Cre}* lineage at the same stage to compare the characteristics of the two lineages (Fig. 4a-c; Supplementary Table 2). Data integration provides consistency in defining common cell types among different samples, in addition to removing batch effects³⁷. We found that the CPM can be identified in both populations (Fig. 4a; bold font). As expected, the *Mesp1* lineage includes the FHF, which is not included in the *Tbx1* lineage, while the *Tbx1* lineage includes the pharyngeal epithelia and otic vesicle, not included in the *Mesp1* lineage (Fig. 4b-c). The relative proportions of CPM populations are shown in Fig. 4d. The pSHF in the *Mesp1^{Cre}* lineage includes the caudal pSHF with lung progenitors that is not included in the *Tbx1^{Cre}* lineage³¹. The MLPs are found in both lineages, marked by expression of *Isl1*, *Aplnr*, *Tbx1* and *Nrg1* (Fig. 4e). Therefore, the data from scRNA-seq using *Tbx1^{Cre}* helps define the CPM better and serves as a replication for the data on the CPM from scRNA-seq using *Mesp1^{Cre}*, which is relevant to *Tbx1* as shown in Fig. 4f. “

#Referee 2

General comments

This paper investigates the role of a multilineage-primed population (MLP) within the cardiopharyngeal mesoderm (CPM) that contributes to cardiac and skeletal cell differentiation. It specifically focusses on *Tbx1*, the gene central to complications in 22q11 deletion syndrome and identifies a mechanism, through single cell-based technologies, as to how *TBX1* directs the development of the pharyngeal apparatus by marking the MLP population.

Major concerns

The use of two different *Cre* lines to conditionally delete *Tbx1* from the developing embryo, particularly for the RNA-seq experiments, is very confusing. Firstly, only one *Mesp1*;*Tbx1* cKO embryo was used for this experiment, and it also appears that only one pool of *Tbx1*Cre;*Tbx1* cKO embryos was used. Not only does this mean that only one biological replicate was used for each *Cre* line for the whole RNA-seq study, these two distinct *Cre* lines were also pooled together as a general cKO line. This does not seem appropriate to me as each *Cre* works in different tissues where *Tbx1* is active and there is no ability to verify the accuracy of the sequencing with data from a separate biological replicate from each *Cre* line.

Response: We now performed a replicate of *Mesp1^{Cre}*;*Tbx1^{+/+}* versus *Mesp1^{Cre}*;*Tbx1^{fl/fl}* at E9.5. The new data is provided in Figs 5-8, Supplementary Figs 1, 5, 7, 10, 11, 12 and Supplementary Tables 3, 4, 7, 8, 9, 10, 12. We also performed a wildtype vs *Tbx1^{-/-}* scRNA-seq experiment at E9.5, where we didn't purify lineages to see if the main findings can be replicated, and this is shown in a Supplementary Fig. 9. We now explained our mating strategy in Supplementary Fig 1; and study design shown in Fig. 4f and 6a (6a was included before).

Our writing was very confusing and we apologize for that. In our design, we used two different *Cre* alleles and generated separate datasets. Our thought is that two independent *Cre* lines is

better than use of one. It enabled us to evaluate gene expression in common in the two *Cre*'s, which define the CPM. There are differences in the two *Cre* lines but they were not our focus. Note that Reviewer #3 commented on this, "**4. A strength of the work is the authors' use of 2 genetic models to knock out *Tbx1* with comparison of the results.**"

Essentially, we used the resulting scRNA-seq data from the *Mesp1* and *Tbx1* *Cre* lines as a filtering process, in which we used the CPM genes shared in both lines to study further in our experiments. This allowed us to have an independent replication using a second *Cre*. We did this because *Tbx1* is expressed in other populations that are not mesodermal, and *Mesp1* is expressed in other mesodermal population that are not relevant to *Tbx1*; therefore, genes that were changed in both conditions are those relevant to the CPM. We previously used the phrase "Tbx1 cKO" that was terribly confusing. So, we now have changed the wording in line with the comparisons. We use the phrase, "*Mesp1*^{Cre} Ctrl vs cKO embryos" to designate *Mesp1*^{Cre} controls vs *Mesp1*^{Cre}; *Tbx1*^{ff} cKO embryos. Separately, we use the phrase, "*Tbx1*^{Cre} Ctrl vs cKO as *Tbx1*^{Cre/+} control vs *Tbx1*^{Cre/f} cKO embryos. In the new Fig 4, we integrated the *Tbx1*^{Cre} vs *Mesp1*^{Cre} scRNA-seq data at E9.5 to compare and highlight the similarities and differences. The reviewer might wonder why we did not analyze each sample independently, vs "pool" them together. Integration of data from cells from different samples are necessary so that we can cluster them together to align cell types (i.e., clusters). Otherwise, we could have a hard time to link cell clusters from different samples in a global and consistent way, because the marker genes identified for the same cell type will not be identical if we analyze each sample separately – we have experienced it. Furthermore, we need to normalize or batch-corrected gene expression across samples in order to define if genes show consistent expression changes. Integration of datasets also increases the power for identifying small (or rare) cell types. Once the integration is done, we can perform any pair-wise sample comparison. Fig 4f shows a diagram that illustrates the relationships between the two lineages. This filtering also makes it possible to distinguish between the first and second heart fields in the *Mesp1*^{Cre} data, that would otherwise be difficult to separate. Please see the wording changes in the manuscript in the paragraph above for a similar comment from Reviewer 1.

The authors state: "In *Tbx1* cKO embryos at E9.5, the BrM and CM/OFT populations were significantly smaller than in controls". This is not surprising as the caudal pharyngeal arches do not form in the *Tbx1* cKO embryos, so there is relatively less tissue present. Have the authors compensated for this change in anatomy with the single cell analysis? Has this, for example, been applied to the data in Figures 4 and 5?

Response: The reviewer raised a very good point. We re-analyzed our scRNA-seq data to address the concern of cell type proportions and we included new data from the second replicate of *Mesp1*^{Cre} Ctrl vs cKO and two stages of *Tbx1*^{Cre} Ctrl vs cKO (original Fig. 4, new Fig. 5). We agree with Reviewer 2 that there are some technical issues in this type of analysis, including how to account for the anatomic different of the samples used for data generation. Upon re-analysis we found that there was variation in the CPM cell type proportions in different replicates/time points in the controls (Ctrl) and mutant (cKO) embryos. We therefore removed the cell type proportion bar graphs from original Fig. 4e and 4i, and the corresponding results, and left this question for future study.

We now include the ratio of numbers of MLPs in lines 225-228, which now states, "The ratio of number of MLPs compared to the total number of cells in each replicate of *Mesp1*^{Cre} Ctrl versus cKO (first replicate of WT, WT1 is 0.07 [284/4046 cells], WT2 is 0.129 [912/7048] vs first replicate of cKO, KO1 is 0.101 [474/4689] and KO2 is 0.108 [1056/9750]) shows variation, nonetheless, they are present and in both sets of cKO embryos versus controls (Fig. 5c-f)."

There are also issues with the low number of cells collected. The data in Table 1 shows

that a very low number of cells were captured overall, and particularly for the *Tbx1*^{Cre} genotypes at E8.5. Sequencing a low number of cells is likely to result in the loss of rare cell populations in the analysis. Can the authors please comment on this?

Response: We agree with the reviewer that the larger the number of cells is sequenced, the higher the chance of capturing rare cell types. Here we analyze cells from specific lineages (*Mesp1* or *Tbx1* lineages). Ideally, we would like to sequence >5,000 cells in each of our samples, but the starting embryonic cells are extremely small, especially at E8-8.5. By analyzing multiple samples separately from the two *Cre* lineages, and multiple time points, as well as focusing on a shared cell population (CPM), we believe that we have overcome potential under-sampling problems. Also note that we did not study cells that were detected only in *Tbx1*^{Cre} or *Mesp1*^{Cre} data. Nevertheless, we have added to Discussion in the revised manuscript that we could have missed some rare cell populations by our approach.

Lines 417-419 in the Discussion now states:

“Since the cell number isolated from these embryos was relatively small, it is also possible that rare populations would be missed, so that using this strategy reduces the concern of under sampling.”

The use of different time points (e.g. E8.0/E8.25 in some figures but E8.5 in others) needs to be justified. Also, the use of control lines needs more clarification - there are some experiments where *Tbx1* hets and *Tbx1* WT are grouped together for a control group, but others where the *Tbx1* hets and cKOs are compared. All experiments should be done with appropriate and consistent controls (e.g. line-specific controls) at appropriate and consistent time points.

Response: We were not clear in our writing of control vs mutant experiments. We now added Supplementary Fig 1 to explain our crosses and lines used. We did not group together embryos or cells from different lines; we did compare datasets, and now better explain all our comparisons throughout the text. We added a new Figure, Fig 4, to better explain and compare our control lines with strategy. Fig. 6a (now, grayscale) explains our strategy for identifying DEGs between control and mutant embryos. It explains that we obtained DEGs from scRNA-seq between control and mutant embryos of *Mesp1*^{Cre};*Tbx1*^{+/+} vs *Mesp1*^{Cre};*Tbx1*^{fl/fl} and separately obtained DEGs from scRNA-seq between control and mutant embryos of *Tbx1*^{Cre/+} vs *Tbx1*^{Cre/fl}. Then the DEGs that occurred in both, were examined further; these are DEGs in the CPM. DEGs found only in the *Mesp1*^{Cre} experiment or only in the *Tbx1*^{Cre} experiment were excluded. Figure 7a (unchanged) explains our strategy where we analyzed two biological replicates of ATAC-seq of *Tbx1*^{Cre/+} vs *Tbx1*^{Cre/fl} embryos at E9.5. We then wanted to focus only on the CPM, so we used the *Mesp1*^{Cre/+} ATAC-seq data to filter the data from the *Tbx1*^{Cre/+} vs *Tbx1*^{Cre/fl} experiment, to remove cell types unrelated to CPM.

In regards to the different stages of E8, E8.25 vs E8.5: We chose four stages in the *Mesp1*^{Cre} study, which was performed first and was needed to help select time points for the *Tbx1* study. Due to the technical complexity of the crosses and trying to match somite counts between control and mutant embryos within each experiment, there was some slight difference in stage between the *Mesp1*^{Cre} and *Tbx1*^{Cre} experiments, being E8.25 and E8.5. To describe the timepoints precisely, we called E8.25 for 6-7 somites and E8.5 for 8-10 somites as in Table 1. We focused on E9.5 for functional genomic studies (scRNA-seq, ATAC-seq and ChIP-Seq). This is because this is the stage when *Tbx1* expression is the strongest and when the cardiac OFT is elongating.

Lines 213-215 now states, “The E8.5 stage used (8-10 somites) is only very slightly different from the E8.25 stage (6-7 somites) used for the *Mesp1^{Cre}* experiment (Table 1). ”

The overall structure of the paper, which includes many multi-panel figures (in the main paper and supplemental) and several supplemental tables, is confusing to read. For example, there are several parts in the text where the reader is referred to four different panels or figures within the span of a couple of sentences, with some genes being demonstrated in the main figure and others being in the supplementary figures. Moreover, the panels are not necessarily referred to in the correct order (e.g. A, B, C), and there are a couple of occasions where a results section will discuss, for example, $\frac{3}{4}$ of a figure, then the next section will discuss the remaining panels on the previous figure plus some other panels in the next figure, plus several supplementary figures/panels. The Supplementary Tables could not confidently be identified in the Suppl files due to the uninformative labelling of each Excel file. The manuscript as whole therefore needs to be restructured to make the data more accessible to the reader.

Response: We are sorry for the confusion. We took the criticism seriously and split several supplement figures for better understanding as described in the table shown below. We also now include a small legend for each and a title in each supplemental table.

Figures and Tables	
New Tables	Original Tables
Table 1 (added Mesp1Cre vs KO data; Tbx1-/- vs WT)	Table 1
Table 2 (added ATAC-seq dataset)	
Table 3 (added CHIP-seq dataset)	
New Figures	Original Figures
Figure 1 (add scalebars 1b)	Figure 1
Figure 2 (add scalebars to 2l, j, l, m)	Figure 2
Figure 3 (added RNAscope data 3g)	Figure 3
Figure 4 (new Mesp1Cre vs Tbx1Cre analysis)	
Figure 5 (Added replicate data Mesp1Cre vs Mesp1Cre cKO to c-f)	Figure 4
Figure 6 (Added replicate data Mesp1Cre vs Mesp1Cre cKO to a-c; added scalebars to g, h; removed Fig5b moved to Suppl Fig 12a)	Figure 5
Figure 7 (Added replicate data Mesp1Cre vs Mesp1Cre cKO to f-h)	Figure 6
Figure 8 (Added replicate data Mesp1Cre vs Mesp1Cre cKO to b; simplified model diagram in e). Included IGV snapshots from mm9 to be consistent with assembly used for ATAC-seq/CHIP-seq.	Figure 7
Supplement Figures	
New Supplement Figures	Original Supplement Figures
Sfigure 1 (New; breeding strategy)	
Sfigure 2	Sfigure 1
Sfigure 3 (a-f from Sfig2)	Sfigure 2 (Split; a-f)
Sfigure 4 (g-j from Sfig2)	Sfigure 2 (Split; g-j)

Sfigure 5 (Added replicate data Mesp1Cre vs Mesp1Cre cKO; a-e from Sfig3)	Sfigure 3 (Split; a-e)
Sfigure 6 (f-h from Sfig 3)	Sfigure 3 (Split; f-h)
Sfigure 7 (Added replicate data Mesp1Cre vs Mesp1Cre cKO)	Sfigure 5 (Split; a-d)
Sfigure 8 (e-h from Sfig 5)	Sfigure 5 (Split; e-h)
Sfigure 9 (wildtype versus Tbx1-/- scRNA-seq analysis at E9.5)	
Sfigure 10 (Added replicate data Mesp1Cre vs Mesp1Cre cKO; a-d from Sfig4)	Sfigure 4 (Split; a-d)
Sfigure 11 (Added replicate data Mesp1Cre vs Mesp1Cre cKO; e-i from Sfig4)	Sfigure 4 (Split; e-i)
Sfigure 12 (Added replicate data Mesp1Cre vs Mesp1Cre cK)	Sfigure 4 (Split; j-l)
Sfigure 13	Sfigure 6
Sfigure 14 (a from Sfig7; added cardiac phenotype analysis of Tbx1-Avi/Avi mice)	Sfigure 7 (Split; a)
Sfigure 15 (b-f from sfig7)	Sfigure 7 (Split; b-f)
Supplement Tables	
New Supplement Tables	Original Supplement Tables
Stable 1	Stable 1
STable 2 (Marker gene list of Mesp1Cre vs Tbx1Cre, for Figure4)	
Stable 3 (Added replicate data for Mesp1Cre Ctrl and cKO)	Stable 2
Stable 4	Stable 3
Stable 5 (Added replicate data for Mesp1Cre Ctrl and cKO)	Stable 4
Stable 6 (Added replicate data for Mesp1Cre Ctrl and cKO)	Stable 5
Stable 7 (Added replicate data for Mesp1Cre Ctrl and cKO)	Stable 6
Stable 8 (Added replicate data for Mesp1Cre Ctrl and cKO)	Stable 7
Stable 9 (Added replicate data for Mesp1Cre Ctrl and cKO)	Stable 8
Stable 10 (Added replicate data for Mesp1Cre Ctrl and cKO, combined decreased and increased)	Stable 9, Stable 10
Stable 11	Stable 11
Stable 12	Stable 12
Stable 13 (indicated 21 genes DEG, DAR, CHIP)	Stable 13
Stable 14	Stable 14
Stable 15 (now provided RNAscope probes)	

Specific comments

Methods

It is not at all clear why the Tbx1 cKO embryos analysed are a mixture of Mesp1Cre and Tbx1Cre deletions. Although the Tbx1 null phenotype is achieved with both lines, the Tbx1Cre allele (which is heterozygous for Tbx1) will almost certainly affect Tbx1 expression in the pharyngeal ectoderm and endoderm and this will change the gene expression profile compared to the Mesp1Cre allele (even non-cell autonomously). Please justify this in the results section.

Response: As explained above, we added a new figure, Figure 4 of the integration of *Mesp1^{Cre}* and *Tbx1^{Cre}* data. We show that we can identify the CPM populations with both datasets. We do not see major differences in the CPM between the different *Cre* lines. Nonetheless, we now indicate in the Discussion, that the *Tbx1^{Cre}* control might have some changes in gene expression. Once cells in common in the two were located (Fig 4), differential expression between controls and conditional mutant embryos were performed separately for the *Mesp1^{Cre}* (Fig. 5c,d) and *Tbx1^{Cre}* lineages (Fig 5e,f). We then only followed up analyses with DEGs shared in both *Mesp1^{Cre}* and *Tbx1^{Cre}* experiments. We also performed global inactivation of *Tbx1* and compared wildtype to *Tbx1^{-/-}* embryos and found similar results (Supplementary Figure 9).

Lines 425-429 in the Discussion now reads, “Although there are likely some gene expression changes in *Tbx1^{Cre}* embryos versus wildtype embryos, we obtained similar findings when we directly compared scRNA-seq results from *Mesp1^{Cre}* versus *Tbx1^{Cre}* experiments. When taken together, using both *Mesp1^{Cre}* and *Tbx1^{Cre}* alleles and investigating changes that occurred in both, allowed for a more complete analysis of the CPM with respect to *Tbx1*. We also performed a scRNA-seq experiment using wildtype versus *Tbx1^{-/-}* embryos at E9.5 and obtained comparable results.”

Lines 466-475 in the Discussion includes some of the possible non-autonomous effects of *Tbx1* inactivation and it now reads, “Some known downstream genes of *Tbx1* were not identified in the multi-omic data, such as *Wnt5a*⁶⁸, *Fgf10*^{69,70} and *Nkx2-5*, possibly due to low transcript abundance, incomplete set of TBX1 target genes from ChIP-seq, or non-autonomous functions in neighboring CPM cells.

The distal pharyngeal apparatus is hypoplastic when *Tbx1* is inactivated in the mesoderm⁷¹. This is in part because loss of *Tbx1* severely affects pharyngeal endoderm-mediated segmentation³⁶ affecting neighboring neural crest cell populations⁷¹. These functions are non-autonomous between the CPM and neural crest cells, given that *Tbx1* is not expressed in neural crest cells that contribute to OFT septation⁷². It is possible that altered signaling from affected pharyngeal endoderm cells or lack of neural crest cells could influence MLP or CPM differentiation, besides cell or tissue autonomous effects. “

The statement “Because of the difficulty to distinguish the sex at those time points, we used both male and female embryos for all the experiments” is not appropriate. Sex genotyping by PCR using Y chromosome specific primers is easily performed at the same time as routine genotyping (was this done? See comment below). The sentence should be modified to just state that embryo sex was not determined.

Response: Lines 531-532 now reads:

“We did not genotype for sex and we used both male and female embryos for all the experiments.”

“The microdissected tissues were pooled in DMEM (GIBCO, Cat# 11885-084) until all the dissections were completed.” At what temperature were the samples maintained at during the dissections?

Response: Lines 539-540 now reads: We modified the sentence to read: “The microdissected tissues were kept on ice, and pooled in DMEM (4 °C, GIBCO, Cat# 11885-084) until all the dissections were completed.”

More importantly it is not clear how many biological replicates have been analysed in

this study. Table 1 shows that multiple embryos were collected from most genotypes, but the methods section implies that the embryo material was pooled at the time of dissection. This can only be possible if the embryos do not need genotyping, but the breeding scheme describes that a wild-type *Tbx1* allele is present in the male for *Mesp1Cre* (*Mesp1Cre/+;Tbx1flox/+* male mice, crossed with *ROSA26-GFPflox/flox;Tbx1flox/flox* female mice). So the embryos in each litter will be a mix of *Tbx1+/f* and *Tbx1f/f*. This is also the case for the *Tbx1Cre* line (*ROSA26-GFPflox/flox;Tbx1flox/flox* female mice were crossed with *Tbx1Cre/+* male mice), so the embryos in each litter will be a mix of *Tbx1+/f* and *Tbx1f/Cre*. Please clarify how embryos of each genotype were collected and identified.

The breeding scheme explicitly states that *Tbx1Cre* was used for RNA-seq and ATAC-seq, but this is not stated for the *Mesp1Cre* line, although the results section says that these two lines were analysed together as the *Tbx1* cKO.

Response: We greatly apologize for not properly explaining our study design. We now show our mating strategy in Supplementary Figure 1. All scRNA-seq experiments were done on live cells taken directly from embryos. All embryos were genotyped and we did not pool embryos of different genotypes.

Does 10' mean 10 mins?

Response: We define this when we first introduce this abbreviation.

The statistical tests used should be detailed in the methods section.

Response: Statistical tests are now included in the Methods, Results and/or Legends.

Why were some experiments done at E8.0 and E8.25 but others done at E8.5?

Response: We explain this above. We tried to get the same stages, but there was slight difference in somite counts accounting for this difference. We did not directly compare the *Mesp1^{Cre}* experiments and the *Tbx1^{Cre}* experiments at these early stages.

Why was ATAC-seq / ChIP experiments only performed at E9.5?

Response: The E9.5 stage was done for all experiments (scRNA-seq, ATAC-seq and ChIP-Seq) because this is the stage that *Tbx1* expression is the strongest and when the cardiac OFT is elongating. Further, due to the great cost of these experiments, we decided to focus on one stage.

Lines 95-98 now reads, "These stages were chosen because they are the critical periods when *Tbx1* is expressed, with the highest expression at E9.5 and when the pharyngeal apparatus is dynamically elongating; this is coordinated with heart development and BrM specification."

Results

Table 1: The data shows that only one *Mesp1Cre;ROSA26-GFPflox/+;Tbx1flox/flox* was collected at E9.5. is this the reason that the cKO group contains *Tbx1*-null embryos from both *Cre* alleles? Can sufficient analysis be performed when only one biological replicate is present for the *Mesp1Cre* deletion? Is the *Tbx1Cre* Cko deletion in fact also one biological replicate? (see above). If so, how can any analysis be confidently carried out without any replicates?

Response: We now performed a replicate of the *Mesp1^{Cre}* experiment and performed scRNA-seq on wildtype versus *Tbx1^{-/-}* embryos all at E9.5. This is included in Table 1. By only using data that can be reproduced by both *Cre* lines and now, the wildtype/*Tbx1^{-/-}* embryos; there are multiple cross-checks in this study. We validated important findings in the embryo by immunofluorescence and RNAscope analysis. We know the number of replicates for scRNA-

seq analysis remains an open issue; another reason we did not emphasize the cell proportion changes in the revised manuscript.

Also, why were more cells submitted and captured from this one sample (13,788 and 5,157) when only 12,000 and 4,055 were acquired from six *Mesp1Cre*; ROSA26-GFPflox/+ embryos? Further analysis includes comparing *Mesp1/Tbx1* cKO at E9.5 with only the *Tbx1* cKO at E8.5. As these are two different models can they be directly compared?

Response: We used a 10x Genomics platform for scRNA-seq library preparation. According to their instructions, the captured cell number depend on cell viability and cell states. And captured cell rate is around 60%. So, we couldn't control how many cells we could capture until sequencing. To get most cells, we applied up to 10,000 cells for each sample. We used the DEGs that were shared but computed independently between the 2 *Cre* control vs mutant datasets at E9.5. We didn't compare *Mesp1^{Cre}* mediated *Tbx1* cKO embryos at 9.5 with *Tbx1^{Cre}* mediated *Tbx1* cKO embryos at E8.5. Our writing has been clarified as explained above.

Figure 1 shows data analysis from the four developmental stages selected, where mesoderm has been FACS as GFP labelled cells from the *Mesp1Cre* allele. On page 4 it states "To better understand the developmental connection, we integrated the four time point datasets..." but it is not clear how this developmental connection is being shown. Clearly there is a lot of anatomical changes occurring between E8.5 and E10.5 in pharyngeal arch development. The concluding sentence states "that CPM progenitors can be distinguished from more mature CPM states by their multilineage primed gene signatures" but it is not clear from Figure 1 how this was determined from a mixture of embryonic stages. Overall this section therefore needs some clarification and more explanation as to why this experiment was done and what it actually shows.

Response: We agree that Figure 1 doesn't show the developmental connection and the conclusion is an overreach at this point in the manuscript. The conclusion only comes after all the data is presented in the manuscript. We therefore removed the last concluding sentence. Also see above for reasons of performing integrated scRNA-seq analysis.

Lines 115-116 now reads, "Based upon this, we suggest that C15 as a multilineage primed progenitor (MLP) population within the CPM."

Figure 1B: include scale bars.

Response: These are now included.

Figure 1F. Consider indicating the CPM, MLP and BrM genes within the figure for clarity.

Response: These are now included.

It would also be useful to clarify in the main figure which genes are also in other groups e.g. C3, C1, C18 to reduce jumping between figures and text.

Response: These are now included.

Page 6. The 4th pharyngeal arch is mentioned twice in this section but is not labelled on the figure panels. This will help with identifying the structure being referred to.

Response: pa4 is now included.

The term "newly appreciated" for *Aplnr* and *Nrg1* should be reconsidered as these genes are already known and described.

Response: **Lines 132-137** now reads: "Based on specific expression of genes (Supplementary Table 1) and distribution of gene expression patterns in the PAGA plot (Fig. 2f, Supplementary

Fig. 3d-f), we focused on two marker genes for MLPs, *Aplnr* (Apelin receptor) and *Nrg1* (Neuregulin 1). *Aplnr* is expressed in the CPM²⁸ but not known for specific presence in MLPs, while *Nrg1* is not known to be a CPM gene, but it is required in the embryonic heart for the development of the chamber myocardium²⁹.”

Figure 2A legend. Do you mean that cluster numbers are consistent with 1D? Figure legend should be stand-alone – define clusters in legend again instead of referring to Figure 1.

Response: These are now included.

Please summarise in the text what the 6 main branches are.

Response: There are only 5 branches, we apologize.

Lines 124-126 now reads, “The five branches include cardiomyocytes (CMs), pSHF with lung progenitor cells (Lung PC), connective tissue (CT), branchiomeric muscle progenitor cells (BrM) and skeleton/limb (Sk/L).”

Figure 2B and 2C are referred to, alongside Supplementary Figure 2B and 2A. The supplementary panels should be switched around so that the authors are referring to the appropriate panels.

Response: This is now fixed in the manuscript.

Figure 2I/J/K/L/M. Include scale bar on all images

Response: This is now fixed in the manuscript.

Figure 2H-M. Why is E8.5 used in this figure while E8.0/E8.25 was used in previous figures? Please clarify why time points are not the same for each experiment.

Response: This is described above.

Figure 2H/I/J. What line/genotype are the WT's from?

Response: This is explained in the figure legend. Wildtype embryos are used in 2h-j, while *Mesp1^{Cre};ROSA26-GFP^{fllox/+}* embryos are used in 2k-m.

Heat map in Figure 3B and Figure 3C don't match up with the description of stages in the legend. B legend and text says E8, E8.25 while C legend says E9.5, E10.5, but all 4 time points are illustrated in the figure. Please correct this. Also clarify colour code in the figure legend for E8-10.5.

Response: This is now fixed. Lines 883-886 now reads,

“b. Heatmap of expression of the genes enriched in expression in earlier stage-MLPs (E8, E8.25) and shown in all four stages. Row indicates the expression of each cell.

c. Heatmap of expression of the genes enriched in expression in intermediate (E9.5) and later stage MLPs (E10.5) and shown in all four stages. Row indicates the expression in each cell.”

Please refer to panels in order – 3A&D, B&E, C&F should be moved around so that they are referred to appropriately within the text.

Response: We apologize for this mistake. This was fixed in the revised text.

Lines 163-169 now reads, “An important question is whether MLPs as CPM progenitors, maintain the same state of gene expression over time. To address this, we examined differentially expressed genes in MLPs from E8-10.5. We identified core CPM genes that are expressed similarly at all time points, including *Isl1*, *Mef2c* and *Nkx2-5* (Fig. 3a). However, we also found that early expressing genes such as *Aplnr*, *Nrg1*, *Irx1-5*, *Fgf8/10* and *Tbx1* (Fig. 3b) are reduced over time, with increasing expression of cardiac developmental genes such as

Hand2, *Gata3/5/6*, *Bmp4* and *Sema3c* (Fig. 3c). These gradient differences are also visualized in violin plots (Fig. 3d-f). “

Figure 4A/B: add scale bars. Why are the experiments done in the *Mesp1Cre* line only at E9.5 but at E8.5 and E9.5 in the *Tbx1Cre* line? Why was the E8.5 used here instead of E8.0 and E8.25?

Response: Scale bars are now added. Please see the issue with stages, explained above.

Is there a statistical test to compare CTR/KO at E8.5 and E9.5?

Response: We removed the bar graphs in Fig. 4 because after re-analyzing the data, we are not confident in the method we used to create them and found variation between the 2 *Mesp1^{Cre}* replicates and between the two stages of *Tbx1^{Cre}* experiments. Please see response above.

This is another example of not referring to panels in order – please move around so that panels are referred to in the correct order in the text.

Response: We fixed the writing so the panels are referred to in the correct order.

Figure 5: Please clarify why the *Mesp1Cre* line was used as the control when both *Mesp1Cre* and *Tbx1Cre* cKO lines were used.

Response: Fig. 5 is now Fig. 6. We show this in a venn diagram in Fig 6a to explain our strategy. We analyzed *Mesp1^{Cre}; Tbx1^{+/+}* control vs *Mesp1^{Cre}; Tbx1^{ff}* cKO and *Tbx1^{Cre/+}* vs *Tbx1^{Cre/ff}* cKO embryos, respectively. We generated 2 independent sets of DEGs. Then we compare each DEG list. We excluded DEGs found in only 1 dataset.

Fig 5G, H. Panels should specify which Cre mutant is being shown. Include scale bars on all images.

Response: We added labels in the Figures and scale bars on all images.

The authors say in the text that the affected genes are involved in cell differentiation or cell signalling. From the GO terms there is muscle cell differentiation but presumably a lot of genes will be involved in cell signalling – what signalling are they involved in?

Response: With the 2 replicates of the *Mesp1^{Cre}* experiment, we updated Fig. 6c - d, with the corresponding text.

When referring to Figure 5 in the text, the authors refer to *Tbx1* cKO (for the RNAScope staining) instead of clarifying whether the embryos are *Mesp1-Cre* or *Tbx1-Cre*. It would be useful to refer to each one specifically in the text, and to indicate this on the figure panels.

Response: Thank you for your suggestion. We added the *Cre* information in the Figure.

Figure 6: What do the grey regions in the pie charts represent?

Response: The grey region indicates CARS or DARs didn't find in *Mesp1^{Cre}* ATAC-seq dataset and they were therefore filtered out, based upon our strategy outlined in Fig 6a.

Page 10. It is not clear which genotypes were analysed for ATAC-seq and ChIP-seq experiments, only “control versus *Tbx1* mutant embryos” is stated. Figure 6A shows that only *Tbx1Cre* cKO embryos were used for ATAC-seq but this should be confirmed in the text. Why was only this line used whereas the *Mesp1* cKO was also used in the RNA-seq analysis?

Response: We now provide our strategy in Fig. 6a, which we explain above (please see above). Original Fig. 6 is now Fig. 7.

Lines 286-289 now reads, “To better understand how TBX1 regulates expression of genes in the CPM at the chromatin level, we used two biological replicates of ATAC-seq experiments of *Tbx1*^{Cre/+} (*Tbx1* Ctrl) versus *Tbx1*^{Cre/f} (*Tbx1* cKO) mutant embryos (Supplementary Fig. 1c, d; Fig. 7a, Supplementary Fig. 13a-c).”

In this section replicates are mentioned. How were these prepared? as it is not mentioned in the ATAC-seq methods section. For CHIP-seq it states that the Tbx13'-Avi;BirA embryos were collected and 20 pooled for one sample. How many pools/samples were used in the experiment? For the RNA-seq embryos a detailed table was provided. There is no equivalent details for the other experiments.

Response: We provide Tables 2 and 3 in the revised manuscript for details of the ATAC-seq and CHIP-seq experiments. For CHIP-seq, we used 20 embryos for each replicate, for three biological replicates.

Lines 591-593 now reads, “After the tissues were washed with PBS, and centrifuged (4 °C, 200 x g, 5'), the pellets were frozen in dry ice and stored at -80 °C. We used 20 embryos for one sample and performed three biological replicates. “

Figure 7: Why were Tbx1 hets assessed separately here when a previous results section stated that the hets and WTs were pooled?

Response: For ATAC-seq, we compared *Tbx1*^{Cre/+} versus *Tbx1*^{Cre/f} embryos. This is now shown in the strategy in Fig. 7a.

Discussion

First sentence is very vague. Please re-write so it puts the study results in context.

Response: Thank you for your comment. We removed the first sentence. The first sentence in the discussion now states, “In this report, we uncovered a new progenitor population within the CPM that we term MLP. “

The model proposed in Fig7F shows that Tbx1 deficiency in the CPM causes caudal pharyngeal hypoplasia and the developmental consequences of this. As the caudal arches do not form properly it is important to know that this change in tissue architecture is not the reason for the differential gene expression in cKO embryos compared to controls with normal arch morphogenesis.

Response: Original Fig. 7 is now Fig. 8. We have now modified Fig. 8f to reflect all our re-analyses, and have clarified the writing. Further, we also added the following

Lines 466-475 in the Discussion includes some of the possible non-autonomous effects of *Tbx1* inactivation and it now reads, “Some known downstream genes of *Tbx1* were not identified in the multi-omic data, such as *Wnt5a*⁶⁸, *Fgf10*^{69,70} and *Nkx2-5*, possibly due to low transcript abundance, incomplete set of TBX1 target genes from CHIP-seq, or non-autonomous functions in neighboring CPM cells.

The distal pharyngeal apparatus is hypoplastic when *Tbx1* is inactivated in the mesoderm⁷¹ This is in part because loss of *Tbx1* severely affects pharyngeal endoderm-mediated segmentation³⁶ affecting neighboring neural crest cell populations⁷¹. These functions are non-autonomous between the CPM and neural crest cells, given that *Tbx1* is not expressed in neural crest cells that contribute to OFT septation⁷². It is possible that altered signaling from affected pharyngeal endoderm cells or lack of neural crest cells could influence MLP or CPM differentiation, besides cell or tissue autonomous effects. ”

Can the authors explain how the *Tbx1*-null mouse model relates to the gene regulation events in 22q11DS patients, who are hemizygous for *TBX1*? These patients show a range of clinical phenotypes despite all lacking one allele of *TBX1*. How does your data fit with arch artery defects such as IAA?

Response: This is a great question. The mouse is a good but not a perfect model for humans. The *Mesp1^{Cre};Tbx1^{fl/fl}*, *Tbx1^{Cre/fl}* and *Tbx1^{-/-}* embryos have a persistent truncus arteriosus in all genetic backgrounds. *Tbx1^{Cre/+}* or *Tbx1^{+/-}* mice have 4th pharyngeal arch artery related defects at reduced penetrance when maintained in C57Bl/6. Genetic interaction studies of other genes with *Tbx1* heterozygous mice can produce interrupted aortic arch type B. The experimental evidence points to the endoderm and ectoderm domains of *Tbx1* being important for 4th pharyngeal arch artery defects and not the mesoderm. *TBX1* haploinsufficiency in the 22q11.2DS patients is more sensitive because >25 other genes are also present in one copy. Our data doesn't directly include the endoderm and ectoderm functions of *Tbx1*.

Reviewer #3 (Remarks to the Author):

Review of Nomaru et al.

Summary

The work presented in this manuscript defines a population of *Mesp1*-derived multilineage progenitor cells in the cardiopharyngeal mesoderm (CPM) that give rise to second heart field (SHF) and branchiomeric muscle (BrM), and attempts to ascribe a specific functional role to *Tbx1* within this population. The work builds on clonal lineage studies and *Tbx1* loss of function studies from several groups that demonstrated that SHF cells and BrM cells have a common progenitor in the CPM and that *Tbx1* is required for CPM to differentiate into SHF and BrM. The novelty of the present study is using clustering analysis on scRNA-seq from *Mesp1*-derived CPM cells to molecularly and spatially define a specific progenitor population at several timepoints during CPM expansion and differentiation that is dubbed a "multilineage primed progenitor" (MLP). MLPs express both BrM and SHF genes and are located bilaterally in the lateral part of the caudal pharyngeal apparatus. To define the role of

Tbx1 in this population, scRNA-seq was performed in control and conditional *Tbx1* knockout embryos using *Tbx1*-Cre and *Mesp1*-Cre driver lines. In the absence of *Tbx1*, both models demonstrated that MLPs were present, but failed to differentiate as efficiently towards SHF and BrM fates, leading to MLP accumulation. Additionally, within the MLP population, genes involved in cell differentiation and signaling decreased, and expression of some genes associated with non-mesodermal lineages was observed. Specifically, *Aplnr* (MLP gene) was downregulated while *Pax8* (non-mesodermal gene) was upregulated. To identify target genes of *Tbx1* in the MLP, the authors use ATAC-seq in control and conditional KO embryos, as well as *Tbx1* ChIP-seq using a novel Avi-tagged knock-in *Tbx1* allele. The authors define a set of differentially accessible chromatin regions (DARs) between control and knockout embryos, the vast majority of which demonstrated reduced accessibility in the knockout. Intersecting ATAC-seq DARs with MLP-enriched genes and *Tbx1*-bound regions identified with ChIP-seq defined 8 putative direct *Tbx1* targets in the MLP, including *Aplnr* and *Nrg1* as well as *Isl1* and other genes. Based on the results of these experiments, the authors advance the following conclusions:

1) A multilineage primed progenitor (MLP) population in the CPM can be molecularly identified through co-expression of *Aplnr*, *Nrg1*, and *Tbx1*.

- 2) MLPs are maintained as a source of progenitor cells in the CPM, deploying cells to the heart and branchiomeric muscles during development.
- 3) *Tbx1* promotes progression of MLPs to more differentiated cell states by direct and indirect regulation of a set of defined transcriptional targets.

General Comments

This manuscript addresses an important problem using a powerful combination of computational and genetic tools, and along the way contributes several new and useful datasets to the field. The datasets are generally consistent with one another and mutually reinforcing, with some minor exceptions noted below. The work may serve as a springboard for future studies on more detailed mechanisms that regulate differentiation of the MLP population.

I do have some concerns, however, about how some of the experiments are interpreted, and whether alternative explanations for the findings presented are adequately considered or excluded. While the data do support qualified versions of the conclusions enumerated above, the manuscript could be strengthened by acknowledging areas where the data are insufficient to support broad conclusions and also by more clearly delineating how this work adds to the body of literature already published on the role of *Tbx1* in the CPM.

Specific Comments

1. In the introduction, the authors state, “Although there are many studies of *Tbx1*, we do not yet understand its functions on a single cell level, which is needed to elucidate the true molecular pathogenesis of 22q11.2DS” Since the role of *Tbx1* in the CPM has been explored previously and it is known that *Tbx1* is required for differentiation from CPM to BrM and SHF, a clearer statement in the introduction about what has been established/hypothesized about the role of *Tbx1* in the CPM, along with a clearer statement of why scRNA-seq is needed to fill these knowledge gaps would be very helpful for readers who may have less familiarity with the literature on CPM.

Response: We have now provided relevant background and more of an explanation and rationale for the work presented in the manuscript.

Lines 63-68 in the Introduction now reads,

“Gene expression profiling of wildtype versus *Tbx1* global null mutant embryos identified genes that changed in expression but it was unclear whether the changes were autonomous in the CPM or in other cell populations, such as neural crest cells^{20,21}. We therefore do not yet understand its functions of *Tbx1* on a single cell level, which is needed to elucidate the true molecular pathogenesis of 22q11.2DS. ”

Similarly, a more thorough discussion of the previous literature on *Tbx1* function in the CPM in the Discussion would be helpful to better highlight the specific ways in which the present work advances the field.

Response: We agree with this comment and now have added a more thoughtful discussion of the results with respect to the previous literature on *Tbx1*.

This is included in lines 363-414. It reads:

“ Previous work showed that the aSHF, pSHF and BrM cells comprising the CPM, derive from a relatively small number of *Mesp1* expressing progenitor cells at gastrulation⁸. Based upon the work presented in this report, it is most likely that not all cells have committed to final CPM fates at gastrulation and MLPs provide a source of progenitors as the pharyngeal arches form. Retrospective clonal analysis has shown that there is a direct clonal relationship between

progenitor cells that form the muscles of mastication and right ventricle, which are derived from the first pharyngeal arch, with distinct clones that forms both the OFT and facial expression muscles, from the second arch, while separate clones form the neck muscles and venous pole of the caudal pharyngeal arches, 3-6^{9,31}. This suggests that different clones contribute to different arches. It is possible that MLPs are comprised of heterogeneous progenitor cells that are allotted to individual arches and/or that they are exposed to different extracellular signals during development, conferring pharyngeal arch specific fates. The heart tube elongates from E8-10.5 by deployment of progenitor cells to the OFT. It is known that cells deployed to the cardiac poles first arrive at the dorsal pericardial wall (DPW), just behind the heart tube⁵⁰. Mesodermal cells lateral and behind the DPW are thought to be incorporated to the DPW and then to the poles of the heart⁵¹. The deployment of mesoderm cells to the DPW provides a pushing force as the epithelial transitioned cells move to the poles of the heart^{51,52}. In *Tbx1* null mutant embryos, there are fewer cells in the DPW resulting in a shortened cardiac OFT^{35,43,53}. We propose that MLPs comprise the dorsal population of mesoderm progenitor cells that are needed to allocate cells to the DPW. This is consistent with reduction of *Wnt5a* expression in our data, as *Wnt5a* is a key downstream gene of *Tbx1*, required for their deployment^{51,52}.

An anterior-posterior border is established in the DPW between *Tbx1* and *Tbx5* expressing cells, respectively that provides cells to the poles of the heart³⁴. Consistent with this, we found that there are few *Tbx5* expressing cells in the *Tbx1* lineage. Understanding the molecular mechanisms of how the anterior-posterior border is established is an active area of research^{5,34}. We previously found that global inactivation of *Tbx1* results in increased expression of caudal pSHF genes such as *Tbx5* and cardiac muscle genes^{20,54}. Data presented in this report and recently³⁴, indicate that rather than changes in expression, there are instead cell population changes in *Tbx1* mutant embryos. Therefore, this scRNA-seq study discerns better between expression versus population changes depending on *Tbx1*, which is often a key challenge in interpreting developmental phenotypes. More work needs to be done in the future to better understand how these borders are formed and maintained.

In addition to deploying cells to the DPW and then to the heart, the MLPs express genes required for BrM formation in each arch. The BrMs form segmentally in a rostral to caudal manner, in which the muscles of mastication form first from the first arch and the other muscles of the face and neck form thereafter from more caudal arches. The BrMs express myogenic regulatory transcription factors including *Tcf21*, *Msc*, *Myf5* and later *Myod1*⁵⁵. In addition, transcription factor genes such as *Isl1*¹¹, *Pitx2*⁵⁶, *Tbx1*¹⁹, *Lhx2*²³ and *Ebf* genes (*Ciona*⁵⁷) are expressed in the CPM and are required for BrM formation. A subset of MLPs expressing *Tbx1* will later express BrM genes as they migrate to the core of the pharyngeal arches. These cells progressively express myogenic transcription factor genes as they become restricted to form BrM skeletal muscle cells. Besides the MLPs, *Tbx1* is also expressed in BrM progenitor cells, and therefore, some of the gene expression changes we observed in these cells might be due to *Tbx1* expression in the BrM progenitor cells themselves.

Although we focus on the CPM as it relates to cardiac and skeletal muscle development, it was shown that the CPM also contributes to mesenchyme of connective tissue including cartilage in the neck⁵⁸. Thus, it is possible that the MLPs could promote connective tissue fates in the craniofacial region that are dependent on *Tbx1*. Further work will be needed to assess their lineage relationships. Given that most patients with 22q11.2DS have craniofacial malformations in the face and neck, it is important to understand the developmental trajectories of *Tbx1*-dependent connective tissue progenitor cells.”

2. The existence of a multipotent progenitor in the CPM has been established with clonal fate mapping in prior studies, but the molecular identity of that population has not been established. In the present work, the MLP is defined through a clustering and pseudotime/PAGA analysis rather than by clonal fate mapping. The authors provide

several lines of evidence that support identifying the MLP with the multipotent progenitor defined in prior work, but without clonal fate mapping of the MLP cells in vivo, one cannot conclude that individual MLP cells are actually multipotent. Therefore, a statement in the Discussion clarifying this point would be helpful and would limit the potential for over-interpretation.

Response: We agree and now provide the following on lines 363-374 in the Discussion, “Previous work showed that the aSHF, pSHF and BrM cells comprising the CPM, derive from a relatively small number of *Mesp1* expressing progenitor cells at gastrulation⁸. Based upon the work presented in this report, it is most likely that not all cells have committed to final CPM fates at gastrulation and MLPs provide a source of progenitors as the pharyngeal arches form. Retrospective clonal analysis has shown that there is a direct clonal relationship between progenitor cells that form the muscles of mastication and right ventricle, which are derived from the first pharyngeal arch, with distinct clones that forms both the OFT and facial expression muscles, from the second arch, while separate clones form the neck muscles and venous pole of the caudal pharyngeal arches, 3-6^{9,31}. This suggests that different clones contribute to different arches. It is possible that MLPs are comprised of heterogenous progenitor cells that are allotted to individual arches and/or that they are exposed to different extracellular signals during development, conferring pharyngeal arch specific fates. ”

3. Related to the latter point, can the authors comment on the presence of the MLPs at E10.5? Given what is known about branchial arch development and timing of addition of cells to the SHF, is it expected to see MLPs this late in development or is it likely that these cells are already specified to either BRM or SHF or other MLP derivatives?

Response: This is similar to the concern by Reviewer 1. As we state above, we agree that there are differences at E10.5. This is now included in the Results.

We now updated Figure 3 with RNAscope data (new Fig. 3g). We performed RNAscope analysis of *Aplnr* (early gene), *Sema3c* (late gene), *Tbx1* (early gene), *Nkx2-5* (early and late) and *Isl1* (early and late) at E9.5 and E10.5.

Lines 163-176 now state, “An important question is whether MLPs as CPM progenitors, maintain the same state of gene expression over time. To address this, we examined differentially expressed genes in MLPs from E8-10.5. We identified core CPM genes that are expressed similarly at all time points, including *Isl1*, *Mef2c* and *Nkx2-5* (Fig. 3a). However, we also found that early expressing genes such as *Aplnr*, *Nrg1*, *Irx1-5*, *Fgf8/10* and *Tbx1* (Fig. 3b) are reduced over time, with increasing expression of cardiac developmental genes such as *Hand2*, *Gata3/5/6*, *Bmp4* and *Sema3c* (Fig. 3c). These gradient differences are better visualized in violin plots as well (Fig. 3d-f). *Nkx2-5* and *Sema3c* are expressed in the caudal pharyngeal apparatus at E9.5, like that of *Isl1*, *Tbx1* and *Aplnr*, defining MLPs (Fig. 3g). The MLP region is reduced in size in the caudal pharyngeal apparatus, pharyngeal arches 3-6, at E10.5 (Fig. 3g). Further, expression of the early MLP marker *Aplnr* is not observed at E10.5, while co-expression of *Sema3c*, *Nkx2-5* and *Isl1* occur strongly in the OFT (Fig. 3g). This is consistent with the model that the MLPs continuously allocate progenitor cells to BrMs and OFT-CMs, while showing a gradual maturation themselves by E10.5 (Fig. 3h). “

4. A strength of the work is the authors’ use of 2 genetic models to knock out *Tbx1* with comparison of the results. However, since *Tbx1* is expressed in other *Mesp1+* populations beyond the MLP, it’s possible that some of the observed changes in percentages of different cell populations as well as some of the gene expression changes within cell populations, are non-cell-autonomous, and this possibility should be addressed when discussing the results of *Tbx1* knockout experiments. The authors

conclude that *Tbx1* plays a specific role in MLP though regulation of defined targets but cannot exclude the possibility that *Tbx1* plays a role in other cell types that is relevant to what happens to the MLP.

Response: We agree completely with this comment.

Lines 466-475 in the Discussion now reads,

“Some known downstream genes of *Tbx1* were not identified in the multi-omic data, such as *Wnt5a*⁶⁸, *Fgf10*^{69,70} and *Nkx2-5*, possibly due to low transcript abundance, incomplete set of TBX1 target genes from ChIP-seq, or non-autonomous functions in neighboring CPM cells.

The distal pharyngeal apparatus is hypoplastic when *Tbx1* is inactivated in the mesoderm⁷¹. This is in part because loss of *Tbx1* severely affects pharyngeal endoderm-mediated segmentation³⁶ affecting neighboring neural crest cell populations⁷¹. These functions are non-autonomous between the CPM and neural crest cells, given that *Tbx1* is not expressed in neural crest cells that contribute to OFT septation⁷². It is possible that altered signaling from affected pharyngeal endoderm cells or lack of neural crest cells could influence MLP or CPM differentiation, besides cell or tissue autonomous effects. “

5. *Tbx13*'-Avi mouse line. Since these mice survive and breed as homozygotes, the presumption is that there is no deleterious phenotype associated with the knock-in and that this allele is not hypomorphic, but it would be ideal to state this explicitly as it affects interpretation of the ChIP-seq data.

Response: This data is now provided in Supplementary Figure 14.

Related to this, can the authors comment on the relatively small number of ChIP-seq peaks identified despite a very large number of DARs in the knockout with reduced chromatin accessibility? Is this because there are actually relatively few binding sites for *Tbx1* and many effects on chromatin are indirect, because of low expression of *Tbx1* in this population, low cell numbers for the experiment, or because *Tbx1* might have other functions that are not related to DNA binding as has been observed by other researchers?

Response: We think the answer is a combination of the possibilities that this Reviewer raises. Only a certain percentage of DARs have putative T-sites (Fig. 8a). Many DEGs do not have DARs (Fig. 7f), so regulation can be indirect or TBX1 binding doesn't always affect chromatin accessibility, for example, if it brings a co-factor to an existing open chromatin region. Further, TBX1 protein can regulate serum response factor, SRF, protein level without changing RNA, thus there are possible other functions of TBX1 that do not involve binding to DNA. On a separate note, we had technical difficulty obtaining enough tissue for our TBX1 ChIP-seq experiments to saturate the possible direct binding sites. In the future, we will microdissect tissue and repeat the ChIP-seq.

Lines 434-445 in the Discussion now reads,

“The TBX1 ChIP-seq provided hundreds of direct target genes, of which some are also reduced in expression in mutant embryos and show a change in chromatin accessibility. One note is that we identified hundreds rather than thousands of direct transcriptional targets that were expected based upon studies of other transcription factors. This could be because we used whole embryos for the ChIP-seq with lower tissue yield than a microdissection. Nonetheless, the ChIP-seq data supports the ATAC-seq findings. Further, this work shows that both direct and indirect regulation occurs downstream from TBX1 because not all differentially accessible sites have T-sites and not all differentially expressed genes have either differentially accessible sites or ChIP-seq peaks. Additionally, TBX1 protein can regulate the protein level of serum response factor, SRF without changing the expression level of RNA, thus there are possible

other functions of TBX1 that do not involve binding to DNA²². “

6. The statement in the results (p.10), “We suggest that Tbx1 provides a balance of specific gene expression required for MLP function” is vague and should ideally be rephrased since it’s not clear what is being “balanced”.

Response: Lines 202-203 now reads, “*Tbx1 regulates MLP development by promoting gene expression needed for differentiation and restricting expression of non-mesodermal genes*”

7. The authors define 8 direct transcriptional targets of interest for Tbx1 in the MLP and present ChIP-seq browser tracks for 2. While the peak for Nrg1 overlaps a Tbx1-dependent ATAC-seq peak the Aplnr peak appears to bind closed chromatin that is far from the DAR, making the mechanism whereby this Tbx1 binding event activates Aplnr in the MLP somewhat unclear. Can the authors identify any other datasets (e.g., ENCODE or other embryonic dataset) that suggest the presence of regulatory element at this site?

Response: Since we incorporated the 2 *Mesp1*^{Cre} replicates, we now have 21 genes that are direct transcriptional targets of interest. Of them, 8 have the DAR and TBX1 ChIP-seq peak that is co-localized. We analyzed ENCODE data to suggest the presence of regulatory elements. We modified the following:

Lines 326-335, in the Results section now reads,

“We then intersected the DEGs reduced in MLPs, the annotated genes from *Mesp1*-DARs and TBX1 ChIP-seq targets (Fig. 8b). We found 21 known genes (Fig. 8b; Supplementary Table 14) common in all three datasets ($P < 0.001$; permutation test). Among them, eight had a DAR that overlapped with a TBX1 ChIP-seq peak (*Slit1*-intron, *Crybg3*-intron, *Nrg1*-upstream, *Trps1*-downstream, *Sox9*-upstream, *Trmt9b*-downstream, *Fn1*-upstream and *Crtc2*-promoter region). Data is consistent with TBX1 binding to accessible chromatin in control embryos, but the interval is not accessible when *Tbx1* is inactivated. The rest had a DAR that did not overlap with a TBX1 binding site (*Fgf1*, *Aplnr*, *Tshz3*, *Rcsd1*, *B3galnt12*, *Ppml1*, *Spon1*, *Mpped2*, *Tbx18*, *Daam1*, *Parvb*); perhaps regulation is by long-range chromatin-interaction with TBX1 binding. ”

Lines 337-345 in the Results section now reads,

“We examined the ENCODE ChIP-seq tracks in the UCSC genome browser tracks⁴⁹. For *Nrg1*, the co-localized peak is within an ENCODE cis-regulatory element (cCRE) that is a poised enhancer in mouse embryonic heart⁴⁹. For the *Aplnr* gene region, the TBX1 binding site that is just downstream of the 3’UTR (Fig. 8d), is in an ENCODE cCRE that is a poised enhancer in mouse embryonic heart, but it was not in a DAR found in our data. The DAR that was identified in the *Aplnr* locus, is in an ENCODE cCRE (E0701748/enhD) and is an ATAC-seq peak region in the embryonic heart⁴⁹. Overall, the regions found appear to be regulatory regions, but TBX1 might not always affect chromatin accessibility, indicating that multiple mechanisms of regulation occur. ”

8. Previous work defined a role for Tbx1 in regulating epithelial properties of the SHF. Were DEGs related to cell polarity or tissue morphogenesis observed in the knockout or as direct targets from the ChIP-seq?

Response: That is a very interesting question. For us, we noticed several that could be related to cell polarity, such as *Wnt5a*, *Daam1*, etc. Many of these genes have multiple other functions, we should pursue how TBX1 regulates the epithelial properties in the DPW.

9. The Discussion states, “Interestingly, both Aplnr and Nrg1 are direct target genes of TBX1 based on our ATAC-seq and ChIP-seq results, suggesting that these genes are

mediators of TBX1 function in MLPs.” The data presented don’t define any specific role for *Aplnr* and *Nrg1* in the MLP population beyond their expression as marker genes, so this statement is overinterpretation.

Response: We made a very serious attempt to determine the mechanism of *Aplnr* and *Nrg1* function in MLPs, but did not succeed. We obtained *AplnrCreERT2* mice from Dr. Red-Horse to test if the *Aplnr* lineage can migrate to the cardiac outflow tract. We noticed that *Aplnr* is also expressed in the dorsal pericardial wall harboring cells after they have left the MLP and migrating to the outflow tract. Therefore, we realized that even if we can find that the *Aplnr* lineage can migrate to the heart from the MLPs, expression in the dorsal pericardial wall, where *Tbx1* is also expressed, would make it difficult to interpret results, therefore we decided not to include these findings. We obtained *Nrg1* floxed mice. Unfortunately, the mice we received was a floxed allele in which loxP sites flanked the IG domain that would not result in a complete null. We crossed the mice using *Tbx1^{Cre}*, but we didn’t find an abnormal phenotype. We obtained sperm from a different *Nrg1* floxed allele that should result in a complete null after recombination, however, none of the founders nor offspring from IVF have the floxed allele.

We have tempered the conclusion and this sentence on lines 460-462 now reads, “Interestingly, both *Aplnr* and *Nrg1* are direct target genes of TBX1 based on our ATAC-seq and ChIP-seq results, however, more work will need to be done to know whether these genes have functional importance in MLPs or in relation to *Tbx1*.”

10. Venn diagrams showing overlap of different gene sets between different experiments (e.g, Figures 6F and 7B) look highly significant but should still have accompanying chi squared tests and P values presented alongside the data.

Response: We added chi square test results to the legend for the Venn diagrams in Fig. 7f and 8b (original 6f and 7b).

Reviewers' Comments:

Reviewer #1:

Remarks to the Author:

The authors have done a good job of answering most of my concerns. The most important concern, however, they have left unanswered and uncommented. I mentioned the caveat that they may be carrying modifiers from the original background(s) onto SwissWebster and asked for them to perform whole genome SNP genotyping on a few representative animals of their N15 backcrossed colony. They did not do this, nor did they explain why not. Mouse SNP genotyping is quick and relatively inexpensive.

I would like to see a supplemental figure delineating the size of the congenic regions around each of the genetic modifications brought onto SwissWebster.

Reviewer #2:

Remarks to the Author:

The authors have made a thorough effort to address our concerns and have produced a greatly improved manuscript with much more clarity. The performing of replicate experiments adds more confidence to the results and conclusions. The rearranging of figure panels to match the order described in the text is very helpful for following the results, as is the addition of significant amounts of explanation in the discussion to support their data.

Reviewer #3:

Remarks to the Author:

For this resubmission, the authors have performed new RNA-seq experiments that have resulted in revision of a significant conclusion (that the MLP expands in the absence of Tbx1), they have attempted to clarify the role of Tbx1 downstream target genes *Aplnr* and *Nrg* without success, and they have made numerous textual revisions to add context, clarify the rationale for specific experiments, and avoid over-interpretation of their findings. Overall, I continue to feel that the manuscript adds important new data to the literature, and my specific concerns -- which were primarily related to interpretation of data -- have all been addressed.

Response to review of NCOMMS-21-00939A.

Reviewer 1 comment: The authors have done a good job of answering most of my concerns. The most important concern, however, they have left unanswered and uncommented. I mentioned the caveat that they may be carrying modifiers from the original background(s) onto SwissWebster and asked for them to perform whole genome SNP genotyping on a few representative animals of their N15 backcrossed colony. They did not do this, nor did they explain why not. Mouse SNP genotyping is quick and relatively inexpensive.

I would like to see a supplemental figure delineating the size of the congenic regions around each of the genetic modifications brought onto SwissWebster.

Response:

We want to thank Reviewer 1 for bringing up very insightful and thoughtful points. Reviewer 1 requests genome-wide SNP genotyping to map the congenic breakpoints around the *Tbx1* locus that is retained in the 129 strain background as maintained in SwissWebster. This is because of strain dependent variation in the *Tbx1* heterozygous phenotype. We have thought about genetic modifiers for many years. Further, we respect the opinion of Reviewer 1 and thought about whether we can perform genome-wide SNP arrays and whether we can use the data to identify such modifiers. However, here are 5 points we want to bring up to explain why we do not think adding this experiment is necessary for the manuscript.

1) The manuscript uses the null phenotype to draw conclusions about the role of *Tbx1* in development, not the heterozygous (haploinsufficient) phenotype. The null phenotype, as two decades of publications indicate, is remarkably similar across multiple genetic backgrounds. Therefore, the genetic background-dependence of the heterozygous phenotype would not affect the conclusions of the manuscript.

2) The effect of genetic background upon the major haploinsufficient phenotype (pharyngeal arch artery defects) has been known for many years and it appears to concern the penetrance of the defects, more than its expressivity. Early work have noted that the original 129 strain (within which the mutations were generated) has a suppressive effect upon penetrance (similarly to FVB or SwissWebster), while the C57Bl/6 background has an enhancing effect, and have excluded that this modifying effect is due to the genomic region surrounding the mutation (Taddei I, Morishima M, Huynh T, Lindsay EA. Genetic factors are major determinants of phenotypic variability in a mouse model of the DiGeorge/del22q11 syndromes. Proc Natl Acad Sci U S A. 2001 Sep 25;98(20):11428-31. doi: 10.1073/pnas.201127298. Epub 2001 Sep 18. PMID: 11562466; PMCID: PMC58746).

3) What we wrote in the original rebuttal and in the manuscript to address this question is that we have the same mild heterozygous phenotype in SwissWebster using 2 different *Tbx1* alleles (*Tbx1^{Cre}* vs *Tbx1^{+/-}*) generated in different labs.

4) I talked extensively with the technical staff at Jackson Labs (Kaitlyn Gilland) and Transnetyx Company (Nate Nowak). Both do genome-wide SNP genotyping (usually for strain determination). Both said that the locus that will be retained around the modified allele, even over 20 years of crossing to a different strain, will be several megabases. This is because when we genotype for the *Tbx1* affected allele, we select for that region to be retained. Therefore, it will be extremely difficult, if not impossible to identify a genetic modifier from the 129 background in the vicinity of *Tbx1*.

5) We do not see bioinformatic evidence that there are novel or unusual transcripts in the *Tbx1* locus in our scRNA-seq data. This includes data from *Tbx1^{Cre}* or *Tbx1^{flox}* modified alleles versus *Mesp1^{Cre/+}* or littermate controls in SwissWebster.

We have added these points, in addition to what is already written in response to the original review, in the manuscript to lines 422-440. This section of the Discussion now reads:

“However, we did not observe cardiac defects in *Tbx1^{Cre}* heterozygous embryos or *Tbx1^{+/-}* embryos in which a different region of *Tbx1* was inactivated, as maintained in the SwissWebster background. This is in comparison to *Tbx1* heterozygous mice as maintained in C57Bl/6, in which heterozygous mice have pharyngeal arch artery defects at reduced penetrance¹⁶⁻¹⁸. This suggests that the mild phenotype observed is not due to localized genetic modifiers in the *Tbx1* locus itself but rather the genetic background.

The effect of genetic background upon the pharyngeal arch artery phenotype of *Tbx1* heterozygous mice has been known for many years. Early work noted that the original 129 strain, in which the *Tbx1^{Cre}*, *Tbx1^{+/+}* and *Tbx1^{+/-}* mutations were generated, has a suppressive effect upon the penetrance of such defects (similarly to SwissWebster), while the C57Bl/6 background has an enhancing effect, and research excluded that this modifying effect was due to modifications in the localized genomic region⁵⁹. Further, we have not observed the presence of novel or unusual transcripts in the *Tbx1* locus in different alleles in our scRNA-seq data. Finally, this work uses the null phenotype to draw conclusions about the role of *Tbx1* in development, not the heterozygous phenotype. The null phenotype, as two decades of publications indicate, is remarkably similar across multiple genetic backgrounds. Therefore, the genetic background-dependence of the heterozygous phenotype would not affect the major conclusions of the manuscript.”